# High order expression dependencies finely resolve cryptic states and subtypes in single cell data

Abel Jansma[1,2,5], Yuelin Yao[1,3,5], Jareth Wolfe [1], Luigi Del Debbio[2], Sjoerd V Beentjes [1,4], Chris P Ponting [1,6✉] & Ava Khamseh [1,2,3,6✉]

## Abstract

**Single cells are typically typed by clustering into discrete locations in reduced dimensional transcriptome space. Here we introduce Stator, a data-driven method that identifies cell (sub)types and states without relying on cells' local proximity in transcriptome space. Stator labels the same single cell multiply, not just by type and subtype, but also by state such as activation, maturity or cell cycle sub-phase, through deriving higher-order gene expression dependencies from a sparse gene-by-cell expression matrix. Stator's finer resolution is clear from analyses of mouse embryonic brain, and human healthy or diseased liver. Rather than only coarse-scale labels of cell type, Stator further resolves cell types into subtypes, and these subtypes into stages of maturity and/or cell cycle phases, and yet further into portions of these phases. Among cryptically homogeneous embryonic cells, for example, Stator finds 34 distinct radial glia states whose gene expression forecasts their future GABAergic or glutamatergic neuronal fate. Further, Stator's fine resolution of liver cancer states reveals expression programmes that predict patient survival. We provide Stator as a Nextflow pipeline and Shiny App.**

**Keywords** Higher-order Gene Expression Dependencies; Single-cell Transcriptomics; Structure Learning; Cell State; Cell Cycle Phases
**Subject Categories** Chromatin, Transcription & Genomics; Computational Biology; Methods & Resources

## Introduction

To attribute disease to cell type, and molecular features of cells to disease state, we need to define and distinguish cell types, subtypes and states (Dann et al, 2023). We follow others (Fleck et al, 2023; Morris et al, 2019; Wagner et al, 2016) by distinguishing cell types —with their more permanent phenotypic features—from cell states, whose features are transient and can be elicited by stimulus; cell

subtypes are sub-populations of the same cell type that share distinctive features. The Human Cell Atlas (Regev et al, 2017) has taken a step in this direction by seeking definition of all human cell types and their molecular features, most often gene expression, within a multidimensional 'cell space' (Regev et al, 2017). Typing of cells is easiest when their lineages are well separated, and hardest when they are distinguished only by state (such as cell cycle phase, level of maturity or response to stimulus) or spatial location.

Two stages of dimensionality reduction are commonly used in scRNA-seq analysis pipelines. The first is the projection of cells into e.g., $K \leq 50$ dimensions (or e.g., $K$ determined via a scree plot for Principal Component Analysis (PCA)) using methods that, whilst not fully preserving Euclidean distances in lower dimensions, produce an embedding whose distances are still quantitatively meaningful. This includes PCA, which features in Scanpy (Wolf et al, 2018) and Seurat's (Stuart et al, 2019) default pipelines. This dimensionality reduction is done to reduce noise and to avoid the curse of dimensionality in downstream analyses that rely on the quantification of Euclidean distances, such as clustering, which would otherwise break down in high dimensions (Aggarwal et al, 2001). Clustering is often used to define a cell type as a collection of cells that group more closely in gene expression space than other cells. This approach has yielded cell-type definitions at relatively low resolution, but requires additional analyses to begin resolving states within continuous trajectories of cell-state change (Dann et al, 2022; Kotliar et al, 2019; Ponting, 2019). The second stage of dimensionality reduction, often used in scRNA-seq analysis pipelines to date, further reduces the K-dimensional space to 2 or 3 dimensions, often using t-distributed Stochastic Neighbour Embedding (tSNE) (van der Maaten and Hinton, 2008) or Uniform Manifold Approximation and Projection (UMAP) (McInnes et al, 2020), for qualitative and exploratory analysis through visual inspection. Due to the lack of guarantees on distance preservation (for tSNE and UMAP), such extreme dimensionality reduction (even for PCA) inevitably results in significant distortions (Chari and Pachter, 2023; Cooley et al, 2022).

Cells adopt a continuum of states, representing cellular activities such as the cell cycle or responses to stimuli (Kotliar et al, 2019; Xia and Yanai, 2019). Labelling cells only by type thus does not finely resolve their dynamic behaviour such as during development or

[1]MRC Human Genetics Unit, Institute of Genetics & Cancer, University of Edinburgh, Edinburgh EH4 2XU, UK. [2]Higgs Centre for Theoretical Physics, School of Physics & Astronomy, University of Edinburgh, Edinburgh EH9 3FD, UK. [3]School of Informatics, University of Edinburgh, Edinburgh EH8 9AB, UK. [4]School of Mathematics, University of Edinburgh, Edinburgh EH9 3FD, UK. [5]These authors contributed equally: Abel Jansma, Yuelin Yao. [6]These authors contributed equally: Chris P Ponting, Ava Khamseh. ✉E-mail: chris.ponting@ed.ac.uk; ava.khamseh@ed.ac.uk

disease (Morris, 2019). Cell states are currently predicted by PCA (Shalek et al, 2014; Steuerman et al, 2018), Independent Component Analysis (ICA) or Non-Negative Matrix Factorisation (NMF) (Puram et al, 2017; Saunders et al, 2018). However, components or factors inferred by these algorithms may not faithfully or finely resolve cellular processes. States previously predicted by NMF among cancer cells, for example, include non-specific descriptors, such as 'stress', 'metal response' and 'basal' (Barkley et al, 2022).

Automatic cell annotation methods, such as CellTypist (Conde et al, 2022) and foundation models such as (Cui et al, 2023), are not intended to identify cell (sub)types de novo in a data-driven manner. This is because training of these models requires pre-existing cell-type annotations. Such cell-type annotations are curated, and so are susceptible to subjective bias (see 'Results'). These methods are also not intended to annotate states present across diverse cell (sub)types.

To identify both cell (sub)types and states at high resolution, we introduce Stator, which eschews cell clustering and instead defines states using the coordinated expression and non-expression of genes in single cells. Higher resolution is achieved by taking advantage of expression interactions at higher than pairwise ($3 \leq n \leq 7$). Expression interactions that commonly co-occur in cells are gathered together as a single state label. The method yields biologically compelling labels of type, subtype and state for cells in healthy and disease contexts without invoking concepts of expression space or pseudotime. These states are neither necessarily proximal in gene expression space nor necessarily categorical, thereby capturing the continuous nature of cell states and, in some cases, previously defined types. As with all cell state or (sub)type markers, Stator labels do not necessarily imply molecular mechanism. Rather, Stator reveals molecular and cellular heterogeneity and dynamics that would otherwise have been overlooked but can now be investigated experimentally. We show how Stator predicts, in a data-driven manner, sub-phases of the cell cycle, capturing transcriptional dynamics across each cell cycle phase, the future neuronal (sub)type fate of immature cell precursors, rare disease-associated human endothelial cell subtypes and cycling transformed hepatocytes whose expressed genes are predictive of liver cancer survival.

## Results

Stator's high-level workflow is illustrated in Fig. 1 with each step detailed in 'Methods'. Briefly, after performing standard Quality Control (QC) (Luecken and Theis, 2019), including doublet removal (Wolock et al, 2019), it initially restricts consideration to the most highly variable genes (HVG; often $N = 1000$) (Wolf et al, 2018) followed by binarisation of gene expression. Binarisation does not substantially alter conclusions when analysing sparse data (Bouland et al, 2021, 2023; Qiu, 2020). Input to Stator is a cell ($M$) by binary gene expression ($G$) matrix (Fig. 1A). The model-free estimator of higher-order interactions (MFI) we introduced in (Beentjes and Khamseh, 2020) then estimates $n$-point interactions among $n = 2, 3, \ldots, 7$ genes (Fig. 1B). Comparison between this estimator of dependence and other estimators such as correlation and mutual information is presented in Fig. EV1 and Appendix Fig. S1 (Jansma, 2023a). In the next step (Fig. 1C), "d-tuples" are

extracted. "d-tuples" are defined as gene tuples that significantly drive interactions ('Methods'). This step is achieved by comparing the expression of each tuple of genes in the MFI estimator to their expression under the null distribution of independence (see 'Methods' for full details).

Next, a new matrix of cell ($M$)-by-binary d-tuple ($K$) is created (Fig. 1D). Entries with 1 in the matrix indicate cells with that particular d-tuple gene expression combination; entries with 0 do not contain that given gene expression combination. Stator next performs hierarchical clustering of gene d-tuples based on these d-tuples' co-occurrence in single cells (Fig. 1E). Absence of expression for a gene is denoted by a *minus*, e.g., $G_1-$ and $G_2-$ on the rightmost branch in Fig. 1E. Crucially, this clustering takes place in a restricted space of d-tuples. Consequently, rather than a cell being placed at a single location in gene expression space, as is usual in scRNA-seq analyses, Stator allows for cells to adopt multiple biological states (Fig. 1F). We show below that a single cell can be thrice (or more) labelled (Appendix Fig. S2), for example as a radial glial-like precursor cell, as an astrocyte progenitor and a cell in G2/M cell cycle phases. Once groups of combinatorial gene signatures are identified, users can tune the modularity parameter that varies the granularity at which Stator states are resolved. Stator's memory and run time are discussed in 'Methods' and Appendix, Section A.1, Appendix Figs. S3 and S4.

Stator states are definable not just by d-tuple genes but also by other genes that are significantly differentially expressed relative to all other states or one other state (Fig. 1G): these are state-to-other DEGs (s2o-DEGs, adjusted $p$ value <0.05, $|\log_2(FC)|>0.25$) and state-to-state differentially expressed genes (s2s-DEGs, adjusted $p$ value <0.05, $|\log_2(FC)|>0.25$), respectively (Fig. 1G). For this analysis, the unbinarised expression values of all genes (not only the 1000 HVG) are considered using existing methods, e.g., (Stuart et al, 2019). The app further permits Stator states to be queried for their enrichment in previously derived annotations, such as experimental condition (healthy versus disease, different time points), or cell (sub)type or biological state labels.

To demonstrate Stator's ability to identify cell states and (sub) types, we investigated three published scRNA-seq datasets in normal and disease contexts, in embryological or adult tissue, and in human or mouse. This first set contains astrocyte and neuron progenitors from mouse late embryonic (E18) brain (10XGe-nomics, 2017), chosen because this is the developmental stage when astrocytogenesis occurs and when cortical radial glial precursors (RPs) asymmetrically divide to generate neurons in the developing mouse cortex (Akdemir et al, 2020; Rubenstein et al, 2020). The second is scRNA-seq data from human liver cells from disease (cirrhosis) and control donors (Ramachandran et al, 2019). Thirdly, we applied Stator to human liver cancer (hepatocellular carcinoma) cells (Barkley et al, 2022). Biological validity of a Stator state is provided when its d-tuple genes, s2o-DEGs and/or s2s-DEGs occur in a common cellular process and/or marker gene set; for further details see 'Methods' "Assigning labels to Stator states". We start by showing how Stator identifies cells present in only a portion of a cell cycle phase before then revealing cell subtypes and states that had hitherto not been inferred from these datasets.

For each dataset, we compare Stator with NMF (Barkley et al, 2022; Gaujoux and Seoighe, 2010). We further compare Stator's output on the RPs' dataset with (i) clustering with pairwise significance quantification (Gao et al, 2022; Stuart et al, 2019)

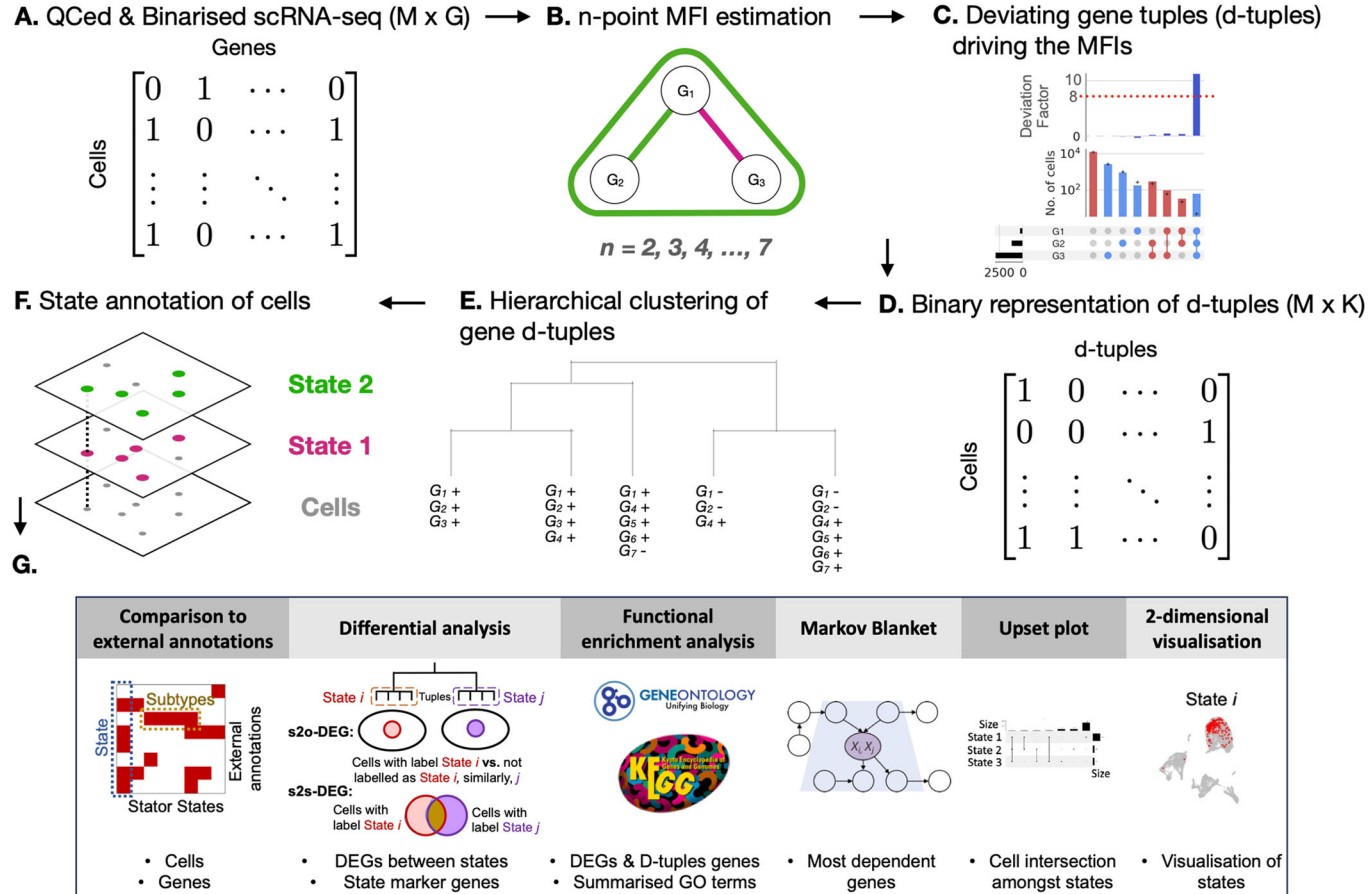

**Figure 1.    Workflow of Stator.**

Steps (**A**–**E**) are fully data-driven; steps (**F**, **G**) require biological interpretation. (**A**) A quality controlled, cell ($M$)-by-binarised gene expression ($G$) matrix is used as input. (**B**) $n$-point model-free interactions (MFI) are estimated ($n = 2, \dots , 7$) from the graph of conditional dependencies among the genes. Green edges denote positive values, red edges denote negative values, the larger green triangle represents a positive 3-point dependence. Prior to this, the graph is inferred with an MCMC graph-optimisation algorithm on an initial structure obtained by the Peter–Clark causal discovery algorithm. The graph itself is not used to claim causation, rather to improve the statistical power of detecting $n$-point interactions among genes with strong inter-dependencies. (**C**) Tuples that are significantly deviating (default: FDR < 0.05, $\log_2$(FC)>3) as compared to the null hypothesis of independence (interaction = 0) are extracted. These gene combinations are deviating tuples, or "d-tuples". The significant tuple in this example is $(G_1, G_2, G_3) = (1,1,1)$ but d-tuples containing zero-values representing unexpressed genes are also found. Red and blue represent even and odd numbers of expressed genes (equal to 1) in the MFI. (**D**) A binary cell ($M$)-by-d-tuple ($K$) matrix is created. Entries with 1 indicate a cell containing a significant given tuple, in this example cells with $(G_1, G_2, G_3) = (1,1,1)$. Entries with a zero represent cells not containing the d-tuple. The matrix is created using all $K$ significant interactions and corresponding d-tuples. (**E**) Hierarchical clustering of d-tuples (rather than cells) is performed to group any d-tuples that co-occur unusually often in single cells. The dendrogram is cut, by default, at a Dice similarity that maximises the modularity score (Newman, 2006), but is adjustable. This procedure results in groups of d-tuples that can contain both the presence and absence of a gene's expression. (**F**) At this stage, the user annotates and interprets the groups of d-tuple genes to infer cell states. Unlike clustering of cells, this procedure can result in cells that exist in multiple biological states simultaneously. (**G**) A Shiny App in R enables the user to compare Stator states against external annotations, such as other data-driven or expert annotations. Left: The horizontal box represents the significant enrichment of several Stator states in cells with a specific externally annotated cell type, demonstrating the existence of multiple cell subtypes that could be explored further. The vertical box represents Stator states spanning multiple externally annotated cell types, representing non-cell-type restricted biological states, e.g., cell cycle phases. The user can also choose to compare Stator state enrichment against biological conditions of an experiment. Stator's Shiny App allows further integrative analyses, such as differential expression of Stator states or Gene Ontology term enrichment (Aleksander et al, 2023; Ashburner et al, 2000; Sayols, 2023; Wu et al, 2021; Yu et al, 2012). We provide a step-by-step guidance for labelling Stator states in 'Methods', section "Assigning labels to Stator States".

(Fig. 2A; Appendix Fig. S6), (ii) LDVAE (deep learning-based) (Gayoso et al, 2022; Svensson et al, 2020), (iii) LDA (topic modelling) (Blei et al, 2003; Gayoso et al, 2022; Srivastava and Sutton, 2017), and (iv) its cell cycle states with Tricycle (Zheng et al, 2022b). Analyses (ii)–(iv) are presented in 'Methods' "Comparison with other methods".

## Stator identifies states in seemingly homogeneous cells

We first applied Stator to 11,950 E18 mouse brain cells (Methods 'Datasets'). These highly express canonical markers (e.g., *Slc1a3*, *Mt3* and *Mfge8* (Yuzwa et al, 2017)) of embryonic radial glial precursors (RPs), which later develop into astrocytes or neurons via

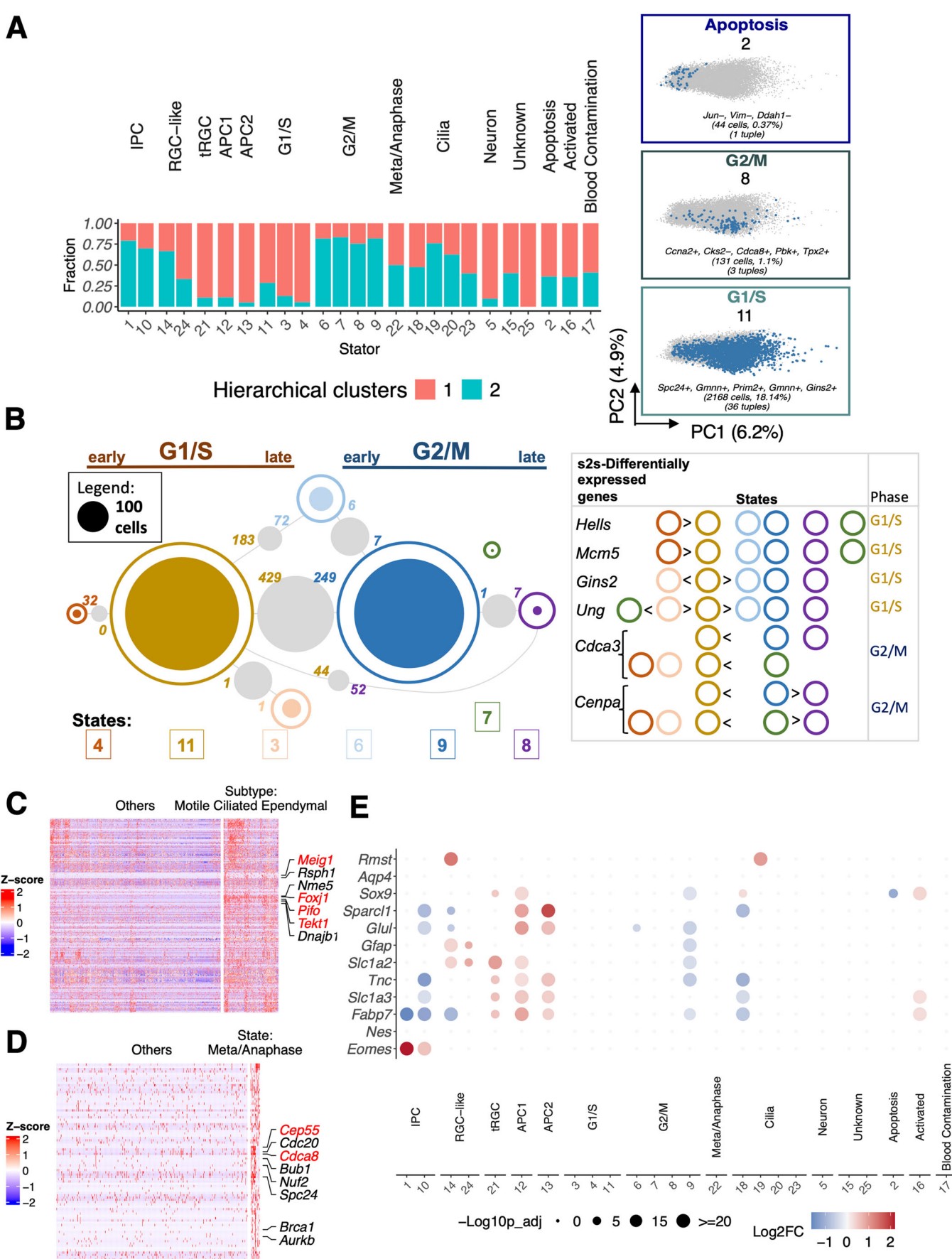

**Figure 2.  Stator identifies cell states in seemingly homogeneous mouse embryo radial glial cell-like precursor cells.**

Stator identifies 25 signatures at maximum modularity. We have labelled 23 of 25 Stator states by performing differential gene expression analysis between cells in one state and all other cells, followed by gene enrichment (GO/KEGG) analyses. The significant differentially expressed genes were also compared with known gene markers of cell types and states. In such cases, we required at least three marker genes to be highly expressed. (A) Barplot colours indicate the proportion of cells captured by each cluster following hierarchical clustering of cells (see Appendix Fig. S6) that resulted in two significantly different clusters only. Right-hand side: three exemplar Stator states (#2, 8, 11) are highlighted in a PCA embedding of the unbinarised expression data, and annotated with the number of cells they label, the d-tuples from which they are defined, and their five most common d-tuples. (B) Numbers of cells labelled with any one of 7 cell cycle states (# 3, 4, 6, 7, 8, 9 and 11); areas of circles are proportional to their number (see legend). Filled circles indicate numbers of cells labelled with only one of these single cell cycle states. Grey circles' areas indicate numbers of cells labelled with two cell cycle states, those indicated by lines. Numbers of significantly differentially expressed genes between cell cycle state pairs (i.e., s2s-DEGs) are provided between the two states being compared; their colours refer to the state showing higher expression. For clarity, state pairs with $\geq$ 25 cells are shown. DEGs between any two states, including state pairs with fewer than 25 co-labelled cells, are provided in Dataset EV5. Appendix Fig. S8A additionally provides the number of co-labelled cells between any two states. Right: s2s-DEGs are indicated by ">" or "<" symbols; for example, *Hells* mRNA expression is significantly higher in State # 4 over States # 11, 6, 9, 8 and 7. Early/late G1/S or G2/M cell cycle phase labels (top) were assigned using these mRNAs' cell cycle phases known from high-throughput (top right; (Giotti et al, 2018)) and targeted experiments (*Ung* mRNA in late G1/S (Slupphaug et al, 1991) and *Cenpa* in G2 (Shelby et al, 1997)). (C) Heatmap of expression level (z-score) for genes upregulated in state #18, versus other states, for cells in State #18 and a random selection of cells from other groups ($n = 1500$). Z-scores are computed on a gene-by-gene basis by subtracting the mean and then dividing by the standard deviation throughout this study. Genes were ordered by hierarchical clustering. Upregulated genes are significantly involved in Cilium assembly (GO:0060271; $q = 3 \times 10^{-11}$). (D) Heatmap of expression level (z-score) for genes upregulated in state #22, versus other states, for cells in state #22 and a random selection of cells from other groups ($n = 500$). Genes were ordered by hierarchical clustering. Upregulated genes reveal state of metaphase/anaphase. (E) Dot plot illustrating differential expression of astrocytogenesis marker genes across all Stator states. The size of the dots represents the $-\log_{10}$(Seurat p-val-adj) from differential expression testing between a state and all other states. Colour intensity represents the $\log_2$(FC) of gene expression.

intermediate progenitor (IP) cells. Upon clustering, these cells appear to be highly homogeneous without being separable into, for example, cells in cell cycle phases. Specifically, they could not be discriminated, using hierarchical clustering with significance quantification of clusters (Gao et al, 2022) on the original (unbinarised) expression space, beyond two significantly different ($p<0.05$ Bonferroni corrected, 'Methods') and robust clusters of 6485 and 5465 cells, respectively (Appendix Fig. S6).

By contrast, Stator predicted 25 States (Fig. 2A; Datasets EV1 and EV2), with the optimal Dice dissimilarity of 0.95. The majority (75.7%; $N = 9044$) of cells occupy one or more state, and 34.7% (4151) of cells are unique to a single state (Appendix Fig. S2A,B). Some states (e.g., #2) are localised in a PCA embedding, but most are not (e.g., #8 and #11). d-tuples in 7 states contained known cell cycle marker genes (Fischer et al, 2016; Tirosh et al, 2016): in the largest (#11; 2168 cells), nearly all d-tuples contained one or more known G1/S-phase markers' genes (33 of 36 d-tuples; 92%): 23 d-tuples contained either *Gins2* or *Gmnn* or both (20, 14 or 11 d-tuples, respectively). For illustration, one d-tuple contains three known S-phase expressed genes (*Dnmt1*, *Hells*, *Pcna*; (Giotti et al, 2018)) with their coordinated expression (i.e., 1-values) in 258 cells, which corresponds to >6.5-fold deviation (default is set at eightfold) from the null hypothesis of independent expression (FDR <0.01, default is set at 0.05); 35 other d-tuples co-occur sufficiently with this d-tuple in these cells to be combined into this single Stator state (#11).

To assess biological validity of these Stator predictions—whether they might indicate cell types, subtypes or states—we undertook differential gene expression analysis (Datasets EV3, EV4, and EV5). The 7 states' s2o-DEGs were predominantly cell cycle markers. Specifically, of 45 genes that were among the 10 most significantly differentially expressed in 1 or more of the set of 7 s2o-DEGs, 40 (89%) were G1/S or G2/M stage marker genes according to (Tirosh et al, 2016) (Table S5) and (Fischer et al, 2016) (Table S10), confirming them as cell cycle states. Many s2s-DEGs were also cell cycle marker genes (Fig. 2B). Pairs of Stator states with s2s-DEGs are transcriptomically non-identical, even if they show some transcriptomic similarity, as expected for states located along a

continuum. Note that pairs of states are concluded to be transcriptomically non-identical when they have significant s2s-DEGs (beyond the Stator state-defining d-tuple genes).

In the second largest prediction (state #9; 2145 cells), all 34 d-tuples contained G2/M phase marker genes (Fischer et al, 2016; Giotti et al, 2018; Tirosh et al, 2016): 23 contained *Pbk*, 17 contained *Cenpa*, and 9 contained both. Stator predicted these states as G1/S (#11) and G2/M phases (#9), respectively, by their cells' transcriptomes differing by 429 and 249 s2s-DEGs Fig. 2B) including for #11: G1/S phases' marker genes (e.g., *Tuba1b*, *Rpa2*, *Mcm4*, *Tipin*, *Mcm2*, *Hat1*, *Rfc3* and *Rfc2*); and, for #9: G2/M phases' marker genes (e.g., *H2afv*, *Arl6ip1*, *Stmn1*, *Ccdc34*, *Tacc3*, *Racgap1*, *Hmgb3*, *Calm3*, and *Cenpe*), all genes that did not contribute to d-tuple definition. As expected, cells co-labelled with both states #9 and #11 preferentially expressed G1/S marker genes (*Pclaf*, *Mcm6*, *Gins2* and *Gmnn*, for example) or G2/M markers (*Pbk*, *Cenpa*, *Ccnb2* and *Cdca3*, for example) compared with cells only labelled with state #9 or with #11, respectively. Stator thus not only identifies cells that are cycling, but further differentiates cells into G1/S versus G2/M cell cycle phases. This justified labelling our method's predictions as "Stator states".

Applying the same approach (comparing Stator states' expressed d-tuple genes and s2s-DEGs with known cell cycle phase marker genes) identified five less-populated states (#3-4, #6–8) as additional cell cycle states, each transcriptionally non-equivalent with respect to states #11 (G1/S phases) and/or #9 (G2/M phases) and to each other, Fig. 2B. These five states' s2s-DEGs again included marker genes for G1/S phases (states #3-4) or G2/M phases (states #6–8) relative to #11 (G1/S) and/or #9 (G2/M). In particular, 2 G1/S cell cycle phases' marker genes (*Hells*, *Mcm5*) are significantly more highly expressed in cells in state #4 over #11, and indeed in states #3, 6, 9, 7 and 8; similarly, *Gins2* has higher expression in #11 than in #3, 6, 9 and 8, Fig. 2B.

Demanding that at least 3 s2o-DEGs are known markers of an annotation (Datasets EV2–4), we labelled the other states as either Intermediate progenitor cells (IPC) (Ruan et al, 2021), radial glial cell-like cells (RGC-like) (Zheng et al, 2022a), or astrocyte progenitor cells (APC) (Liu et al, 2022); or in the metaphase/

anaphase of the cell cycle (significant enrichment of GO:0045841 (Ashburner et al, 2000), FDR<0.05) or apoptosing or activated cells (expressing mitochondrial genome genes or intermediate early genes or activation markers (Lacar et al, 2016), respectively); or blood cell contaminants that highly expressed not just globin genes (Biagioli et al, 2009) but also *Alas2*, an erythroid-specific gene (Fig. 2). More specifically, from differential expression of s2s-DEGs *Sparc* and *Sparcl1* (Dataset EV6), states #12 and #13 appear to label APC1 and APC2, two known astrocyte progenitor cell types (Liu et al, 2022), and state #21 is associated with higher expression of truncated radial glial cell markers (*Anxa2*, *Cryab*, and *Tmem47* (Yang et al, 2022)) relative to APC1 cells (state #12). We illustrate raw gene expression differences defining states #18 (Cilia) and #22 (Metaphase/Anaphase) in Fig. 2C,D. In Fig. 2E, we show how expression of the few established markers of precursor and intermediate cell states (Akdemir et al, 2020; Götz et al, 2015) varies across the 25 Stator states.

Stator was also applied to a second subset ($N = 11,950$) of the E18 RPs, independent of the first, replicating APC1, APC2, IPC and RGC-like states, multiple G1/S and G2/M cell cycle phases' states, and activated and blood contamination states (Fig. EV2; Datasets EV7, EV8, and EV9; Appendix Fig. S2B). For more details, see "Stator state projection to disjoint data" and Fig. EV2C,D for reproducibility of Stator states for RPs (and Fig. EV5C,D for the neurons dataset).

In addition to clustering, we compared Stator states for the mouse embryonic RPs with those obtained by three other methods, NMF, LDVAE and LDA; see "Comparison with other methods" for details. In summary, these methods consistently replicate multiple Stator states. However, there is greater expression specificity for Stator states over NMF, LDVAE or LDA states/modules, i.e., there is higher relative expression of known gene markers for the cell state as defined by Stator than the equivalent cell state defined by NMF, LDVAE or LDA (Table 1). Furthermore, these other methods lack uncertainty quantification for the reported gene modules, which can result in reported modules not being biologically identifiable (e.g., NMF results on embryonic RPs, Fig. 7A). Had these methods benefited from uncertainty quantification and FDR control, similar to Stator, then some reported modules may then be "statistically zero" which would avoid false positives and over-interpretation of results. We also compared Stator's cell cycle states to Tricycle (Zheng et al, 2022b) analysis of this data (see Fig. 8 in "Comparison with other methods").

## Cell cycle states in embryonic neurons and RPs

We next showed that Stator can also identify cells in G1/S or G2/M phases within an admixture of two cell types, neurons and RPs ($n = 13,605$ and 5395), from a single E18 mouse brain (10XGenomics, 2017) ('Methods'). In all, Stator predicted 110 states from these combined cells (Datasets EV10 and EV2), of which 34 were specific ($\geq 99\%$) to neurons, and 19 to RPs. The remaining 57—common to both neurons and RPs—annotate cells that are dispersed in whole transcriptome space. The median number of predicted states for a cell was 3 (Appendix Fig. S2C).

Stator does not rely on Euclidean distances, and thus does not require the first-stage dimensionality reduction to $\leq 50$ dimensions to avoid the curse of dimensionality, nor does it require the 2D

visualisation in the second step. Moreover, because Stator does not rely on the proximity of cells in expression space, it permits different subpopulations to co-exist in the same biological state. For example, cells of different type, here e.g., RPs and early neurons, can exist in the same biological state, e.g., G2/M phases of the cell cycle. If the proximity of cells in expression space is influenced most by cell type, then states attained by multiple cell types will often be missed. For example, Fig. 3A presents a heterogeneous dataset containing a combination of neurons (predominantly left) and RPs (right), for which PC1 explains $\sim 10\%$ of the total variation, with the remaining PCs explaining <2% each. Yet, each of these cell types clearly includes some cells occupying the same cell cycle state, e.g., G2/M. Stator readily detects such states from homogeneous cells or the combination of two cell types.

Among 12 cell cycle Stator predictions were G1/S (#51), S/G2 (#55), early G2/M (#58) and late G2/M (#59) states (Figs. 3 and EV3; Datasets EV10–12). RP cells in G1/S cell cycle phases (States #51 and #55) had not previously been detected in this embryonic stage by cell cycle classification (Yuzwa et al, 2017). An additional state (#57) involving cells that were predominantly labelled neurons (86%) showed multiple s2s-DEG markers for both newborn neurons (e.g., *Dcx*, *Tubb3*, *Gad2*, *Stmn2*) and G2/M phases (e.g., *Cenpa* and *Cdca3*) and hence is likely a post-G0 phase neuron state (Datasets EV10–12).

Eight G2/M states contained *minus* gene markers (i.e., those without expression evidence) that are nonetheless known markers of G2/M phases: *Cenpa* for States #1 and #2, *Cks2* #53 and #54, *Cenpf* #56, *Racgap1* #58, *Ube2c* #60 or *Cdca8* #61. We found similar combinatorial markers for the cell cycle in RPs, which included S-phase states without expression of one of three S-phase markers, *Ung*, *Mcm3*, *Gins2*. To investigate whether these states demarcate portions (sub-phases) of G2/M cell cycle phases, we highlighted cells from an external dataset along a cell cycle projection that expressed all but one G2/M or S-phase cell cycle marker genes, specifically the *minus* gene marker (Figs. 3 and EV3). This illustrated that cells in these Stator states differentially occupied parts of the cell cycle continuum, consistent with cell cycle sub-phases. For example, Stator states differentiated between early G2/M (d-tuples with an absence of *Ube2c* or *Cenpa* expression, i.e., *Ube2c-*, or *Cenpa-*), early- or mid-G2/M (*Cdc25c-* or *Racgap1-*), or mid-to-late G2/M (*Cks2-* or *Cdca8-* or *Cenpf-*) or early S-phase (*Gins2-*), mid-S-phase (*Mcm3-*) and late S-phase (*Ung-*) (Fig. 3B). Rather than single genes, it is the combinatorial gene expression pattern that provides high resolution of cell states. This is because populations of cells defined only by the expression of various combinations of cell cycle marker genes, without requiring that the *minus* gene is unexpressed, are not localised to a cell cycle (sub)-phase (Fig. EV4).

Having successfully identified cell cycle sub-phases for RPs and for a combined RP and neuron dataset, we next used Stator to identify additional cell states within the combined dataset. Embryonic RPs were previously described as homogeneous at E17.5 (Yuzwa et al, 2017). By contrast, 47 Stator states could be labelled as RPs either because their d-tuple genes were embryonic RP markers (Yuzwa et al, 2017) or else they significantly more highly expressed such genes over all other states (Datasets EV10, EV11 and EV2). This number of statistically distinct RP states is an order of magnitude greater than predicted in a previous study of neurogenesis (Shin et al, 2015). Of these 47 RP states, 21 were transcriptionally heterogeneous owing to their d-tuples including a

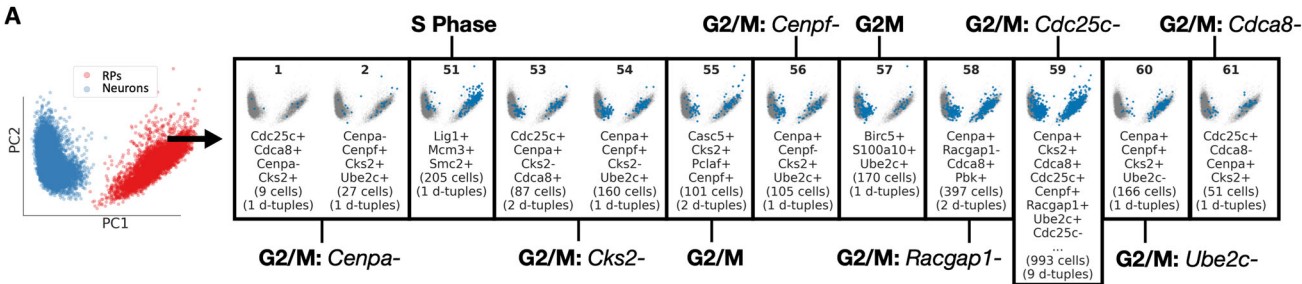

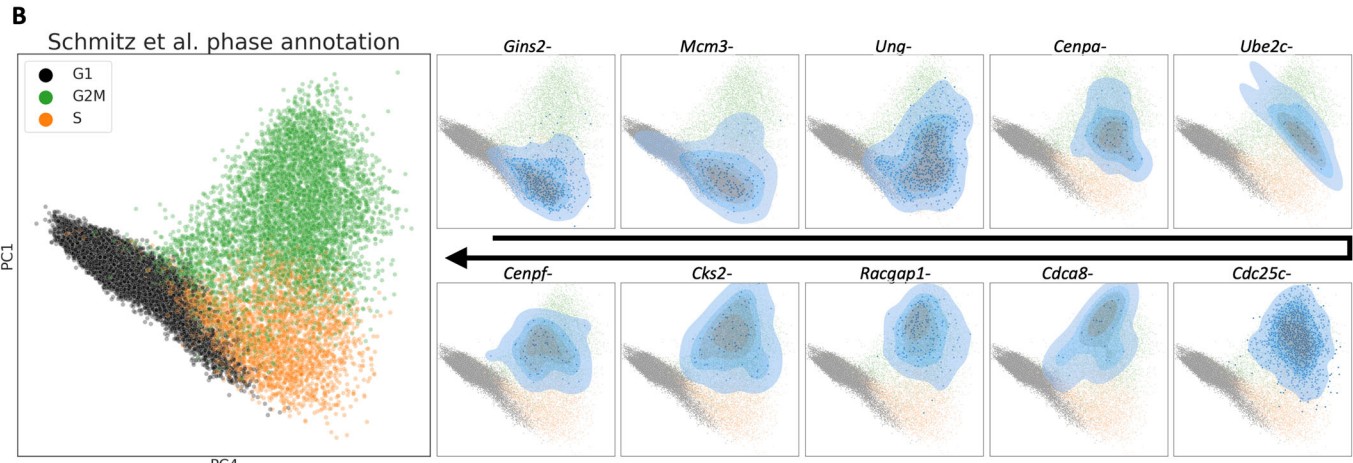

**Figure 3. Stator identifies states present in two different cell types.**

(A) Stator states labelling both developmental neurons and radial glial cell precursors (RPs) from an E18 mouse brain dataset (10XGenomics, 2017). Stator identifies 12 states that include one or more cell cycle phase gene markers as d-tuple genes, and that delocalise throughout expression space. PC1 explains ∼ 10% of the variation. State #51's d-tuple genes (*Lig1*+, *Mcm3*+, *Smc2*+) encode proteins active in S phase, while the remaining 11 express known markers of G2/M phase (Giotti et al, 2018; Riba et al, 2022; Tirosh et al, 2016). It is notable that many G2/M states are defined by an absence of expression of d-tuple genes that are nevertheless known G2/M marker genes (*Cenpa*, *Cks2*, *Cenpf*, *Racgap1*, *Cdc25c*, *Ube2c* and *Cdca8*). (B) Left: Externally-derived cell cycle annotations of a mouse brain dataset sourced from five different experiments (Schmitz et al, 2022), with cells from the 10XGenomics E18 mouse dataset GSE93421 removed, mostly separate along principal components 1 and 4. Right: Cells and embedding as left, but marked by the expression of all but one of the marker genes. Different intensities of blue represent 'densities' of cells in the 2D embedding. Note that since a two-dimensional PCA embedding can distort distances, densities cannot be directly interpreted, so no legend or axis is shown. Each state contains the expression of known cell cycle markers as well as a single non-expressed cell cycle marker gene (indicated above each box), predicted to be a combinatorial marker by Stator in RPs and/or neurons. These gene combinations thus demarcate cell cycle sub-phases, and a suggested ordering is shown here.

*minus* gene marker, such as *Hes5* (states #26-27), *Qk* (#29), and *Pax6* (#34), each of which is involved in neural progenitor cell fate choice ((Ericson et al, 1997); Imayoshi and Kageyama, 2014; Takeuchi et al, 2020).

These RP states were transcriptionally heterogeneous (Appendix Fig. S9; Dataset EV13): (i) 13 RP states yielded large number of s2s-DEGs, compared with the most populous RP state (#44); (ii) 3 states (#13, #36 and #39) showed significantly lower expression of 7 core RP genes, *Mt3*, *Phgdh*, *Slc1a3*, *Ddah1*, *Aldoc*, *Vim*, and *Fabp7* (Yuzwa et al, 2017) than state #44; (iii) 15 states contained G2/M cell cycle phases' marker genes among their s2s-DEGs relative to state #40; (iv) and 15 states yielded large ribosomal subunit genes as s2s-DEGs with state #40, a transcriptional signature of embryonic RP reactivation to become activated neural stem cells (Borrett et al, 2022; Dulken et al, 2017).

Thirty-four RP states had neuronal marker genes among their s2s-DEGs with states #40 or #44 (Datasets EV12 and EV13),

consistent with these embryonic RPs having a future neuronal fate. Seventeen states co-express *Ascl1* and *Neurog2* (often with *Gadd45g*, a transcriptional target of ASCL1), two genes that are expressed in more mature cells in a mutually exclusive manner (Parras et al, 2002). These states thus likely label early neural progenitor cells that have yet to attain their GABAergic (*Ascl1*) or glutamatergic (*Neurog2*) neuronal fate in the forebrain.

Finally, we projected Stator RNA states from the E18 merged RPs and neuron dataset, into an independent scRNA-seq dataset (10XGenomics, 2021) of 5000 cells (3343 cells after quality control) acquired in the same biological condition that has an additional modality, namely scATAC-seq, to investigate the heterogeneity of states using an orthogonal mode of data in a disjoint dataset. Transcriptomic heterogeneity was retained across the two datasets and was additionally recapitulated by open chromatin status (see Supplementary section "Comparison with multimodal data" Appendix Fig. S10).

## Neuronal states

For our final analysis of embryonic mouse brain cells, we analysed two disjoint subsets each containing 19,000 mouse E18 neurons. As the modularity was maximised at Dice dissimilarity of 0.97 and 0.91, respectively, we applied a mean similarity of 0.94, resulting in 29 states in each (Datasets EV14 and EV15), allowing us to compare the disjoint subsets at equivalent resolution. The number of predicted states per cell is presented in Appendix Fig. S2D,E.

This number of Stator states was five-fold more than the four pairwise significantly distinct clusters found by hierarchical clustering in expression space for the first disjoint dataset (Appendix Fig. S11). Stator successfully distinguished striatal medium spiny neurons (MSN) from interneurons by expression of known marker genes (e.g., *Ngef*, *Nrxn1*, *Pou3f1*, *Tshz2* (Arlotta et al, 2008; Fuccillo et al, 2015; Su-Feher et al, 2022), versus *Arx*, *Epha5*, *Lhx6*, *Prox1* (Li et al, 2022; Miyoshi et al, 2015; Poirier et al, 2004), Fig. 4A; Datasets EV2, EV16 and 17). It further separated MSNs into their two known subtypes, Direct or Indirect pathway cells (Cirnaru et al, 2021; Cui et al, 2013) via markers: Direct: *Ebf1*, *Foxp1*, *Isl1*, *Nrxn1*, *Zfhx3* and *Zfp503* (Fuccillo et al, 2015; Li et al, 2022; Precious et al, 2016; Shang et al, 2022; Zhang et al, 2019) and Indirect: *Adora2*, *Ebf1*, *Gucy1a3* and *Gucy1b3* (Li et al, 2022) (Fig. 4B), and separated interneurons into *Htr3a* and/or *Npy* expressing subtypes (Tremblay et al, 2016) (Datasets EV2, EV16 and 17). Three RP-like states were additionally detected (Fig. 4C). States could be further labelled as early or late via markers of neuronal maturation (Rubenstein et al, 2020), specifically the temporal sequence of expression of *Dlx2*, *Dlx1*, *Dlx6os1* and *Dlx6* genes, and the later expression of MSN or interneuron markers (Fig. 4D) (Liu et al, 1997).

Increasing the resolution of Stator state identification can resolve multiple constituent biological states. At a Dice dissimilarity of 0.94, Stator's state #26 labelled neural precursor cells, as evidenced by high expression of *Zeb2*, *Mdk*, *Ctnna1*, *Arx* and *Prox1*. Nevertheless, this state was found to be a composite of three component sub-states largely following the branching order of co-occurring d-tuples (see Fig. 1D,E; 'Methods'). From their d-tuple genes, these sub-states are readily distinguished as labelling G2/M cell cycle phases, neural stem cells and newborn neuronal precursors, respectively (Appendix Fig. S12).

Stator states representing the same neuronal subtypes (e.g., interneurons, direct or indirect MSNs and late-born neurons) for the second disjoint dataset are shown in Fig. EV5 and Datasets EV18 and EV19.

## Stator resolves cell (sub)types in human liver disease at higher resolution

To demonstrate application of Stator in a human disease context, we analysed 20,000 cells from patients with uninjured or cirrhotic livers. These cells had previously been annotated as one of 12 types (Ramachandran et al, 2019). Stator identified 53 states (Dataset EV20), 28 that were differentially enriched between cirrhotic and uninjured liver sample cells (Fig. 5A). Enrichment of these states showed that Stator retrieved previous cell-type annotations, yet also found multiple states for each previous annotation (Fig. 5B). For example, cells previously annotated as being endothelial are uniquely enriched in 7 states (#4–6, #23, #32–34; green box in

Fig. 5B). To cross-reference the same states in panels A and B we use an alluvial plot. Rather than calculating enrichments for disease status (panel A) or cell-type annotations (panel B) separately, Stator also can perform an enrichment analysis for cells with Stator state labels with both previous cell-type and disease/uninjured status annotations (panel C). This shows, for example, that whereas state #4 is enriched among cirrhotic sample cells (Fig. 5A) and among annotated endothelial cells (ECs) (Fig. 5B), it is enriched not just in cirrhotic but also uninjured ECs (Fig. 5C). Equivalents to Stator states #5, #33 and #34 were found by (Ramachandran et al, 2019) (i.e., Endo(2), Endo(7) and Endo(1)) and then validated by cell staining, flow cytometry and/or immunofluorescence.

None of the seven EC-labelled Stator states co-occur in five or more cells (Fig. 5D) suggesting that each of the seven represents a distinctive EC subtype. Cross-referencing these states' s2s-DEGs (Dataset EV21) to literature EC gene markers (Przysinda et al, 2020; Trimm and Red-Horse, 2023) identified state #33 as the *ACKR1*, *WNT2*, *COL3A1* and *COL6A2*-expressing immunomodulatory subpopulation which is specific to the fibrotic niche (Ramachandran et al, 2019) (Fig. 5E). In addition, state #5 was identified as *PDPN*, *FOXC2*, and *PROX2*-expressing lymphatic-specific ECs; state #6 as a subpopulation expressing *PLAT*, whose protein level is increased in patients with liver disease (Leiper et al, 1994); and, states #23, #32 and #34 as liver-specific liver sinusoidal ECs (LSECs). The most populous state (#4) labels ECs that are not tissue or organ-specific (Dataset EV22). Differential expression of these EC subtype marker genes across these ECs is illustrated in Fig. 5F. In summary, Stator has labelled cell subtypes from among a previously homogeneous set of ECs that were scarce in this dataset (<2.5%), demonstrating its identification of rare disease-specific subtypes.

For comparison, NMF analysis was performed on the liver cirrhosis dataset from (Ramachandran et al, 2019) using the NMF procedure in (Barkley et al, 2022), described in the Methods section 'NMF procedure and gene modules'. Of the 25 NMF modules identified, 5 were significantly enriched in endothelial cells. Of these five modules, only one could be annotated based on $\geq 2$ marker genes used in the original submission (Przysinda et al, 2020; Trimm and Red-Horse, 2023): module $m_{FCN3}$ genes included *CLEC4G* and *STAB2*, both markers for liver sinusoidal endothelial cells. The other four NMF modules could only be labelled as endothelial cells of unknown type. By comparison, using the same data, seven Stator states could be labelled with $\geq 3$ marker genes from (Przysinda et al, 2020; Trimm and Red-Horse, 2023) (Fig. 5E).

## Stator recapitulates cancer cell types and NMF state annotations, yet at a higher resolution

Finally, we applied Stator to a cancer (hepatocellular carcinoma (HCC)) dataset. Stator's cancer cell states were then compared against two sets of annotations that were defined previously in (Barkley et al, 2022) by (i) clustering (Stuart et al, 2019) and comparison against reference datasets using SingleR (Aran et al, 2019), or (ii) non-smooth, non-negative matrix factorisation (nsNMF; Pascual-Montano et al, 2006).

For this analysis, 51 Stator states were predicted from 14,698 cells derived from 4 patients' hepatocellular carcinoma (HCC) samples (Barkley et al, 2022) (Dataset EV23). These states were enriched for 11 of 12 cell types previously annotated using

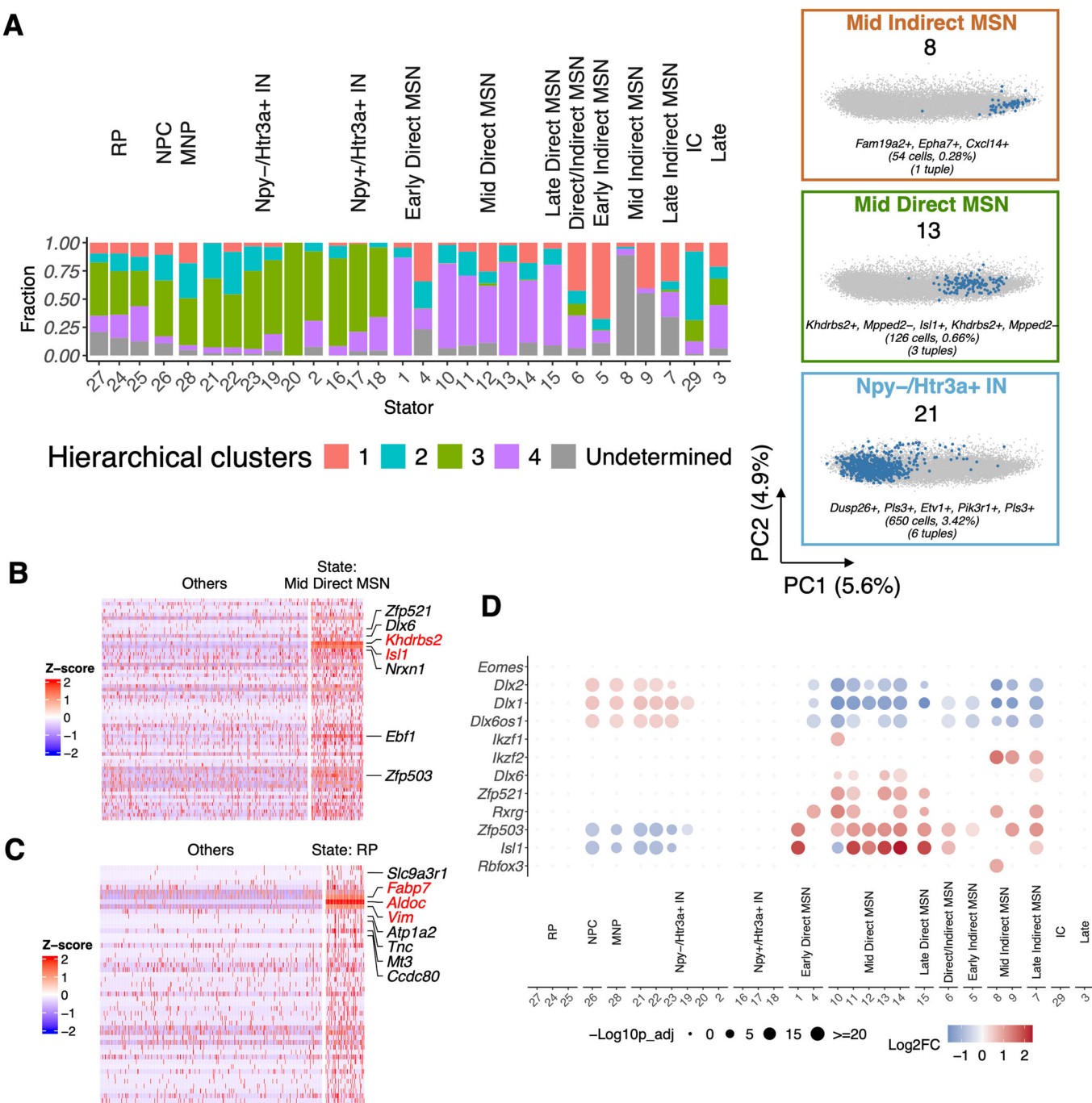

**Figure 4. Stator identifies states for developmental neurons.**

(A) Stator states for 19,000 E18 mouse cells, previously annotated as neurons. States were labelled by matching s2s- and s2o-DEGs with literature gene markers, as before (Fig. 2). The barplot colours indicate proportion of cells captured by each cluster following hierarchical clustering of cells (see Appendix Fig. S11), resulting in four significantly different clusters. On the right-hand side of panel A, three representative Stator states (# 8, 13, 21) are highlighted in a PCA embedding of the unbinarised expression data, and annotated with their total number of cells and d-tuples, as well as the five most common individual gene states across the states' d-tuples. IC intercalated cells of amygdala (*Erbb4+, Tshz1+, Foxp2+, Pbx3+*) (Kuerbitz et al, 2018; Peters et al, 2023), IN interneurons, Late late-born neurons, MNP migratory neuronal precursors (*Vax1+, Shtn1+, Pcdh9+, Tiam2+*) (Asahina et al, 2012; Coré et al, 2020; Kawauchi et al, 2003; Sapir et al, 2013), MSN medium spiny neurons, NPC neural precursor cells, RP radial glial cell precursors. (B) Heatmap of expression level (z-score) for genes upregulated in state #13: Mid Direct MSN, versus other states, for cells in state #13 and a random selection of $n = 500$ cells from other groups. (C) Heatmap of expression level (z-score) for genes upregulated in state #24, versus other states, for cells in state #24 and a random selection of $n = 500$ cells from other groups. (D) Dot plot illustrating differential expression of neurogenesis marker genes (Rubenstein et al, 2020) across all Stator states. The size of the dots represents the $-\log_{10}$(Seurat p-val-adj) from differential expression testing between a state and all other states. Colour intensity represents the $\log_2$(FC) of gene expression.

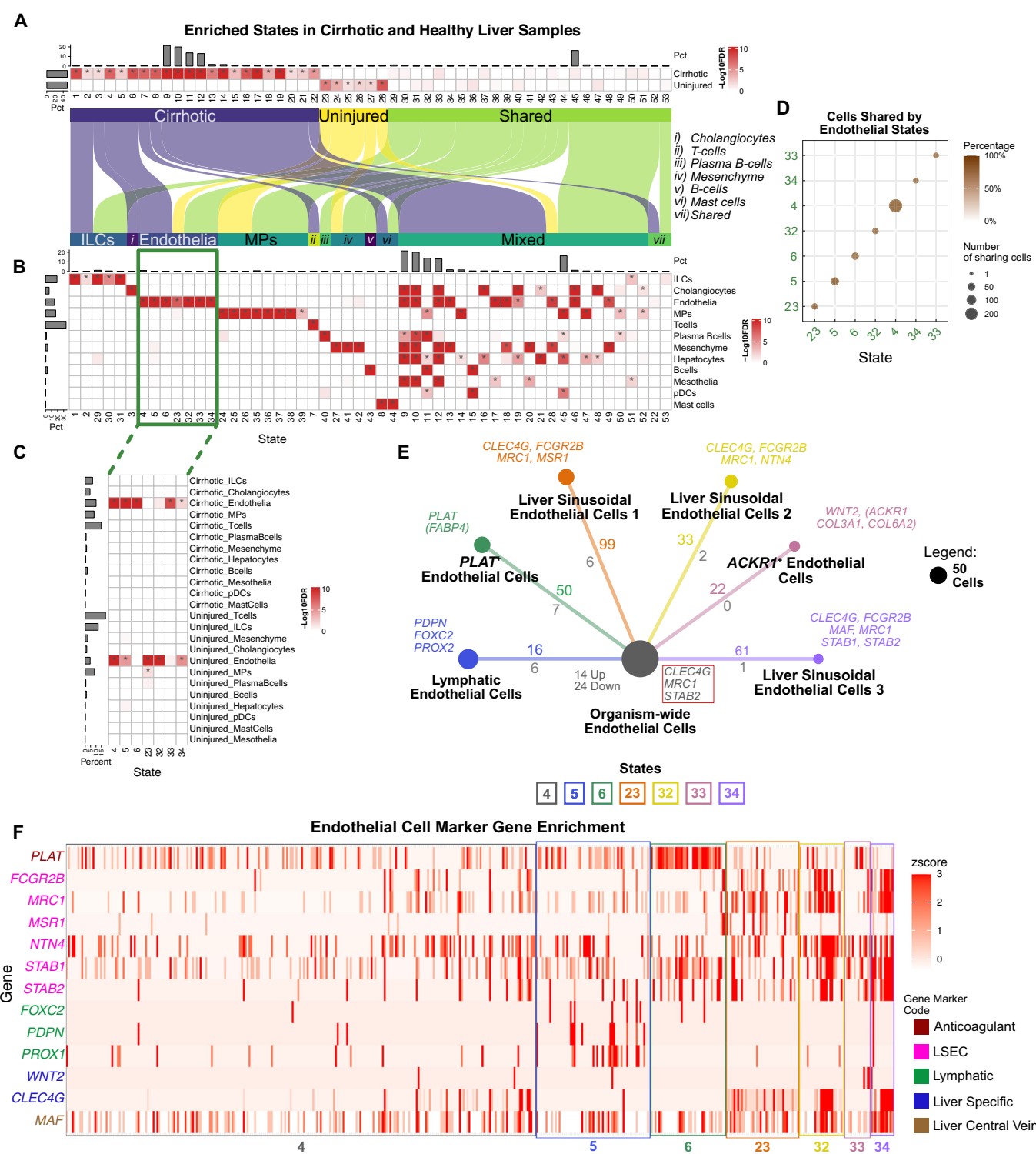

**A** Enriched States in Cirrhotic and Healthy Liver Samples

**D** Cells Shared by Endothelial States

**E** States
4  5  6  23  32  33  34

**F** Endothelial Cell Marker Gene Enrichment

clustering and SingleR (Barkley et al, 2022) (Fig. 6A); the exception, epithelial cells, were low in number (*n* = 21). As before, Stator resolved single-cell types into multiple subtypes, for example, a single B-cell annotation into 13 sub-states. Myeloid lineage (macrophages, dendritic cells [DC] and neutrophils) states and lymphoid lineage (T cells, natural killer [NK] cells and B cells) states were distinct, highlighted in Fig. 6A by blue and red boxes

respectively. Stator states were often easily annotated by their d-tuple genes. For example, state #43's d-tuple genes contained *CD4* and other T-cell markers; the myeloid lineage state #32 [*PLBD1+*, *SPI1+*, *LYZ+*, *MS4A6A+*] is in part defined by *MS4A6A*, a known marker for neutrophils, macrophage and dendritic cells (Franzén et al, 2019a); and the lymphoid lineage state #48 [*IGHG4+*, *IGKC+*, *FGFBP2+*, *IGHG1+*] is largely

**Figure 5.   Stator states in cirrhotic and healthy human liver cells previously annotated by (Ramachandran et al, 2019).**

(A) States (columns) enriched in single cells from cirrhotic or healthy liver samples (rows). (B) Heatmap showing states significantly enriched in these cells' previous annotations (indicated by asterisks). Seven states (#4, #5, #6, #23, #32, #33, and #34) are significantly enriched only in the endothelial cell (EC) annotation (green box). This panel implies that Hepatocytes, Mesothelia and pDCs do not correspond to any Stator state that is exclusively enriched for these cell types. Nevertheless, this is due to the conservative thresholds applied here. Expected correspondences emerge when thresholds are further relaxed (Dice dissimilarity >0.5, log2FC >1) where there are 6, 2 and 2 Stator states that are exclusively enriched for Hepatocyte, Mesothelia and pDC annotations, respectively (see Appendix Fig. S13). (C) States significantly enriched in both cirrhotic/uninjured status and a previous cell-type annotation (indicated by asterisks). (D) Virtually all cells with previous EC annotations are labelled with just one of the 7 EC-specific cell states. These states were not detected by the original study (Ramachandran et al, 2019) or differential abundance analysis by Milo (Dann et al, 2022). (E) Numbers of cells labelled with EC states (#4, #5, #6, #23, #32, #33 and #34); areas of circles are proportional to their number (see panel legend). For states #5 to #34, numbers of significantly differentially expressed genes between cell cycle state pairs (i.e., s2s-DEGs) are indicated relative to state #4; colours refer to the state showing higher expression. Coloured numbers indicate significantly differentially expressed genes in cells labelled with state #4 compared to cells in any other EC state (i.e., s2o-DEGs); numbers of significantly differentially expressed genes between state #4 and all other EC states (increased and decreased expression) are shown in grey. Colour-coded marker genes used to annotate cell states are provided adjacent to each state's circle; a red box contains three genes whose expression is decreased in state #4 relative to the other EC states. (F) Heatmap of expression levels (z-scores) for marker genes used to annotate each EC state. Genes are grouped and colour-coded by their associated annotation from the literature (Dataset EV2). The five categories of gene markers are colour-coded as indicated in the panel legend. Cells (columns) are enclosed within a coloured box designating the EC state labelling that cell.

defined by immunoglobulin genes, known markers for terminally differentiated B cells, i.e., plasma cells (MacParland et al, 2018).

The most populous state, #45, labels C1Q+ macrophages, an immunosuppressive population (Revel et al, 2022), annotated because 24 of 25 gene markers for these macrophages (cluster 10 of (Sharma et al, 2020)) are s2s-DEGs relative to state #40 (Dataset EV24). These are tissue-resident, rather than tumour-associated, C1Q+ macrophages because state #45 cells significantly more highly express *FOLR2*, rather than *TREM2*, relative to state #40 (Revel et al, 2022).

Twelve Stator states were enriched among cells labelled previously as hepatocytes (Barkley et al, 2022). These states labelled largely distinct sets of cells (Fig. 6B) that are transcriptionally distinguishable, as evidenced for example by large numbers of s2s-DEGs (Fig. 6C). A large minority (8–23%) of these states' s2s-DEGs are not expressed in normal hepatocytes ('Methods'), thereby reflecting their transformed status. The 12 transformed hepatocyte states showed considerable cell cycle gene expression heterogeneity. For example, State #7 expressed 6 cell cycle genes (*BIRC5, CCNA2, CCNB2, CDK1, TOP2A* and *UBE2C*) significantly more highly than the most populous State #37 (Dataset EV25). Other states (#17, 18, 19, 38) showed lower expression of these genes. These six cell cycle genes are rarely expressed in normal liver samples (Andrews et al, 2022) (Methods) and each gene's high expression is known to be prognostic of worse outcome in liver cancer (Uhlén et al, 2017).

In the previously published analysis, these HCC cells were annotated both by type and state (Barkley et al, 2022). The enrichment of Stator states in NMF states is presented in Fig. 6D. To investigate whether Stator could resolve these cells more finely, we analysed only those with both 'Hepatocytes' and 'Cycle' annotations, finding them to be enriched in 7 Stator states, most frequently in #37 (44.9% of 1447 cells) and/or #7 (40.8% of 1447 cells) (Fig. EV6). Despite their previous identical annotation, cells in these 2 Stator states are transcriptionally divergent, with 78 s2s-DEGs separating them (Dataset EV26). Cells in state #37 had increased expression of transcripts that are abundant in normal hepatocytes (34 of 34 s2s-DEGs e.g., *AHSG, PLA2G2A, CYP2E1* and *HPD*) whereas cells in state #7 had increased expression of genes that are rarely or never expressed in normal hepatocytes (13 of 44 s2s-DEGs, e.g., *TFF1, TFF2, TFF3* and *NDUFA4L2*). This suggests that Stator state #7 cells are in a more advanced state of cellular transformation than #37 cells.

To test this hypothesis, we used TCGA liver cancer prognosis data (Uhlén et al, 2015, 2017), plotting s2s-DEGs' mean expression fold change (state #7 over #37, this study; Fig. 6E, *X* axis) against the 5-year percentage survival rate (*Y* axis) for TCGA patients whose expression of this gene is above a pre-determined threshold (*Y* axis, (Uhlén et al, 2017)). This showed that genes that are more highly expressed in cells in state #7 over #37 tend to be those genes that are more highly expressed, at diagnosis, in liver cancer samples of patients with lower survival rates. Conversely, genes that are more highly expressed in state #37 over #7 tend to be genes that are more highly expressed in liver samples of patients with higher survival rates. In summary, Stator has revealed previously unappreciated HCC cancer states whose differential expression involves genes that are predictive of patient survival.

## Discussion

Single-cell transcriptomics is being translated into clinical practice for biomarkers of disease progression, patient stratification and antitumour treatment (Jia et al, 2022; Lim et al, 2023; Van de Sande et al, 2023). Concurrently, cell fate trajectories are being predicted for multiple cell types and subtypes across development (Imaz-Rosshandler et al, 2024). Nevertheless, virtually all such studies project high-dimensional single-cell transcriptome data into two or three dimensions, which distorts both clusters and developmental trajectories (Chari and Pachter, 2023). Cell localisation in expression space presents an additional problem: how to label a single cell with multiple labels (type, subtype, cell cycle phase, maturity and activity (Kotliar et al, 2019)) which are non-localised in expression space. Stator's alternative approach circumvents cell clustering and this distortion by identifying 3–7 genes with unexpectedly coordinated expression (or non-expression) across single cells. Stator's further advance is the identification of cell (sub) types and states at substantially higher resolution than existing methods.

Stator results show that a wealth of biological information can be inferred from the higher-order statistics of single-cell expression data. Evidence exists for higher-order and combinatorial genetic interactions (Antebi et al, 2017; Arnosti et al, 1996; Kuzmin et al, 2018; Watkinson et al, 2009) and pairwise quantities at different pseudotimes have been investigated (Ghazanfar et al, 2020).

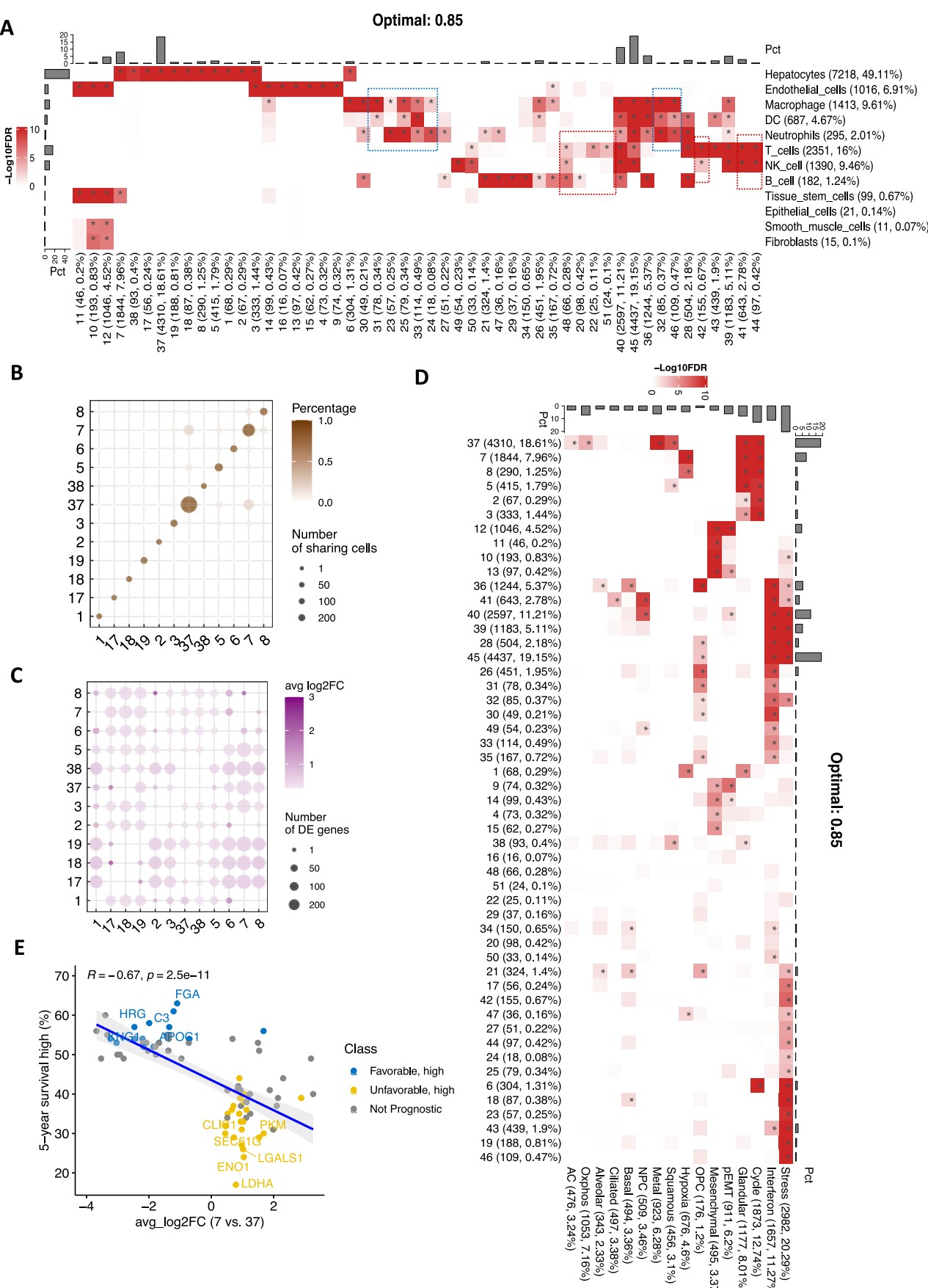

**Figure 6. Stator identifies HCC cell types and states at higher resolution than other methods.**

(A) Heatmap showing significant enrichment (asterisks) among 51 Stator states with 12 cell-type annotations previously defined by clustering followed by SingleR annotation (Aran et al, 2019; Barkley et al, 2022; Stuart et al, 2019). Stator identified multiple sub-populations for previously identified single-cell types; for example, 12 Stator states occur unusually often only among cells previously annotated as hepatocytes. (B) Since Stator allows for cells to acquire multiple states, hepatocyte states can co-label single cells. Nevertheless, most cells are labelled only as single Stator states. (C) Numbers of s2s-DEGs and their mean log2-fold change between Stator states enriched in cells previously annotated as hepatocytes. The 12 hepatocyte-enriched states are transcriptionally distinguishable. (D) Statistically significant enrichment (notified by asterisks) of Stator states (Y axis) in cells previously annotated (Barkley et al, 2022; Gaujoux and Seoighe, 2010; Puram et al, 2017) into 16 NMF-defined states (X axis). (E) Two Stator states are differentiated by genes that are predictive of liver cancer patient survival. Mean s2s-DEG expression fold change (X axis) for state #7 over #37 plotted against the percentage of 5-year survival (Y axis) for TCGA patients whose expression of this gene lies above a pre-determined threshold (Uhlén et al, 2017).

Nevertheless, the biological value of higher-order statistics in single-cell gene expression has not previously been shown. The picture we have seen emerging from applying Stator is of cells adopting a spectrum of states (or colours, in this metaphor) with their primary colour representing their strongest transcriptomic signature, most often indicating cell type. Differential expression between cells of the same type, but in two different states, filters out their primary colour thereby revealing secondary colours, representing cellular dynamics differences. This metaphor can be continued with respect to tertiary and quaternary colours, representing even more finely resolved aspects of cell state.

Finer resolution of cell state will enhance understanding of state transitions in cancer, in ageing and age-related diseases, and during development (Barkley et al, 2022; Griffiths et al, 2018; Traxler et al, 2023). Our analysis of HCC cells, for example, uncovered a cancer state predictive of patient survival (Fig. 6E). Further, our results more finely resolve neural stem and progenitor cell types that can now be investigated using mouse models. For example, 29 of 43 *minus* genes that partly define 110 neuronal and/or RP states (see above), have morphology or behaviour phenotypes when disrupted in mice (Dataset EV27; Blake et al, 2020). The roles of these genes in specifying cell state transitions during neurogenesis and in neurological disease, can now be investigated at greater cellular and developmental resolution.

Gao et al (2022) recently solved the issue of selective inference bias, or *double-dipping*, specifically when cells are clustered by optimising their transcriptional differences before calculating their transcriptional differences. Each of these two operations occurs on gene expression space. Stator clusters not cells, but rather d-tuple gene signatures, prior to s2s-DEG analysis. Even if present, Stator will mitigate selective inference, at least in part, by differences not being maximised on the same space, and by demanding significant s2s-DEGs to not just be d-tuple genes, when states are declared to be transcriptomically non-identical. The Gao et al method is also not immediately applicable here due to its reliance on clustering algorithms that compute Euclidean distances, whereas Stator relies on Dice dissimilarity.

Due to current computational constraints, Stator is limited to approximately 1000 HVG and 40,000 cells to estimate higher-order $n$-point interactions ($n = 2, 3, \ldots, 7$). Estimation of conditional dependencies contributes most to computational cost, so Stator's efficiency and accuracy could be greatly improved as new causal discovery methods are developed. In addition, accuracy could be improved by integrating biological knowledge into the dependency graphs. The limitation of up to 7-point interactions is statistical rather than computational: we did not find evidence for significant 7-point interactions in the datasets analysed. Stator takes advantage

of sparse gene-by-cell matrices, and so is not intended for analysing deep coverage transcriptomes until more sophisticated binarisation schemes are explored (e.g., Li and Quon, 2019). Other challenges relate to how Stator predictions should be interpreted, particularly those states lying on a continuum whose biology is poorly understood. Further, the resolution (i.e., Dice dissimilarity) at which states should be defined and can be interpreted will vary by dataset. Lastly, conditioning on absent gene expression in the Markov blanket (Eq. (1), 'Methods') may overlook some states despite large numbers of biologically plausible states being returned.

Stator can be applied to a variety of scRNA-seq datasets in biomedicine, including those with a temporal label (e.g., developmental or disease progression), as well as data from different individuals, to compare and contrast cell states of individuals with different disease progression trajectories, or responders and non-responders to therapy. Finally, Stator's general methodology can also be applied to other datasets with variables that are binary or can be approximated well by binarisation, such as disease comorbidities, scATAC-seq, or sparse single-cell proteomics.

# Methods

**Reagents and tools table**

| Software | |
|---|---|
| Stator nextflow pipeline | https://github.com/AJnsm/Stator/tree/main |
| pclag | https://cran.r-project.org/web/packages/pcalg/index.html |
| BiDAG | https://cran.r-project.org/web/packages/BiDAG/index.html |
| Scanpy | https://github.com/scverse/scanpy |
| Stator R shiny app code | https://github.com/YuelinYao/MFIs |
| Stator R shiny app server | https://shiny.igc.ed.ac.uk/MFIs/ |
| Complete list of packages used within Stator R shiny app (with version number) | https://github.com/YuelinYao/MFIs/blob/main/renv.lock |
| Seurat (v4.3.0) | https://github.com/satijalab/seurat/releases |
| Clusterpval | https://lucylgao.github.io/clusterpval/ |
| NMF | https://cran.r-project.org/web/packages/NMF/index.html |

| Software | |
|---|---|
| scVI | https://github.com/YosefLab/scVI/ |
| LDVAE | https://docs.scvi-tools.org/en/stable/user_guide/models/amortizedlda.html |
| LDA | https://docs.scvi-tools.org/en/stable/user_guide/models/amortizedlda.html |
| Milo | https://github.com/MarioniLab/miloR |
| Tricycle | https://bioconductor.org/packages/release/bioc/html/tricycle.html |

## n-point interaction estimation

In previous work, we developed a model-independent estimator of higher-order interactions amongst binary variables (Beentjes and Khamseh, 2020). Here, we refer to the multiplicative interaction in (Beentjes and Khamseh, 2020) as Model-Free Interaction (MFI) due to its definition being without reference to any subjective parametric model, but in terms of probabilities and their expectation values. Similar notions (for 2-point interactions) have been proposed in the statistics literature (Hernan and Robins, 2023; VanderWeele and Knol, 2014). For completeness, we summarise the main definitions and interpretations of MFIs (Beentjes and Khamseh, 2020) below. A 2-point MFI is defined, and can be rewritten, as follows:

$$
\begin{aligned}
I_{G_i,G_j} &= \log\left(\frac{p(G_i=1,G_j=1|\underline{G}=0)p(G_i=0,G_j=0|\underline{G}=0)}{p(G_i=0,G_j=1|\underline{G}=0)p(G_i=1,G_j=0|\underline{G}=0)}\right) \\
&= \log\left(\frac{p(G_i=1,G_j=1|\underline{G}=0)}{p(G_i=0,G_j=1|\underline{G}=0)}\right) - \log\left(\frac{p(G_i=1,G_j=0|\underline{G}=0)}{p(G_i=0,G_j=0|\underline{G}=0)}\right) \\
&= \log\left(\frac{\mathbb{E}[G_i|G_j=1,\underline{G}=0]\,(1-\mathbb{E}[G_i|G_j=0,\underline{G}=0])}{\mathbb{E}[G_i|G_j=0,\underline{G}=0]\,(1-\mathbb{E}[G_i|G_j=1,\underline{G}=0])}\right),
\end{aligned}
$$

(1)

where $\underline{G}$ is the set of all other genes, aside from $G_i$ and $G_j$, that are not independent of $G_i$ and $G_j$. The first line in Eq. (1) has the interpretation of a generalised conditional log-odds ratio and is symmetric in $G_i$ and $G_j$. The second line provides the following interpretation: "Does the likelihood of gene $G_i$'s expression being *on* vs *off* depend on the status of gene $G_j$'s expression?". To elaborate further, the first term represents the likelihood of gene $G_i$ being *on* vs *off*, whilst gene $G_j$ is *on*, while the second term represented the same quantity with gene $G_j$ is *off*. If the expression values of the two genes $G_i$ and $G_j$ are completely independent of each other, then these two terms cancel and result in a zero interaction as desired. The third line represents the same quantity in terms of expectation values, which are then taken as averages over the data for estimating the interactions. Uncertainties in these estimates are quantified via the bootstrap procedure (Efron, 1979). In (Beentjes and Khamseh, 2020) we generalised this definition and estimator to $n$-point interactions. For example, a 3-point interaction, where $p(G_{i,j,k}=1,1,1)$ is shorthand for $p(G_i=1,G_j=1,G_k=1|\underline{G}=0)$ and so on, is defined

as follows:

$$
\begin{aligned}
&I_{G_i,G_j,G_k} \\
&= \log\left(\frac{p(G_{i,j,k}=1,1,1)p(G_{i,j,k}=1,0,0)p(G_{i,j,k}=0,1,0)p(G_{i,j,k}=0,0,1)}{p(G_{i,j,k}=1,1,0)p(G_{i,j,k}=1,0,1),p(G_{i,j,k}=0,1,1)p(G_{i,j,k}=0,0,0)}\right),
\end{aligned}
$$

(2)

and has the interpretation of whether the expression status of a third gene, $G_k$, changes the 2-point interaction between $G_i$ and $G_j$ expression. We presented previously (Beentjes and Khamseh, 2020) that this definition recovers, in a data-driven manner, known ground truth interactions in statistical physics systems such as the Ising model, and more generally energy-based models, as well as any other Markovian complex system. We further demonstrated that our MFI definition, used to directly estimate the interaction, results in the same estimate as when training a Restricted Boltzmann Machine, both analytically and numerically within statistics. The advantage of the MFI direct estimation on binary data is its model-independent definition interpretability and its avoidance of having to fit the joint probability distribution amongst the variables. The latter is a much more complex quantity to estimate robustly than the combination of expectation values in the MFI estimator. Finally, we note that conditioning on $\underline{G}=0$ in Eq. (1) is equivalent to finding the 'pure' 2-point interaction between $G_i$ and $G_j$ without the influence of the other genes' expression. Note that $\underline{G}$ need not contain the set of all other genes when estimating the interaction. Indeed, it is sufficient to only condition on the Markov blanket (MB) of $G_i$ and $G_j$, i.e., the smallest set of genes $\underline{G}$ conditional on which $G_i$ and $G_j$ are independent of all other genes. Once conditioned on the MB, the information from other genes no longer influences the interaction between $G_i$ and $G_j$. Therefore, restricting $\underline{G}$ to only contain the MB of genes for each pair $G_i$ and $G_j$, improves statistical power, whilst simultaneously ensures that the 2-point interaction remains stable by measuring the direct dependence between $G_i$ and $G_j$, rather than indirect correlations. The same argument holds for higher-order interactions. Figure EV1 presents a comparison between MFIs, correlation, partial correlation and mutual information, computed on data generated from a set of DAGs in accordance to Appendix Fig. S1, first presented in (Jansma, 2023a). The set of MFIs is distinct between distinct DAGs, whereas other dependence metrics are only able to distinguish some, but not all, distinct DAGs.

Currently, performing conditional independence tests amongst all groups of genes to determine their MBs, is statistically and computationally prohibitive. For this reason, we restrict the estimation of $n$-point interactions to the top 1000 HVGs, after quality control. Stator then infers the MBs of the HVGs via a hybrid Bayesian network inference technique (Kuipers et al, 2022) which sequentially performs (conditional) independence testing, starting from a fully connected undirected graph of genes (Peter–Clark algorithm (Spirtes et al, 2001)), followed by a score and search MCMC approach to obtain the optimal completed partially directed acyclic graph (CPDAG), introduced in (Kuipers et al, 2022). We emphasise that we do not claim any causal inference or regulatory relationships amongst these genes based on the inferred network. Instead, we utilise this algorithm to infer a gene signature dependence network structure to obtain the MB and estimate higher-order interactions with sufficient statistical power, with the

final aim of inferring cell (sub)types and states. Inferring this dependence network massively reduces the search space for potentially significant interactions. For run-time considerations, see Supplementary Material A.1.

Finally, we note that MFIs are symmetric. Therefore, when estimating, e.g., a 2-point interaction using line 3 in Eq. (1), one can choose to estimate the terms $\mathbb{E}[G_i|G_j=1, G_i^{MB}]$ or $\mathbb{E}[G_j|G_i=1, G_j^{MB}]$, whichever results in the greatest statistical power, i.e., when either the MB of $G_i$ or $G_j$ is smaller, or more generally, when the MB of $G_i$ or $G_j$ is more populated.

Having identified the set of MBs, Stator then estimates up to 7-point interactions amongst the genes in the expression data. The 2-point interactions are estimated between all pairs of genes, the 3-, 4- and 5-point interactions are estimated amongst all gene tuples that are in each other's MB (the interaction amongst Markov disconnected genes vanishes (Jansma, 2023a)), and 6- and 7-point interactions are calculated amongst genes that are in the MB of a tuple of genes with a significant 5- or 6-point interaction. Every interaction is estimated using the smallest possible MB.

In order to prioritise candidate interactions for the next step ("Deviating gene tuples (d-tuples)"), each interaction is estimated 1000 times by bootstrap resampling the data. An interaction is prioritised as a 'non-zero' candidate for the next step if the fraction $\lambda$ of bootstrap estimated interactions with a different sign from the original estimate is less than 0.05. This procedure is more permissive than testing for the hypothesis that the 95% two-sided percentile bootstrap confidence interval does not contain zero. For the datasets studied in this work, we verify numerically that this procedure is equivalent to demanding $90-95\%$ confidence, depending on the order of the interaction.

## Deviating gene tuples (d-tuples)

In a finite sample of $N$ cells, the observed frequency $\Phi_s$ of a tuple $s = \{s_1, \ldots, s_n\}$ of $n$ independently expressed binarised genes is binomially distributed as:

$$P(\Phi_s = k) = \binom{N}{k}\pi_s^k(1-\pi_s)^{N-k}, \text{where } \pi_s = \prod_{i=1}^{n}(s_i\mu_i + (1-s_i)(1-\mu_i)),$$
(3)

and $\mu_i$ is the mean expression of gene $i$ across all cells under consideration (i.e., the cells for which the relevant MB is zero). Equation (3) describes the null hypothesis that the observed cell counts are the result of independently expressed genes, and gives the expected number of cells under this null: $\mathbb{E}[\Phi_s] = \pi_s N$. An observation $\Phi_s = \phi_s$ of one of the $2^n$ joint states of $n$ genes can be assigned a $p$ value:

$$p = 1 - \sum_{k=0}^{\phi_s-1} P(\Phi_s = k),$$
(4)

and log twofold change, or deviation:

$$\text{Log2FC} = \log_2\left(\frac{\phi_s}{\pi_s N}\right) \in (-\infty, \infty).$$
(5)

The $p$ values are calculated for all tuples with a positive Log2FC, and corrected for multiple hypothesis testing with the Benjamini–Yekutieli procedure (Benjamini and Yekutieli, 2001). A non-zero interaction can thus have one or more *deviating tuples* (d-tuples), those tuples of genes that significantly deviate from the null hypothesis. Since a non-zero interaction reflects a higher-order dependency in the data, its d-tuple describes the gene expression patterns that are (at least in part) responsible for this dependency. The set of cells that have the $n$ genes in that particular expression state—ignoring the state of the MB—form the associated set of cells. Note that cells carrying a certain combination of d-tuples need not cluster in expression space: whilst these cells all share a particular gene expression pattern among the $n$ genes, the expression of all other genes can vary greatly. This makes it in principle possible for a cell state to be widely dispersed in expression space.

For further simulations where fictitious d-tuples are induced or removed from real data, see Supplementary Material, Sec. A.3.

## Hierarchical clustering of d-tuples

Given all d-tuples, Stator creates a cell-by-d-tuple matrix, with binary entries 1 or 0, representing whether or not a cell contains a particular gene d-tuple. Stator then hierarchically clusters these d-tuples (rather than cells) based on a notion of distance, here the Sørensen–Dice coefficient, to identify d-tuples that more commonly co-label the same cells. This hierarchy of separation among d-tuples can be visualised in a dendrogram. Note that the Sørensen–Dice coefficient, sometimes referred to as the Dice similarity coefficient, is not a distance metric because it does not satisfy the triangle inequality. More specifically, the Dice dissimilarity between two boolean vectors $X$ and $Y$ is defined as:

$$d(X, Y) = 1 - \frac{2|X \wedge Y|}{|X| + |Y|}.$$
(6)

In order to group the d-tuples together, we cut the dendrogram at a Dice dissimilarity that, by default, is set at the value that maximises the weighted modularity score of the resulting clustering (Newman, 2006), where a pair of d-tuples is assigned an edge weight of one minus their Dice similarity. At the set Dice dissimilarity threshold, cells expressing these gene d-tuples are grouped together forming Stator states. In particular, cells can exist in multiple multiple Stator states depending on different gene signature similarities. Lowering the Dice value threshold increases granularity, the resolution by which states are predicted, which we have shown, in some instances (e.g., Appendix Fig. S12), to better resolve subtypes or sub-states for large and transcriptionally heterogeneous groups of cells.

## Stator pipeline

Stator is a Nextflow pipeline (written using Nextflow version 21.04) that consists of a main Nextflow script (DSL1) managing a number of Python and R scripts and modules (see Appendix Fig. S3 for an overview of the pipeline). Stator aims to balance modularity and ease-of-use with flexibility, so is fully containerised (Docker images hosted on Dockerhub) and allows the user to specify different

preferences and settings in a separate json file, meaning that it should run reproducibly on any Sun Grid Engine compatible cluster. The only file that has to be supplied by the user is a .csv file (called `rawDataPath` in the json settings file) containing the expression data of $G$ genes (columns) and $C$ cells (rows), where the first row contains the column/gene names. Optionally, the user can provide a file `userGenes` that contains the names of genes that should be included in the final analysis regardless of their variability, a file `genesToOne` containing genes whose Markov blanket state should be 1 instead of 0 (not used in this paper) which allows for conditioning on different Markov blanket states, and a file `doubletFile` containing a Boolean exclusion list, for example based on a doublet annotation, that indicates which cells should be excluded, regardless of other QC metrics. The user should further specify the total number of cells (`nCells`) and genes (`nGenes`) to be used in the analysis.

The pipeline then initiates the first process in the pipeline, defined in the makeTrainingData.py script. By default, Stator assumes that the data is already quality controlled (QCed) and only performs very basic data preparation (specified by the setting `datatype='agnostic'`): all cells specified by `doubletFile` are excluded, PCA and UMAP embeddings are calculated, and up to `nGenes` genes are included, starting with those specified in `userGenes`. A total of `nCells` are then randomly selected for downstream analysis. Alternatively, Stator can run in `datatype='expression'` mode and perform basic scRNA-seq QC, where parameters such as the threshold of mitchondrial gene reads can be set by the user. In `expression` mode, Stator first includes the `userGenes`, but then adds the most highly variable genes until `nGenes` are included. The final count matrix of size `nCells × nGenes` is then binarised and sent to the next process.

Stator then aims to estimate the graph of conditional dependencies among the genes. It does this by generating a first guess using a parallelised implementation of the Peter–Clark (PC) algorithm (parallelPCscript.R, based on (Le et al, 2016)). The PC-algorithm starts with the fully connected graph of dependencies, and then iteratively performs dependency tests among connected pairs, removing an edge when no evidence for dependence is found (delaying removal until all tests are done to ensure order-independence. In addition, we use the majority rule suggested by (Colombo et al, 2014)). Somewhat counter-intuitively, a larger significance threshold for the dependency tests corresponds to a more conservative estimate, since preserving more edges will result in larger MBs which are necessarily more conservative. The default threshold is set at $p<0.05$, not corrected for multiple hypothesis testing, but can be adjusted by the user. Reducing this threshold makes the estimate less conservative, but can significantly speed up the estimation procedure by eliminating more edges and reducing the size of the estimated MBs. This initial guess is then iteratively improved upon using the score-based MCMC method outlined in (Kuipers et al, 2022) (iterMCMCscript.R). This method is based on an efficient exploration and scoring of the space of possible DAGs, and allows new edges to be introduced into the initial guess if they significantly increase the score. The CPDAG equivalence class corresponding to the graph found by parallelPC-script.R is used as the starting point, and the script iterates until increasing the search space no longer increases the score. To be as conservative as possible in our estimates, the final MBs used in all downstream analyses are those based on the full final search space

on which this algorithm terminated (not, for example, only the *maximum-a-posteriori* estimate or its associated CPDAG).

Using these MBs, all 2-to-5-point interactions are calculated among genes that are mutually Markov connected (calcHOIs-WithinMB.py). By default, uncertainty is quantified by bootstrap resampling, but this can be done more efficiently using an estimate for the asymptotic error rate of the MFIs by setting `asymp-Bool=1` in the settings; agreement with bootstrapped confidence intervals was confirmed previously (Jansma, 2023b).

The higher-order interactions are analysed (createHOIsum-maries.py) and used to calculate the significant d-tuples and final Stator states (identifyStates.py). In addition, if there are interacting 5-tuples that are Markov connected to additional genes, a targeted search for 6- and 7-point interactions is performed. Run time using reasonable settings is discussed in Supplementary Material A.1.

Stator's output includes files containing both the binarised and unbinarised QCed expression data, a list of all d-tuples, and cell embedding coordinates. These can then be used for further downstream analysis, for which we provide an R Shiny app. More information on the various settings available to the user, as well as a complete list of output files, is available at https://github.com/AJnsm/NF_TL_pipeline/tree/main.

## Stator's R Shiny App

The Stator App was implemented as a web application for downstream analyses, following the general code structure of previously developed shiny apps (Danger et al, 2021; Ge et al, 2020). It used the R Shiny package (v1.7.4) from R studio (shiny.rstudio.com). As an open-source application, the code is available through GitHub at github.com/YuelinYao/MFIs. The Docker container image is available on Dockerhub: hub.docker.com/r/yuelinyao120/stator-app. The Stator App is hosted at shiny.igc.ed.ac.uk/MFIs/. A complete list of packages used can be found at github.com/YuelinYao/MFIs/blob/main/renv.lock. The app consists of 13 main panels (*About, Table, Heatmap-Cells, Heatmap-Genes, GO & KEGG, Using rrvgo, Upset Plot, DE analysis, Find Markers, Automatic Annotation, Markov Blanket, 2D Plot, Dendrogram*).

### Data upload and file input

The app begins with an About page, providing general information about the app and a tutorial on its use. It includes a liver cancer dataset (Barkley et al, 2022) already uploaded, and users can upload their own files in '.csv' format (size <100GB running from server); most of these are output files from the Stator Nextflow pipeline. The Tutorial explains how to prepare a dataset, and provides information on the app's parameters and statistical tools.

### Summary table

The app generates a statistics summary table by filtering and clustering significantly deviating tuples (d-tuples) after file upload and parameter setting. The minimum enrichment factors in Log2 transformation have a default value of 3, i.e., eightfold change, the minimum number of cells labelled by each d-tuple has a default value of 0, and the FDR is by default set to 0.05. These parameters are used to filter d-tuples. The Dice dissimilarity is employed for hierarchical clustering of d-tuples. This table presents tuple genes

and their state, along with their respective enrichment factor in log2 transformation, adjusted enrichment $p$ value, the number of cells labelled by each d-tuple, and the cluster that includes this d-tuple. The d-tuples in each cluster define a Stator cell state.

### Cell states with external annotations

The Shiny App offers the ability to explore cells and genes in each state using externally provided annotation through Heatmap-Cells and Heatmap-Genes panels, respectively. Users can additionally specify the type of analysis they wish to perform, such as annotation term enrichment analysis (over-representation test), depletion analysis (under-representation test), or a two-sided Fisher's exact test.

(a) Enrichment analysis for cells: Enrichment analysis allows users to test for the enrichment of external annotation terms in Stator states. We use the following notation:

$N$: Total number of cells.

$m$: Number of cells (of total $N$) in a given Stator cell state.

$k$: Number of cells (of total $N$) in the external annotation.

$q$: Number of cells shared between a given Stator cell state and an external annotation.

The corresponding random variable is denoted by $X$. The null hypothesis is that the observed overlap between the identified cell state and the external annotation is no greater than is expected by chance. The $p$-value is calculated as the probability of observing more overlapping cells than expected under this null hypothesis:

$$P(X \geq q) = 1 - P(X \leq q-1) = 1 - \sum_{i=0}^{q-1} \frac{\binom{k}{i}\binom{N-k}{m-i}}{\binom{N}{m}}.$$

(7)

The probability is computed with the R function:

`phyper(q-1, m, N-m, k, lower.tail = FALSE, log.p = FALSE)`

Once the $p$-value is computed for all pairs, we use the Benjamini and Hochberg (BH) method (Benjamini and Hochberg, 1995) for correcting for false positives arising from multiple tests. The corrected $p$-values are transformed by taking the negative logarithm (base 10) before then being visualised as a heatmap, using ComplexHeatmap (v2.14.0) (Gu, 2022; Gu et al, 2016).

(b) Depletion analysis for cells: the null hypothesis is that the observed overlap between the identified cell state and the external annotation is no fewer than would be expected by chance. The $p$-value is computed with the R function:

`phyper(q, m, N-m, k, lower.tail = TRUE, log.p = FALSE) .`

(c) The two-sided Fisher's exact test: As an option, a two-sided Fisher's test may be performed with the following R function and the heatmap provided from this test is coloured by the log10 transformed odds ratio:

`fisher.test(matrix(c(q, m-q, k-q, N-m-k+q), 2, 2), alternative='two.sided').`

A similar statistical test is performed, using the Heatmap-Genes function, to test for the overlap between an externally supplied gene list with genes listed among Stator state d-tuples.

### Gene ontology and KEGG pathway enrichment analysis

The Shiny App allows users to perform Gene Ontology (GO) and KEGG Pathway Enrichment analysis for d-tuple genes in each Stator state using the R package clusterProfiler (v4.6.2) (Yu et al, 2012). Users can specify the cell state(s) of interest and reference genome for the dataset (e.g., hsapiens_gene_ensembl, org.Hs.eg.db, and hsa for human), or background genes. Significantly enriched terms (FDR < 0.05) are displayed in the app. The app also implements Rrvgo (Sayols, 2023) to reduce the redundancy of GO terms and for their visualisation as word cloud, treemap or scatter plots.

### Cell Upset plot

The app shows Upset plots, with rows corresponding to numbers of cells labelled by each state, and columns providing the number of cells labelled in common. This uses the ComplexHeatmap package (v2.14.0) (Gu et al, 2016) in R.

### Differential expression analysis

The Shiny App allows users to perform differential analysis: (i) between two cell states, disregarding all cells co-labelled with both states (termed s2s) from the DE analysis tab, or (ii) between cells labelled with a state and all cells without this label (termed s2o) from the Find Markers tab. Differential gene expression analysis was implemented using the Find Markers function from Seurat (v4.3.0) (Stuart et al, 2019). Users can define log2-fold change and adjusted $p$ value thresholds. The app then displays an expression heatmap of differentially expressed genes, a volcano plot, a summary statistics table for differential expression, and Gene Ontology and KEGG term enrichment significance results for differentially expressed genes.

In the automatic annotation tab, users can provided a table of genes of interest, and the app will identify s2o-DEGs for all Stator states, and automatically return the DEGs in the provided gene list for easy anotatation.

### Markov Blanket, MB

The app provides functionality for users to extract and visualise the MB for a particular gene. For this visualisation, it imports the inferred MCMC graph and extracts the MB covering all parents, children and spouses of this gene. This was implemented by the R package, igraph (v1.4.1) (Csardi et al, 2006).

### UMAP plot

The app allows users to visualise a cell state of interest within an uploaded set of UMAP cell coordinates. This was implemented using the DimPlot function from Seurat (v4.3.0) (Stuart et al, 2019).

## Assigning labels to Stator States

In this section, we provide a three-step guide for annotating Stator states.

Step 1. Provisionally label Stator state $i$ by cell type: Compare all genes that are significantly more highly expressed in $i$ over all other stages ($j \neq i$; i.e. s2o-DEGs) against sets of known cell-type gene markers. For example, higher expression of *Tubb3* in cells in state $i$, over cells not in state $i$, provisionally labels the cell type of $i$ as "Neuron" ((Ferreira and Caceres, 1992); Dataset EV2). Cell-type marker resources include: (i) the cell-by-gene resource (Abdulla

et al, 2023; Megill et al, 2021), (ii) PanglaoDB (Franzén et al, 2019b), (iii) MSigDB (Castanza et al, 2023), (iv) the Human Protein Atlas (Karlsson et al, 2021) and (v) CellMarker 2.0 (Hu et al, 2022). These resources are not comprehensive, for example, because they have not captured developmental stage-specific cell-type markers such as those listed in (Yuzwa et al, 2017) for mouse E17.5 radial glial cells. Stator state $i$ is provisionally labelled as cell type $T_i$ when $m$ s2o-DEGs are markers for $T_i$. We recommend using $m \geq 3$. The label has greater confidence when these $m$ are a larger fraction of all $i$'s s2o-DEGs. Note that a gene can be a marker for diverse cell types, for example, *Tnc* for basal respiratory cells, astrocytes and smooth muscle cells (Human Protein Atlas (Karlsson et al, 2021)). Also, be aware that gene markers derived from differential gene expression between clusters in whole transcriptome space may conflate cell (sub)types and/or cell states.

Step 2. Provisionally label Stator state $i$ by cell state: State $i$ may lack cell-type markers, but its s2o-DEGs may contain $m \geq 3$ markers for cell state, for example, cell cycle phase (G1/S or G2/M, Supplementary Table S5 in Tirosh et al, 2016) or immediate early response (Wu et al, 2017) or cellular process (e.g., metaphase/anaphase transition of cell cycle gene ontology term (Ashburner et al, 2000)), which allows $i$ to be labelled as a cell state. Note that Stator's data-driven approach can result in state $i$ being labelled by both cell type and cell state, for example a radial glia-like cell in G2/M cell cycle phases.

Step 3. Resolve provisional labels by cell (sub)type and/or state: Compare the differentially expressed genes between state pairs ($i, j$; i.e., s2s-DEGs) with known cell (sub)type and state markers (as in Steps 1 and 2). Stator state $i$ is labelled as cell (sub)type and/or state $T_i$ that distinguishes it from state $j$ when its $m$ ($\geq 3$) s2s-DEGs are markers for $T_i$. For example, states $i$ and $j$ may have been provisionally both assigned as embryonic radial glial cell-like cells due to many of their s2o-DEGs being markers for this cell type (as tabulated in Yuzwa et al, 2017), before $i$ is then differentiated from $j$ by its upregulated s2s-DEGs including neuronal marker genes (e.g., *Ascl1*, *Neurog2* and *Gadd45g*) that predict state $i$'s neuronal fate in the forebrain (Main Text). If state $i$ can be labelled in a mutually inconsistent manner (e.g., newborn neuron and neural stem cell; Appendix Fig. S12) then lower the Dice dissimilarity threshold, as this may reveal a deep branch in the dendrogram (Fig. 1E) that separates these cell types.

## Datasets

To showcase Stator's prediction of cell types, subtypes and/or states in diverse normal and disease samples, we chose three diverse datasets: (i) Normal brain tissue E18 mice from the 10XGenomics '1.3 Million Brain Cells from E18 Mice' dataset (10XGenomics, 2017), downloaded from https://www.10xgenomics.com/resources/datasets, (ii) human liver tissue from control and disease (cirrhosis) samples (Ramachandran et al, 2019), and (iii) human liver cancer (hepatocellular carcinoma) tissue from (Barkley et al, 2022). Dataset (i) contains an unannotated Louvain clustering (60 clusters in total) (Blondel et al, 2008) of 1,306,127 cell transcriptomes distributed over 133 libraries, sequenced on an Illumina HiSeq 4000 using paired-end sequencing at a moderate read depth of 18,500 reads per cell, keeping only uniquely mapped reads. To annotate these clusters by cell type, we identified upregulated (with respect to all other cells) marker genes using the R-function

`scran::findMarkers` (Lun et al, 2016). Cluster 7 had top 10 marker genes (all at FDR<$10^{-10}$) {**Syt6**, Gm27032, Slain1,**Pbx3**, Rgs8, Fgf3, Nkx2-3, Otor,**Six3,Myh7**}. Gene symbols shown in bold are listed on `mousebrain.org/adolescent/genes.html` (Zeisel et al, 2018) as markers for CNS-neurons, while the other genes are not markers for any cell type (except for *Rgs8* which marks trilaminar cells). Furthermore, when inferring markers against specific other clusters, *Dlx2*, *Dlx5* and *Dlx6os1* appeared as top markers; these genes control GABAergic neuron differentiation in developing mice (Petryniak et al, 2007). Cluster 10 had top 10 marker genes (all at FDR<$10^{-6}$) {**Gm11627**, Abhd4, Mpv17, Cldn10, Dhrs1, Thbs3, **Aldoc**, **Prdx6**, Gm20515, Chil1}; gene names in bold show upregulated expression in radial glial cell precursors at E17.5 (Yuzwa et al, 2017). Although radial glial cell precursors and astrocytes are challenging to distinguish by differential gene expression (Dulken et al, 2017), mature astrocytes are not abundant at this early developmental stage (E18) (Akdemir et al, 2020). We therefore concluded that clusters 7 and 10 are composed of neurons and radial glial cell precursors (RPs), respectively. We analysed two disjoint subsets of RPs, each with $N = 11,950$ cells, and two disjoint subsets of neurons, each with $N = 19,000$ cells. To create a merged dataset containing both neurons and RPs, we first merged both clusters, and then downsampled these to $19,000$ cells, of which 13,905 were neurons, and 5395 were RPs.

Dataset (ii) was generated downsampled from 58,358 to $20,000$ cells, specifically by sampling $10,000$ cells from uninjured samples and $10,000$ cells from cirrhotic samples. When Stator states were compared with expert annotations, lineage annotations from the original publication were used (Ramachandran et al, 2019). No cells annotated as "cycling" by (Ramachandran et al, 2019) remained after sub-sampling. Stator states for dataset (ii) used a Dice dissimilarity of 0.97, with a minimum eightfold enrichment of tuples over expected, a maximum FDR corrected enrichment significance of 0.05, and a minimum of 10 cells labelled by each d-tuple.

Dataset (iii) was generated from a pan-cancer dataset by selecting the liver tumour type, resulting in $14,698$ cells (Barkley et al, 2022). Three types of annotations were defined in the original study: (a) cell type by clustering (Stuart et al, 2019) and SingleR (Aran et al, 2019), (b) cell state by nsNMF (Gaujoux and Seoighe, 2010; Puram et al, 2017), and c) malignant or not by inferCNV (Patel et al, 2014). We defined a gene as being normally expressed in untransformed hepatocytes when it was expressed ($\geq 1$ read) in >0.1% of hepatocytes (Andrews et al, 2022). Stator states for dataset (iii) used a Dice dissimilarity of 0.85, with a minimum eightfold enrichment of tuples over expected, a maximum FDR corrected enrichment significance of 0.05, and a default minimum of 0 cells labelled by each d-tuple (for this dataset the number of cells in each d-tuple is $\geq 13$).

## Quality control (QC) and expression binarisation

Data used as input to Stator was pre-processed using standard Quality Control (QC) best practice (Luecken and Theis, 2019). When doublet removal was not performed in a study, or this information was absent, we removed doublets using Scrublet (Wolock et al, 2019). We restricted the analysis to the 1000 most highly variable genes (HVG), quantified using Scanpy (Wolf et al, 2018), followed by binarisation of gene expression. Droplet-based

protocols commonly result in sparse data with many dropouts. Justification for gene expression binarisation has been previously demonstrated for a variety of scRNA-seq analyses including dimensionality reduction, clustering, differential gene expression and pseudotime analyses (Bouland et al, 2021, 2023; Qiu, 2020). Following the literature, we binarise expression values, with genes without expression evidence as zeros, and those with evidence of expression as ones.

## Comparison with clustering

We applied hierarchical clustering on the two mouse brain datasets to compare Stator with the conventional clustering approach. We processed and selected the 2000 most HVG to compute principal components (Stuart et al, 2019), and then the top 20 PCs were used to calculate the Euclidean distances. Specifically, we used Ward's method for hierarchical clustering, which is based on minimising the loss of information from joining two groups (Murtagh and Legendre, 2014).

Using the same data to both cluster cells and test the differential expression will result in an extremely inflated type I error rate (Gao et al, 2022). To compare Stator with robust clustering results, we applied a selective inference approach to test for a pairwise significant difference between two clusters (Gao et al, 2022). This approach protects against selective inference by correcting for the hypothesis selection procedure (Gao et al, 2022). We applied Bonferroni method to correct the *p* values for multiple comparisons. Ideal clustering should result in a significant *p* value for any pair of clusters. To declare the final number of distinct clusters, we take the largest number of clusters such that all clusters are pairwise significantly distinct as the total number of clusters is varied.

## NMF procedure and gene modules

Following methods used in (Barkley et al, 2022), we applied NMF to the mouse embryonic RP dataset (data as in Fig. 2) and the liver disease dataset (data as in Fig. 5). For the liver cancer dataset, we re-use the cell annotation provided, following application of NMF by (Barkley et al, 2022). The input to NMF is the normalised centred expression data of the 2000 HVGs, with all negative values set to zero. Specifically, we applied 'nsNMF' within a reasonable initial range (10–30) for the number of components to be identified, using the R package NMF (Gaujoux and Seoighe, 2010). The output of NMF is a (gene-by-component) weight matrix, whose entries represent the contribution of a gene to that component, and a (component-by-cell) coefficient matrix, whose entries represent cell usage, defined as how much each set of gene modules is 'used' by each cell in the dataset (Kotliar et al, 2019). To construct non-overlapping gene modules, we ranked genes using the algorithm described in (Barkley et al, 2022) via two lists: list 1 ranks the genes' contribution to each component, and list 2 ranks the components to which each gene contributes. For each component, genes were added in the order of their rank (list 1), until a gene was reached that contributed more to a second component (list 2). Components with fewer than 5 genes were removed, and the procedure repeated. We obtained the gene modules for each number of initial components (10–30). The number of gene modules thus never exceeds the number of components. The largest initial number of components was selected

for downstream analysis, for which the number of gene modules equals the number of components. For mouse RPs, we obtained 27 gene modules; for the liver disease dataset, we obtained 25 gene modules.

Once gene modules were predicted, we then scored each cell based on the expression of these gene modules' genes as before (Barkley et al, 2022): for each module, we generated 1000 random gene lists of the same number whose genes have similar expression levels (defined by the MakeRand function in seurat_functions_public.R of (Barkley et al, 2022)). Then, for each cell, the average centred expression of these random gene lists and the NMF gene module were calculated. We computed a *p* value as the proportion of random gene lists that have a higher value of this expression than the corresponding value for the given gene module. The score was then calculated as $-\log10(p\text{-value})$ and rescaled linearly to [0,1]. We only considered a gene module to be expressed in a cell if the corresponding score exceeded 0.5. Finally, a cell was assigned to the highest-scoring module. We then performed a hypergeometric test, controlling for FDR, for the overlap between cell types or states inferred by Stator and NMF.

For the purpose of comparing Stator with NMF, we followed the approach in (Barkley et al, 2022), in the step-by-step manner described above. In this approach, each gene is restricted to belong to one NMF programme, and subsequently each cell to a single NMF state. Alternative NMF methods exist for selecting gene modules and for assigning cells to different (multiple) modules, although there is no definitive approach for gene selection and cell assignment (Gavish et al, 2023; Kim and Tidor, 2003; Kinker et al, 2020; Kotliar et al, 2019; Wang and Zhao, 2022).

## Comparison with other methods

In this section, we show the results of comparing the mouse embryonic RP Stator states with those obtained by NMF (as described above), LDVAE (with default settings in scVI function scvi.model.LinearSCVI) and LDA (with default settings in scVI function scvi.model.AmortizedLDA). Methodologically, we note that these three methods involve subjective choices of (hyper)-parameters and lack uncertainty quantification; also, they do not perform hypothesis testing against an appropriate null distribution and thus, they do not support valid multiple hypothesis correction on the predicted gene modules. Accurate interpretation of these gene modules becomes subjective when there are spurious gene modules that are due to "noise", which would have been "statistically zero" had appropriate uncertainty quantification been applied. For NMF, for example, these effects could have led to several gene modules for which we could not assign a biological label based on the previously used criteria (marker genes reference set, described in Fig. 7). Many states are indeed replicated by these methods. However, in general, there is greater specificity for Stator over NMF, LDVAE and LDA, i.e. there is higher relative expression of gene markers that were previously defined for a cell state as defined by Stator relative to an equivalently-labelled state defined by NMF, LDVAE or LDA (Table 1).

We compared Stator's cell cycle predictions with Tricycle predictions on the RPs dataset, Fig. 8. Tricycle makes use of pre-defined gene markers for G1/S, S, and G2/M phases from (Schwabe et al, 2020) of (Zheng et al, 2022b). By contrast, state definition by Stator is fully data-driven and so it need not classify by these phases, or use these (or any) pre-specified gene markers. As a

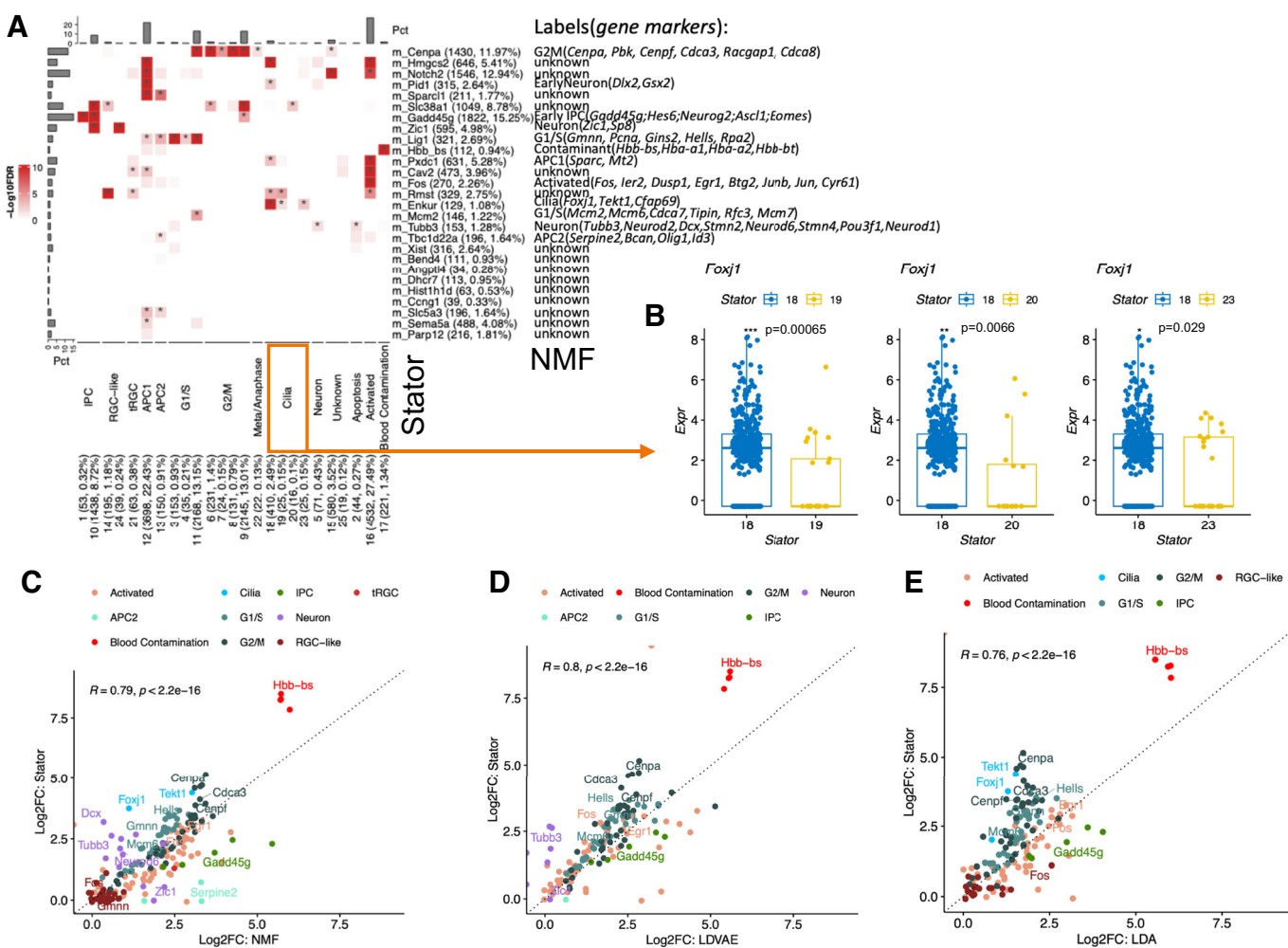

**Figure 7. Comparison with other methods.**

(A) Heatmap presenting the -log10FDR from the hypergeometric test comparing the co-labelling of cells by Stator states and NMF states. To biologically annotate NMF states, marker genes were taken from 4 sources of literature: Dataset EV2; IEGs ((Wu et al, 2017), as previously); cell cycle ((Tirosh et al, 2016), as previously); RGC-like marker genes at E17.5 ((Yuzwa et al, 2017) as previously). The biological labelling of NMF states has been performed exactly as before for Stator, although now more generously using 2 (rather than 3) or more marker genes for non-Stator methods. NMF recapitulates many of the Stator states. (B) For ciliated cells, Stator identified four ciliated cell subtypes (#18 ($n = 410$), #19 ($n = 25$), #20 ($n = 16$), #23 ($n = 25$)), among which one (#18) shows higher expression of *Foxj1*, an ependymal cell marker gene. These four subtypes were not distinguished by NMF. In the boxplot, the median (middle quartile) marks the mid-point of the data and is shown by the line that divides the box into two parts, and the box itself indicates the range in which the middle 50% of all values lie. (C–E) Scatter plot of genes' differential expression for NMF, LDVAE and LDA vs Stator states. Y axis: genes' log2-fold expression change between cells in a Stator state and all other cells; X axis: genes' log2-fold expression change between cells in a state or type identified by NMF/LDVAE/LDA and all other cells. Gene symbol colour reflects cell state or type, as annotated using literature marker genes (as above). Genes' differential expression is highly significantly correlated between Stator and (C) NMF, (D) LDVAE or (E) LDA. However, Stator can detect cell states or types with mostly higher differential expression of known marker genes. In general, there is higher specificity for Stator in most categories over NMF, LDVAE and LDA (Table 1).

result, it maps cells to a continuum of cell cycle sub-phases, defined and ordered by the expression (Fig. 2B) and non-expression (Fig. 3B) of ab initio-discovered genes that are known markers for cell cycle phases.

## Stator state projection to disjoint data

In this section, we discuss how Stator states can be projected from one dataset, on which Stator has been run, to a new dataset without the need to re-run Stator. This offers a wide range of applications such as in the following scenarios: (i) having biologically annotated

Stator states in a given dataset, the biology of a new dataset in a similar biological condition can be inferred without expending further computational resources or effort for repeated annotation, and (ii) allowing for the tracking of equivalent states across different conditions, e.g., time-course disease progression data or cross-species analyses. Here, we introduce Stator state projection from one dataset to another and present a feasibility study to demonstrate the reproducibility of Stator states across disjoint cells in the same condition.

Stator states obtained from dataset 1 can be projected into dataset 2 in the following way: Given a state $A$, list its constituent

d-tuples. If a cell in dataset 2 contains one or more of these d-tuples, it is considered to be in state *A*. By repeating this process for all Stator states, we project the states obtained from dataset 1 to dataset 2.

This projection technique can be used to show the reproducibility of Stator states in two disjoint sets of cells in the same biological condition. This is done by running Stator on both sets separately, and comparing the original Stator states from one dataset to the projected states of the other via an enrichment analysis. We have performed this reproducibility analysis for RPs (see Fig. EV2C,D) and neurons (see Fig. EV5C,D) separately. More specifically for RPs, Fig. EV2C contains cells from Fig. EV2A only. The *X* axis shows the original states obtained by running Stator on this dataset, while the *Y* axis shows states obtained by running Stator on the dataset from Fig. 2. The *Y* axis states are projected onto cells from Fig. EV2A, and their enrichment in original states is computed using a hypergeometric test followed by the Benjamini–Hochberg procedure to control FDR at 5%. In Fig. EV2D, the same procedure is performed in the opposite way:

The cells are from Fig. 2, so now the *Y* axis shows the original states, while the *X* axis shows states obtained by running Stator on the dataset from Fig. EV2A. The *X* axis states are projected onto cells from Fig. 2, and their enrichment in original states is computed using a hypergeometric test followed by the Benjamini–Hochberg procedure to control FDR at 5%. The same procedure is repeated for neurons in Fig. EV5C,D. These panels demonstrate the reproducibility of Stator states on disjoint sets of cells in the same biological condition.

## Data availability

Stator's code and nextflow pipeline, as well as documentation on installation and a vignette, are available on GitHub at https://github.com/AJnsm/NF_TL_pipeline/tree/main. The code for Stator's R Shiny App is available on GitHub at https://github.com/YuelinYao/MFIs, with a Docker container image available on Dockerhub https://hub.docker.com/r/yuelinyao120/stator-app. In addition, the Stator App is hosted at https://shiny.igc.ed.ac.uk/MFIs/. All datasets analysed in this manuscript are publicly available and can be found using the following URLs. scRNA-seq, 10X 1.3 million Brain Cells from E18 Mice, accessible via the 10X website https://www.10xgenomics.com/datasets/1-3-million-brain-cells-from-e-18-mice-2-standard-1-3-0. scRNA-seq and scATAC-seq, Fresh Embryonic E18 Mouse Brain (5k): Single Cell Multiome ATAC + Gene Expression Dataset by Cell Ranger ARC 2.0.0, accessible via the 10X website https://www.10xgenomics.com/datasets/fresh-embryonic-e-18-mouse-brain-5-k-1-standard-2-0-0. scRNA-seq, 10X cirrhotic and healthy human liver, with raw sequencing data available at Gene Expression Omnibus (GEO) under accession GSE136103. scRNA-seq, 10X liver cancer (HCC) dataset, available at GEO under accession GSE203612 and https://github.com/yanailab/PanCancer.

The source data of this paper are collected in the following database record: biostudies:S-SCDT-10_1038-S44320-024-00074-1.

**Table 1. The percentage of genes for each cell state label that show greater expression fold change in Stator over NMF, LDVAE and LDA, respectively.**

|  | Stator vs NMF | Stator vs LDVAE | Stator vs LDA |
|---|---|---|---|
| Activated | 27% (18/67) | **66% (44/67)** | 48% (32/67) |
| Contamination | **100% (4/4)** | **100% (4/4)** | **100% (4/4)** |
| Cilia | **67% (2/3)** | 100%[a] | **100% (3/3)** |
| G1/S | **86% (30/35)** | **97% (34/35)** | **97% (34/35)** |
| G2/M | **57% (21/37)** | **78% (29/37)** | **92% (34/37)** |
| Neuron | **75% (9/12)** | **92% (11/12)** | 100%[a] |
| RGC-like | 31% (31/101) | 100%[a] | **62% (63/101)** |
| APC2 | 0%(0/4) | 50% (2/4) | 100%[a] |

[a]Indicates the state is not detected by LDVAE/LDA.
Values more than 50% are in bold.

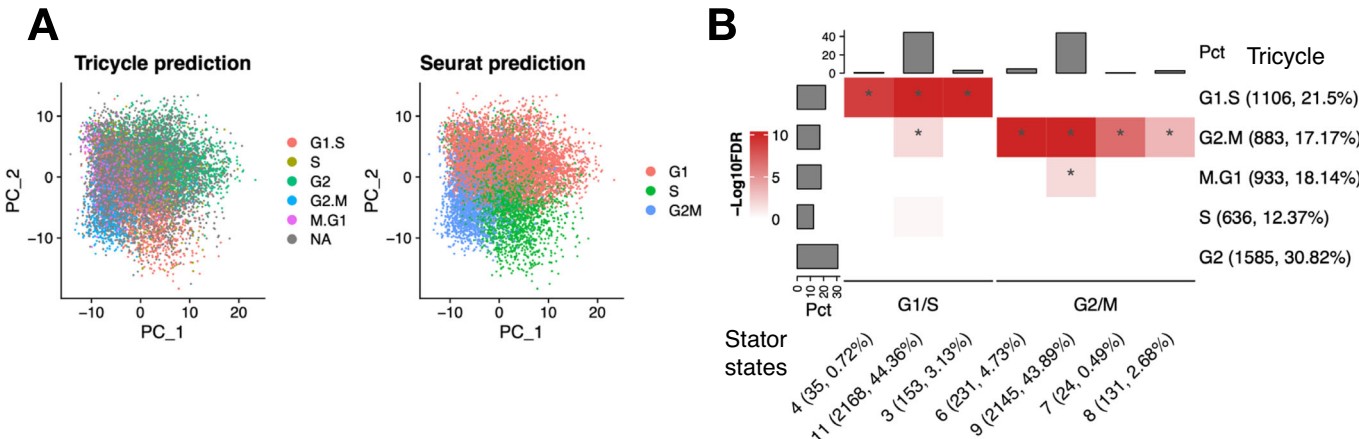

**Figure 8. Comparison of cell cycle states with other methods.**

(A) Cell cycle phases predicted by Tricycle (left) and Seurat (right) for the embryonic RP dataset. Seurat requires all cells to belong to one of the three phases shown and appears to have lower resolution than Tricycle. (B) Stator's 7 states recapitulate Tricycle's 5 cell cycle phase predictions. Heatmap showing the -log10FDR value of the hypergeometric test comparing the overlap of cells from cell cycle relevant Stator states with cells from cell cycle phase identified by Tricycle.

## Peer review information

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

## Acknowledgements

CPP and JCW were funded by the MRC (MC_UU_00007/15). AK was supported by the XDF Programme from the University of Edinburgh and Medical Research Council (MC_UU_00009/2) and is supported by a Langmuir Talent Development Fellowship from the Institute of Genetics and Cancer, and a philanthropic donation from Hugh and Josseline Langmuir. AJ was supported by an MRC Precision Medicine Grant (MR/N013166/1). AJ

thanks Øyvind Almelid for many helpful discussions on Nextflow. YY thanks Xinyi Jiang for improving the Dockerfile of the shiny app, John Ireland and Ewan McDowall for hosting the shinyApp on an IGC server.

## Author contributions

**Abel Jansma**: Data curation; Software; Formal analysis; Validation; Investigation; Visualisation; Methodology; Writing—original draft; Writing—review and editing. **Yuelin Yao**: Data curation; Software; Formal analysis; Validation; Investigation; Visualisation; Methodology; Writing—original draft; Writing—review and editing. **Jareth Wolfe**: Formal analysis; Validation; Investigation; Visualisation; Writing—original draft; Writing—review and editing. **Luigi Del Debbio**: Conceptualisation; Software; Formal analysis; Supervision; Funding acquisition; Validation; Investigation; Methodology; Writing—review and editing. **Sjoerd Beentjes**: Conceptualisation; Software; Formal analysis; Validation; Investigation; Methodology; Writing—original draft; Writing—review and editing. **Chris P Ponting**: Conceptualisation; Resources; Data curation; Software; Formal analysis; Supervision; Funding acquisition; Validation; Investigation; Visualisation; Methodology; Writing—original draft; Project administration; Writing—review and editing. **Ava Khamseh**: Conceptualisation; Resources; Data curation; Software; Formal analysis; Supervision; Funding acquisition; Validation; Investigation; Visualisation; Methodology; Writing—original draft; Project administration; Writing—review and editing.

Source data underlying figure panels in this paper may have individual authorship assigned. Where available, figure panel/source data authorship is listed in the following database record: biostudies:S-SCDT-10_1038-S44320-024-00074-1.

## Disclosure and competing interests statement

The authors declare no competing interests.

# Expanded View Figures

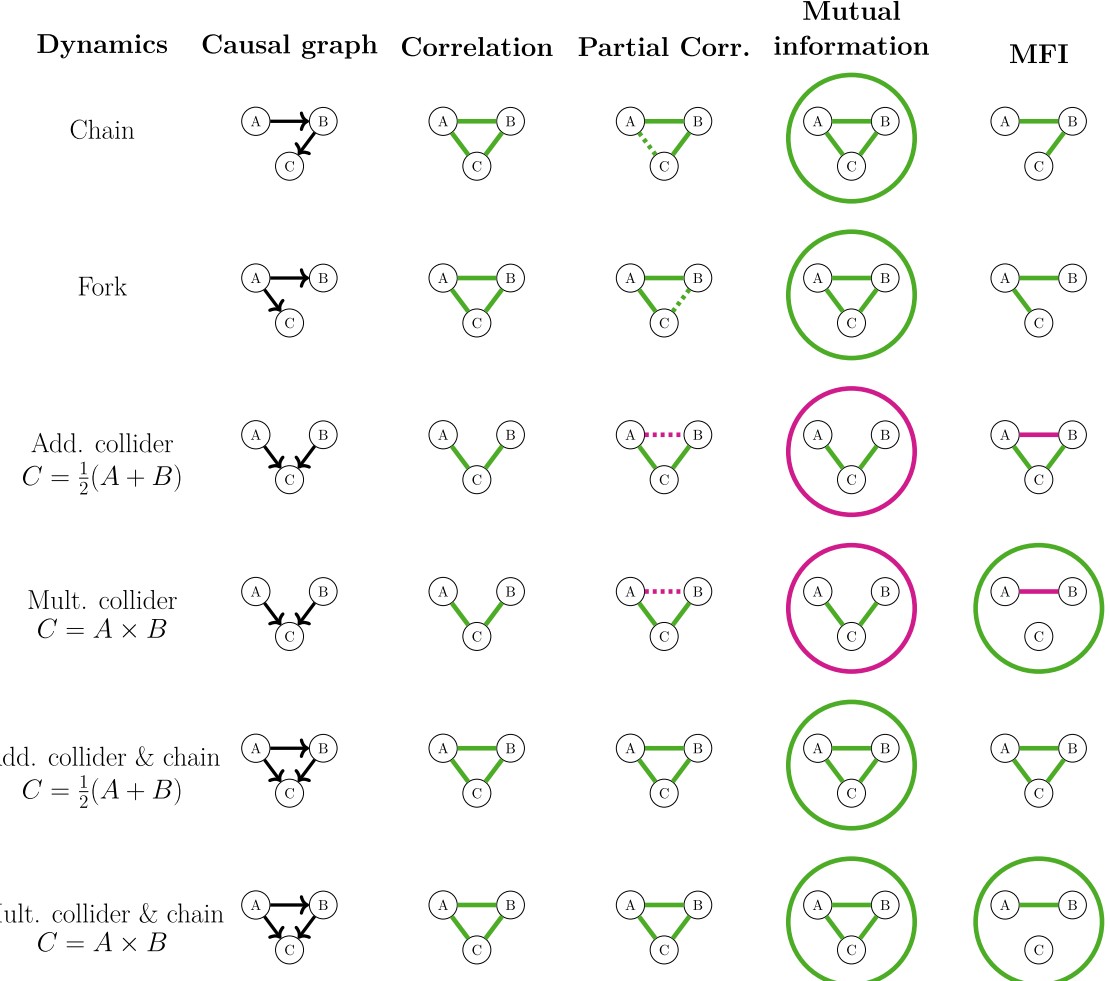

**Figure EV1. Expanded view for Fig. 1.**

Comparison of MFIs with other estimators of dependence. Different causal dynamics lead to different association metrics, and only MFIs can distinguish all 6 scenarios and reveal the combinatorial effect of a multiplicative interaction. Green edges denote positive values, red edges denote negative values, circles denote a 3-point quantity, and dashed lines show edges that show marginal significance that depends on the level of simulated noise. Correlations and mutual information cannot distinguish between most dynamics, and while partial correlation can, for certain noise levels, identify the correct pairwise relationships, it falls short of distinguishing additive from multiplicative dynamics. See Appendix Fig. S1 for the simulation parameters and precise values. Reproduced from (Jansma, 2023a) with permission from the author.

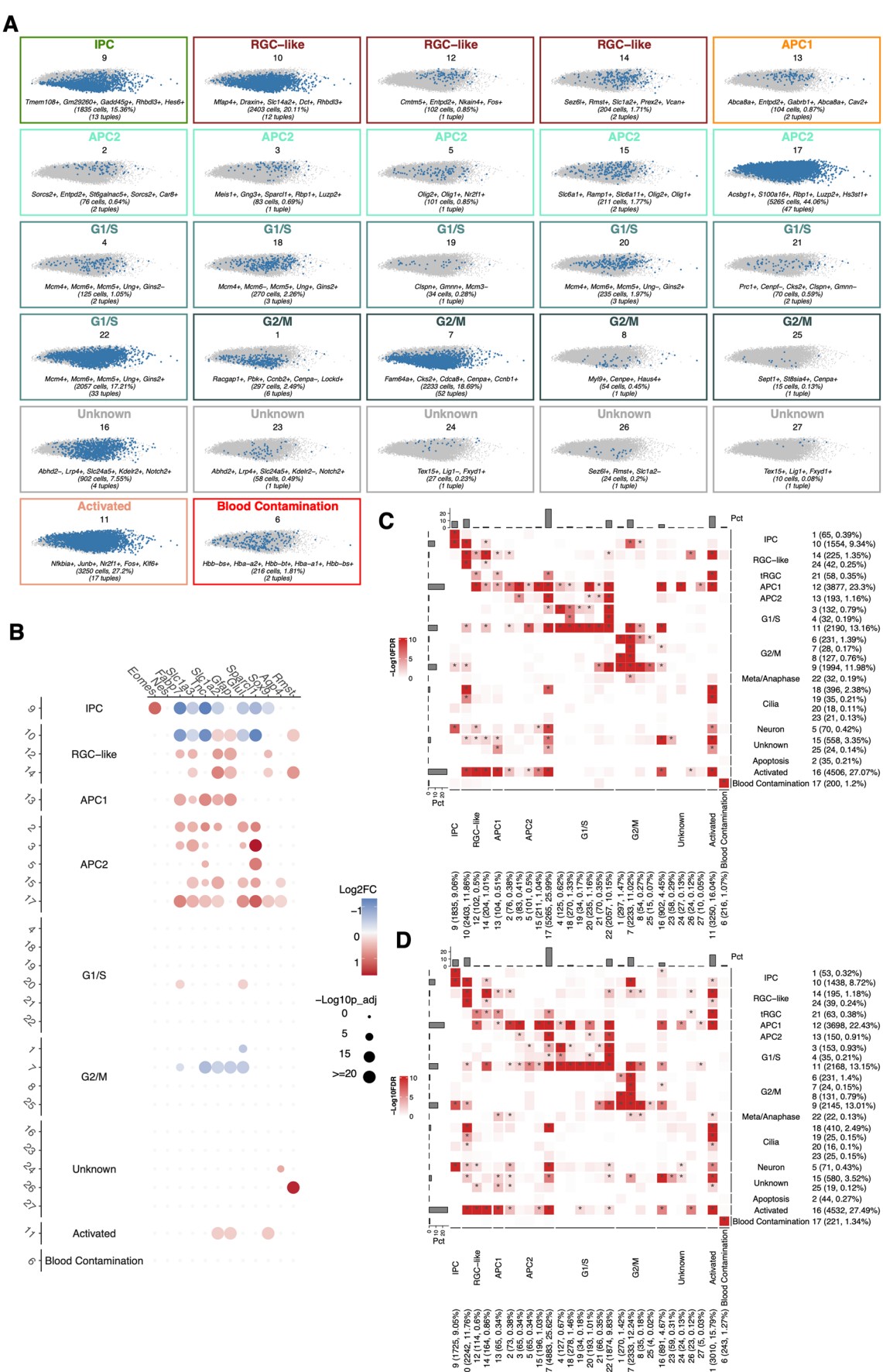

◀  **Figure EV2.   Expanded view for Fig. 2.**

Disjoint RP dataset tested for reproducibility. (**A**) Stator identifies 27 states for a disjoint set of $N = 11,950$ embryonic radial glial precursor-like cells (set RP2) at maximum modularity, annotated on the basis of d-tuple and s2o-DE genes as markers for cell types or cell cycle phases. (**B**) Dot plot illustrating differential expression of astrocytogenesis marker genes across all 27 Stator states. The size of the dots represents the $-\log_{10}$(Seurat p-val-adj) from differential gene expression testing between a state and all other states. Colour intensity reflects the $\log_2$(FC) of gene expression. (**C**) The *x* axis shows the states obtained by running Stator on the RP2 dataset, while the *y* axis shows states obtained by running Stator on the dataset from Fig. 2 (set RP1). The RP1 states, obtained by running Stator on RP1 cells, are projected onto RP2 cells. The enrichment of these projected RP1 states in RP2 Stator states is computed using a hypergeometric test followed by the BH procedure to control the FDR at 5%. (**D**) As (**C**), but for RP1 cells. The *y* axis shows the RP1 states, while the *x* axis shows states obtained by running Stator on RP2 cells. The RP2 states are projected onto RP1 cells, and their enrichment in each RP1 cells' state is computed using a hypergeometric test followed by the BH procedure to control the FDR at 5%. These panels demonstrate the reproducibility of Stator states on two disjoint sets of cells in the same biological condition.

   

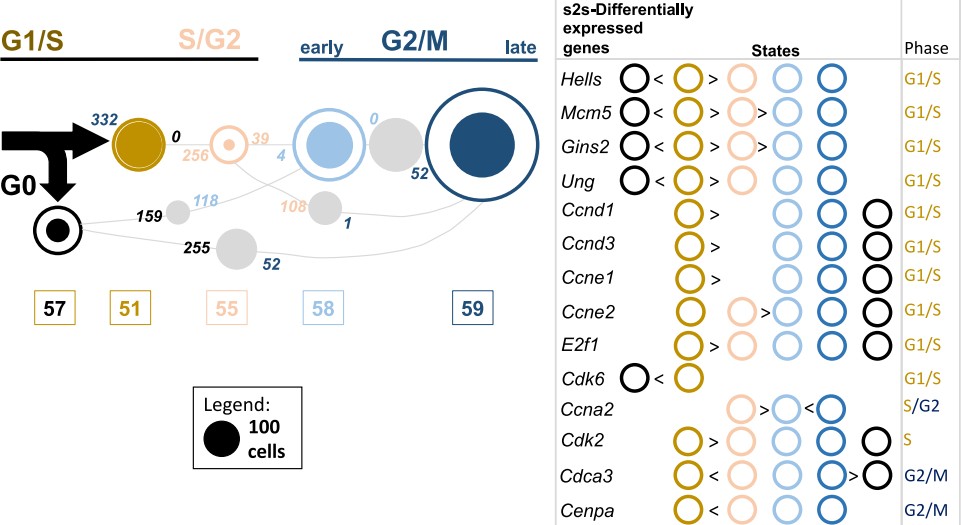

**Figure EV3. Expanded view for Fig. 3.**

s2s-DEG analysis for 5 cell cycle states. Numbers of cells labelled with any one of 5 cell cycle states (#57, 51, 55, 58, and 59) in embryonic RPs and neurons; areas of circles are proportional to their number (see legend). Filled circles indicate numbers of cells labelled with only of these single cell cycle states. Grey circles' areas indicate numbers of cells labelled with two cell cycle states, those indicated by lines. Numbers of significantly differentially expressed genes between cell cycle state pairs (i.e., s2s-DEGs) are provided between the two states being compared; their colours refer to the state showing higher expression. For clarity, state pairs with ≥ 25 cells are shown. DEGs between any two states, including state pairs with fewer than 25 co-labelled cells, are provided in Dataset EV12. Appendix Fig. S8B additionally provides the number of co-labelled cells between any two states. Right: s2s-DEGs are indicated by ">" or "<" symbols; for example, *Hells* mRNA expression is significantly higher in State #51 over States #57, 55, 58 and 59. Early/late G1/S or G2/M cell cycle phase labels (top) were assigned using these mRNAs' cell cycle phases known from high-throughput (top right; (Giotti et al, 2018)) and targeted experiments (*Ung* mRNA in late G1/S (Slupphaug et al, 1991) and *Cenpa* in G2 (Shelby et al, 1997)).

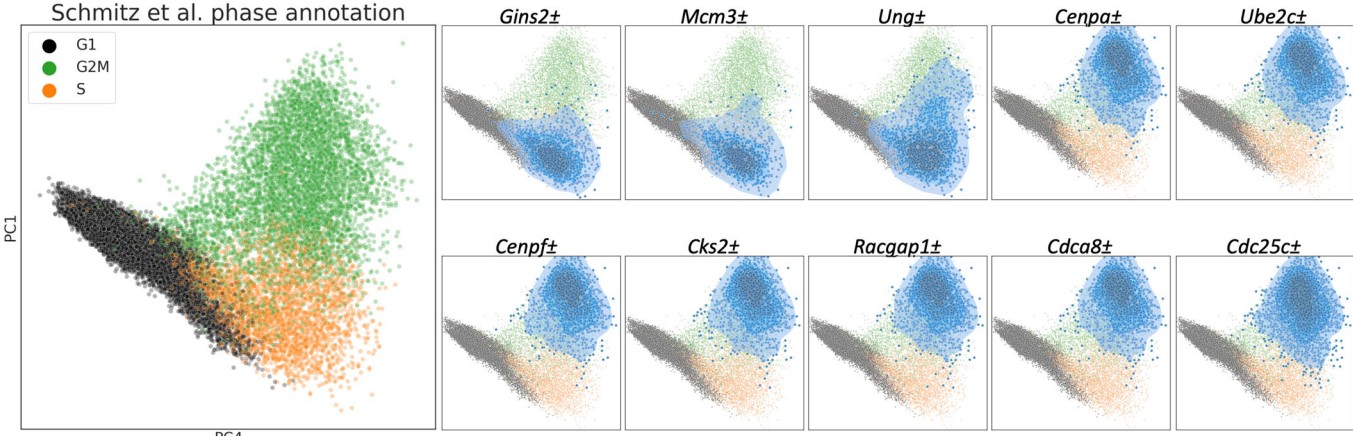

**Figure EV4. Expanded view for Fig. 3.**

*Minus* gene expression is required to specify cell cycle sub-phases. Left: External cell cycle annotations of a merged mouse brain dataset sourced from five different experiments (Schmitz et al, 2022), with the cells from dataset GSE93421 removed. Right: In contrast to Fig. 3B, highlighting cells (in blue) based only on the expression of all but one of the marker genes, leaving the gene indicated above each plot (the 'minus' gene) unrestricted, results in more diffuse cell cycle specificity, and simply marks cells in G1/S or G2/M phases.

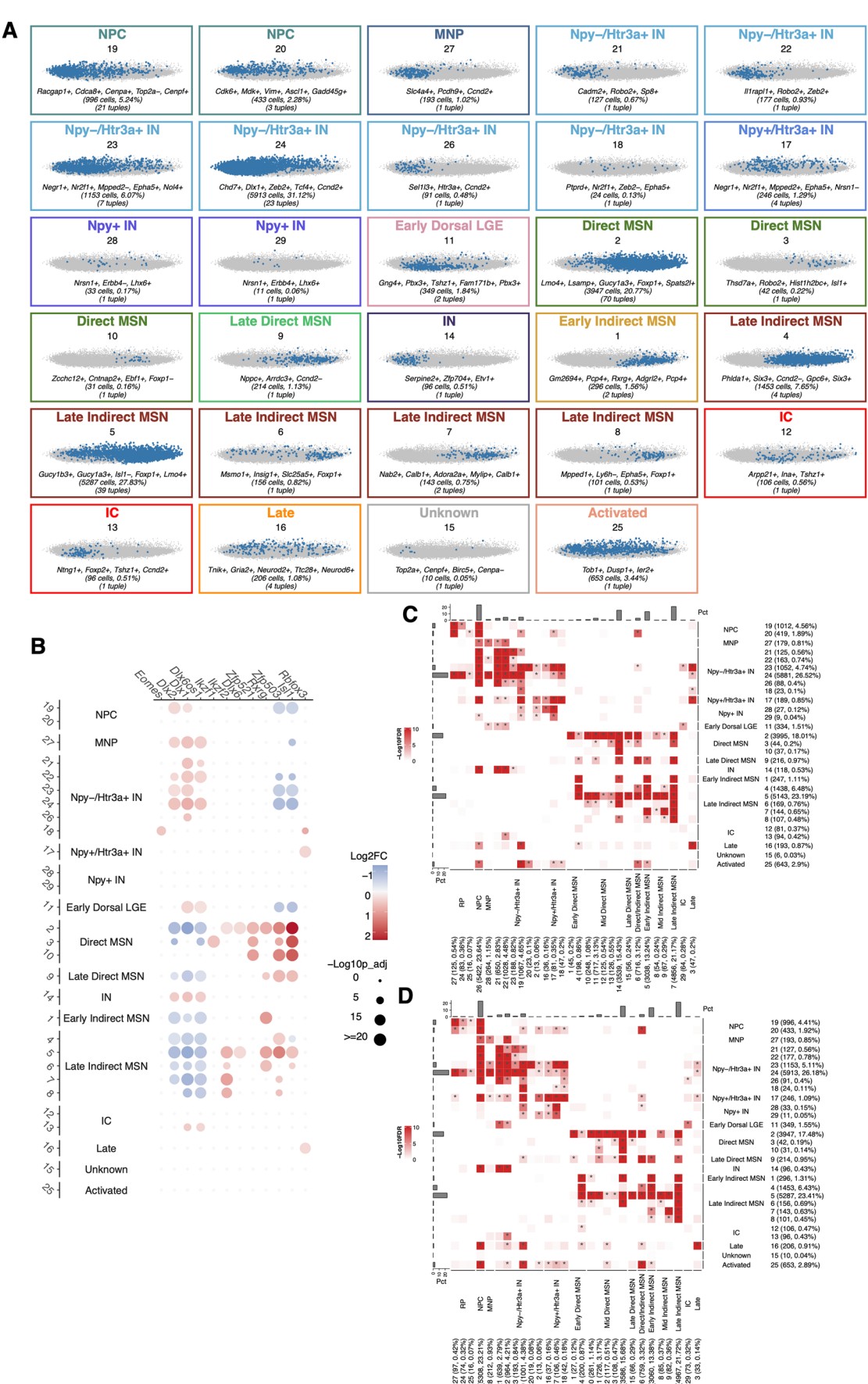

**Figure EV5. Expanded view for Fig. 4.**

A second dataset of developmental neurons testing for reproducibility. Disjoint neurons dataset for reproducibility. (**A**) Stator identifies 29 states for a disjoint set of $N = 19,000$ developmental neurons (set N2) at a Dice dissimilarity of 0.94, annotated when d-tuple and/or s2o-DEGs are marker genes for cell (sub)types or states. (**B**) Dot plot illustrating differential expression of neurogenesis marker genes across all Stator states. The size of the dots represents the $-\log_{10}$(Seurat p-val-adj) from differential expression testing between a state and all other states. Colour intensity represents the $\log_2$(FC) of gene expression. (**C**) The *x* axis shows the states obtained by running Stator on the Fig. 4 dataset (set N1), while the y axis shows states obtained by running Stator on the dataset in panel A (set N2). The N1 states, obtained by running Stator on N1 cells, are projected onto N2 cells. The enrichment of these projected states in N2 Stator states is computed using a hypergeometric test followed by the BH procedure to control the FDR at 5%. (**D**) As (**C**), but for N1 cells. The y axis shows the N2 states, while the x axis shows states obtained by running Stator on N1 cells. The N2 states are projected onto N1 cells, and their enrichment in each N1 cells' state is computed using a hypergeometric test followed by the BH procedure to control the FDR at 5%. These panels demonstrate reproducibility of Stator states on two disjoint sets of cells in the same biological condition.

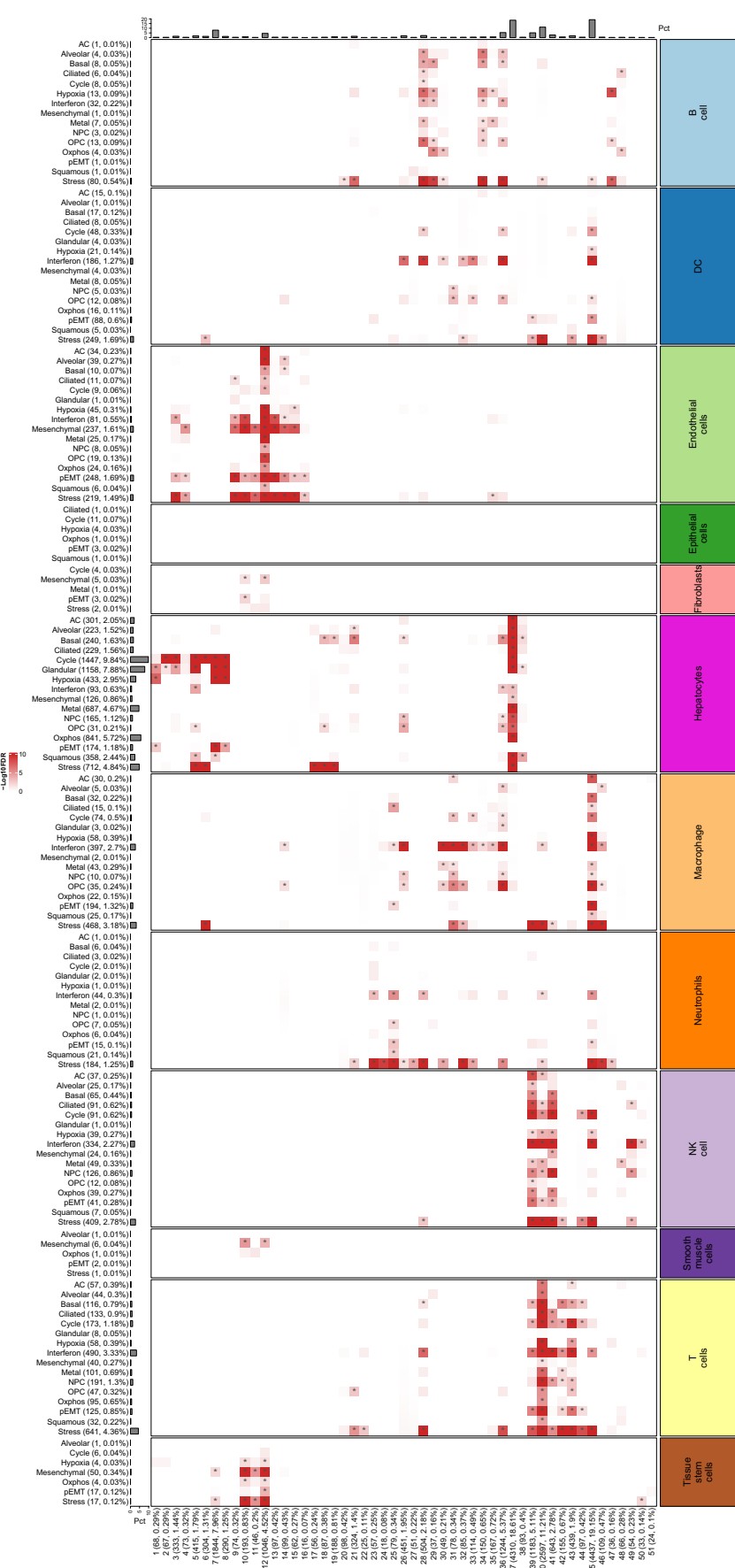

**Figure EV6.  Expanded view for Fig. 6.**

Stator states (columns) that are significantly enriched in HCC cells previously doubly-annotated by cell type (right) and cell state (left) inferred using singleR and NMF, respectively.

