## [Peer Review File · Molecular Systems Biology]

High order expression dependencies finely resolve cryptic states and subtypes in single cell data

Abel Jansma, Yuelin Yao, Jareth Wolfe, Luigi Del Debbio, Sjoerd Beentjes, Chris Ponting, and Ava Khamseh

Corresponding author(s): Ava Khamseh (ava.khamseh@ed.ac.uk) , Chris Ponting (chris.ponting@ed.ac.uk)

Review Timeline:

Submission Date:	9th Jan 24
Editorial Decision:	16th Jan 24
Appeal Received:	24th Jan 24
Editorial Decision:	30th Jan 24
Revision Received:	11th Feb 24
Editorial Decision:	22nd Mar 24
Revision Received:	29th Jul 24
Editorial Decision:	27th Sep 24
Revision Received:	24th Oct 24
Accepted:	31st Oct 24

Editors: Maria Polychronidou and Poonam Bheda

Transaction Report:

16th Jan 2024

Manuscript Number: MSB-2024-12214

Dear Dr Khamseh,

Thank you for submitting your manuscript "High order expression dependencies finely resolve cryptic states and subtypes in single cell data" to Molecular Systems Biology.

I have now had the chance to read your study and I have discussed it with the team. We have also consulted a member of the Advisory Editorial Board and I am afraid that the outcome was not positive.

In this study, you present Stator, a method for defining cell types, subtypes and states using the coordinated expression and non-expression of combinations of genes in single cells, without relying on local proximity of cells in gene expression space. We appreciate that you illustrate Stator on scRNAseq data from mouse embryonic brain, human healthy or cirrhotic liver and liver cancer and show that it recapitulates cell types defined by other methods but additionally provides higher resolution by resolving further sub-types and states e.g. newly defined liver cancer states whose differential expression involves genes that are predictive of patient survival. While Stator sounds potentially interesting, we feel that the value of the additional cell types/states defined by Stator, in terms of enabling new biological insights or applications in a clinical/diagnostic setup, remains to be further demonstrated.

Please note that we have also consulted a member of the Advisory Editorial Board who mentioned that in absence of further comparisons to other methodologies (e.g. SOTA methods such as CellHint (PMID: 38134877) or CellTypist (PMID: 35549406), recent foundation model efforts and deep learning approaches for automatic annotation) the methodological advance presented by Stator is not well supported. They also mentioned that concrete applications better supporting the value of the newly inferred states are lacking. As such, they did not recommend sending the study out for peer review.

Taken together and based on the recommendation we received from an AEB member we unfortunately have decided to not send the study out for review. I apologize for not bringing better news, but I hope that this early decision will allow you to decide how to proceed with your manuscript without undue delay.

Yours sincerely,

Maria Polychronidou, PhD
Senior Editor
Molecular Systems Biology

** As a service to authors, EMBO Press offers the possibility to directly transfer declined manuscripts to another EMBO Press title or to the open access journal Life Science Alliance launched in partnership between EMBO Press, Rockefeller University Press and Cold Spring Harbor Laboratory Press. The full manuscript and if applicable, reviewers' reports, are automatically sent to the receiving journal to allow for fast handling and a prompt decision on your manuscript. For more details of this service, and to transfer your manuscript please click on Link Not Available. **

Dear Dr Polychronidou,

RE: Manuscript Number: MSB-2024-12214

Thank you for your email (16 January) on our manuscript ("High order expression dependencies finely resolve cryptic states and subtypes in single cell data").

We respect Editorial decisions and views of Advisory Board members and so we ordinarily accept a desk rejection with equanimity and move on.

However, we think there is a fundamental problem with the Advisory Board member's suggestion that our method (Stator) does not provide a methodological advance compared to other methods "such as CellHint (PMID: 38134877) or CellTypist (PMID: 35549406), recent foundation model efforts and deep learning approaches for automatic annotation". Simply put, the problem is that these methods serve a different purpose from Stator's:

More specifically, each method (CellTypist, CellHint, foundation models, and deep learning):

- (1) Has an absolute requirement (for training) that cell type annotations already exist.
 - a. This means that all of these methods do not, as Stator does, identify cell types/states de novo, in a data-driven manner.
 - b. Moreover, such pre-existing cell type annotations are curated, and so are susceptible to subjective bias.

Rather than being applicable to the same problem as Stator, each of these other methods would be better applied subsequently, to Stator's data-driven annotations.

- (2) Is fundamentally different from Stator in that each does not annotate cell states, only cell (sub)types. Stator does both.
- (3) Has not yielded cell state annotations, described in the current literature, at the high resolution obtained by Stator, with respect to – for example – cell cycle sub-phases and cell type fates.

Stator's results are more extensive, as well as being at higher resolution. CellTypist is a logistic-regression-based model for automated cell annotation. To allow direct comparison, we applied CellTypist to the single cell data for mouse embryonic radial precursor cells (RPs) and neurons discussed in the submitted manuscript. This showed that CellTypist, using pre-defined labels, does not predict the cell states and subtypes highly resolved by Stator. CellTypist predicts only **4 types for RPs and 3 types for neurons** (with $n \geq 10$ cells in each type) whereas **Stator** predicts an order of magnitude more: **25 and 29** states/(sub)types for RPs and neurons respectively:

- For embryonic RPs: "Blood: Erythrocyte" [10,280; 86%], "Glioblast: Forebrain" [1,499; 13%], "Glioblast: Forebrain Astrocyte" [107; 1%], "Neuroblast: Neuronal intermediate progenitor" [30, 0.3%.
 - In comparison, Stator defines **25 states** ($n \geq 10$ cells) with only 221 (not 10,280, as above) labelled as blood cell (erythrocyte) contamination (Figure 2A of our submission).
- For neurons: "Blood: Erythrocyte" [14,268; 75%], "Forebrain GABAergic" [3,660; 19%], "Neuroblast: Forebrain GABAergic" [1,032, 5%.
 - In comparison, Stator defines **29 states** ($n \geq 10$ cells) (Figure 4A of our submission).

In summary, unlike Stator, CellTypist is unable to distinguish embryonic RPs and neurons from blood cells, and its results are at much lower resolution.

We would welcome detailed feedback from MSB peer reviewers on Stator's methods and applications. In light of our responses above, we hope that you reconsider your earlier decision.

Best regards,

Ava

Ava Khamseh
Lecturer in Biomedical AI
School of Informatics, and,
Institute of Genetics and Cancer
University of Edinburgh

31st Jan 2024

Manuscript Number: MSB-2024-12214R-Q

Title: High order expression dependencies finely resolve cryptic states and subtypes in single cell data

Dear Dr. Khamseh,

Thank you for your message asking us to reconsider our decision on your manuscript MSB-2024-12214. I have now had the chance to consider the manuscript and the points raised in your appeal letter and I have discussed them with the team. As I will explain below, we would offer to send out the manuscript for peer review, pending some revisions.

As I mentioned in my previous decision letter, our main concerns, which were echoed by our EAB member, were the following:

- i) the methodological advance presented by Stator seems somewhat unclear. Comparisons to other methodologies (e.g. SOTA methods such as CellHint (PMID: 38134877) or CellTypist (PMID: 35549406), recent foundation model efforts and deep learning approaches for automatic annotation) are lacking.
- ii) the value of the additional cell (sub-)types and states defined by Stator, in terms of enabling new biological insights or applications in a clinical/diagnostic setup, remains to be further demonstrated (i.e. Stator indeed defines substantially more (sub-)types and states, but what is their biological or clinical etc. relevance?)

We appreciate that in your appeal letter you explain the differences between Stator and CellHint, CellTypist and foundation model-based approaches. While Stator may indeed present differences and advantages compared to these approaches, we think that these approaches would nevertheless need to be mentioned in the manuscript, as related questions are also likely to come up during the review process. We would therefore ask you to revise the text to ensure a more inclusive mention of existing approaches (including those summarized above) and how Stator improves over them, in order to address point (i). Regarding point (ii), we understand that deriving concrete conclusions on the relevance of the additional cell (sub-)types and states derived by Stator involves substantial additional analyses, which you may not have at hand immediately. While we would encourage you to include some additional analyses along those lines, we understand if you prefer to not do so at this point and we would offer to send the manuscript out for review without the addition of such analyses.

I hope that the comments above will be helpful in deciding how to further proceed with this work.

Please use the link below to submit your revised manuscript:

Sincerely,

Maria Polychronidou

Maria Polychronidou, PhD
Senior Editor
Molecular Systems Biology

INSTITUTE OF
GENETICS & CANCER

11 February 2024

Medical
Research
CouncilTHE UNIVERSITY
of EDINBURGH
Dear Maria,

Thank you again for reconsidering our manuscript submission. We have now made the changes that you requested. Specifically:

(i) We have “revised the text to ensure a more inclusive mention of existing approaches”. **First**, we now include a paragraph (introduction on methods for automatic cell annotations) that cites CellTypist and provides an example of a foundation model (scGPT), and have explained, as in our appeal letter, why these methods are different from Stator. In particular, we state that they (i) require pre-existing cell type annotations for their training, and (ii) are not intended to annotate states. **Second**, we have run CellTypist across each of the 3 datasets in the manuscript, for (1) RPs and neurons, (2) normal and cirrhotic liver, and (3) HCC, and include the results in each corresponding “Results” subsection. In summary, we provide quantitative evidence that CellTypist is unable to distinguish embryonic RPs and neurons from blood cells in (1), recapitulates known major cell types in (2) without being able to detect rarer subtypes, e.g., subtypes of endothelial cells, and cell states, and in (3) inconsistently tags, as T-cells, ~40% of cells identified as transformed hepatocytes by the original publication’s authors.

(ii) With regard to “the value of the additional cell (sub-)types and states defined by Stator, in terms of enabling new biological insights or applications in a clinical/diagnostic setup”. We have **first** re-written the abstract and added two new paragraphs to the Discussion section making more explicit biological and clinical advantages of high-resolution data-driven quantification of cell subtypes and states in scRNA-seq data; we also provide further references. **Second**, we add a new Supplementary Table (27). This exemplifies how Stator’s finer resolution of neural stem and progenitor cell types relate to morphology or behaviour phenotypes when disrupted in mice, particularly in relation to 20-43 *minus* genes that partly define 110 neuronal and/or RP states. The roles of these genes in specifying cell state transitions during neurogenesis and in neurological disease can now be investigated at greater cellular and developmental resolution.

These changes further strengthen, we believe, the explanation of our method’s distinctions over existing approaches. We hope that they now allow the manuscript to undergo full peer review at *Molecular Systems Biology*.

Yours sincerely,

Ava Khamseh

Lecturer in Biomedical Artificial Intelligence
School of Informatics, and,
Institute of Genetics and Cancer
The University of Edinburgh
E: ava.khamseh@ed.ac.uk

Chris Ponting

Chair of Medical Bioinformatics
MRC Human Genetics Unit, and
Institute of Genetics and Cancer
The University of Edinburgh
E: chris.ponting@ed.ac.uk

Institute of Genetics and Cancer

The University of Edinburgh, Western General Hospital, Crewe Road, Edinburgh EH4 2XU
T: +44 (0)131 651 8500 ed.ac.uk/institute-genetics-cancer  @EdinUni_IGC

22nd Mar 2024

Manuscript Number: MSB-2024-12214RR

Title: High order expression dependencies finely resolve cryptic states and subtypes in single cell data

Dear Ava,

Thank you again for submitting your work to Molecular Systems Biology. We have now heard back from the three reviewers who agreed to evaluate your study. As you will see below, the reviewers raise a series of concerns, which preclude the publication of your study in its current form. However, given that the reviewers acknowledge that the study has the potential to be a relevant contribution to the single-cell biology field, we have decided to invite you to address the issues raised in a major revision.

Without repeating all the comments of the reviewers, some of the more fundamental issues are the following:

- A more thorough comparison to alternative approaches is required.
- Further analyses are required to better support that the derived cell types/states are "true and reproducible".
- The reviewers raise several technical concerns related to the data analysis and the method description.

The reviewers provide constructive suggestions on how to address these (and the rest of their) concerns. All issues raised would need to be satisfactorily addressed. As you may already know, our editorial policy allows in principle a single round of major revision. It is therefore essential to provide responses to the reviewers' comments that are as complete as possible. If you have any questions or if you would like to discuss your revision plan with me, please feel free to get in touch.

On a more editorial level, we would ask you to address the following points:

- We would recommend changing the article type to "Method".
- Please provide a .doc (or LaTeX) version of the manuscript text (including legends for main figures) and individual production quality figure files for the main Figures (one file per figure).
- Please include 5 keywords.
- The References should be formatted according to the Molecular Systems Biology reference style (i.e., ordered alphabetically and listing the first 10 authors followed by et al).
- All Materials and Methods need to be described in the main text. We would ask you to use 'Structured Methods', our new Materials and Methods format, which is mandatory for Methods and Articles with a strong methodological focus. According to this format, the Material and Methods section should include a Reagents and Tools Table (listing key reagents, experimental models, software and relevant equipment and including their sources and relevant identifiers) followed by a Methods and Protocols section in which we encourage the authors to describe their methods using a step-by-step protocol format with bullet points, to facilitate the adoption of the methodologies across labs. More information on how to adhere to this format as well as downloadable templates (.doc or .xls) for the Reagents and Tools Table can be found in our author guidelines: . An example of a Method paper with Structured Methods can be found here: .
- We have replaced Supplementary Information by the Expanded View (EV format). In this case all additional figures should be provided in a PDF called Appendix. Please include a Table of Contents (with page numbers) in the beginning of the Appendix. Appendix figures should be labeled and called out as: "Appendix Figure S1, Appendix Figure S2..." etc. Each legend should be below the corresponding Figure in the Appendix. For detailed instructions regarding expanded view please refer to our Author Guidelines: .
- Please include a Data availability section describing how the data, code etc. have been made available. This section needs to be formatted according to the example below:
The datasets and computer code produced in this study are available in the following databases:
 - Chip-Seq data: Gene Expression Omnibus GSE46748 (<https://www.ncbi.nlm.nih.gov/geo/query/acc.cgi?acc=GSE46748>)
 - Modeling computer scripts: GitHub (<https://github.com/SysBioChalmers/GECKO/releases/tag/v1.0>)
 - [data type]: [full name of the resource] [accession number/identifier] ([doi or URL or identifiers.org/DATABASE:ACCESSION])
- Please include a "Disclosure and Competing Interests Statement" in the main text.
- Please note that our editorial policy does not allow "Data not Shown" statements.

- Please make use that all figure panels are correctly called out in the text. E.g. currently a callout is missing for Fig. 3A, and there is one Fig. 3C, but no such panel exists.
 - Supplementary Tables 1-27 should be provided as EV Tables (for less complex tables, not longer than one page) or EV Datasets (for more complex tables, longer than one page). Please provide one file per EV Table/Dataset. In each file, a description of the table/dataset should be provided in a separate tab.
 - Please provide a "standfirst text" summarizing the study in one or two sentences (approximately 250 characters), three to four "bullet points" highlighting the main findings and a "synopsis image" (550px width and max 400px height, jpeg format) to highlight the paper on our homepage.
 - For data quantification: please specify the name of the statistical test used to generate error bars and P values, the number (n) of independent experiments (specify technical or biological replicates) underlying each data point and the test used to calculate p-values in each figure legend. The figure legends should contain a basic description of n, P and the test applied. Graphs must include a description of the bars and the error bars (s.d., s.e.m.).
 - The manuscript sections should be in the following order: Title page - Abstract & Keywords - Introduction - Results - Discussion - Materials & Methods - Data Availability - Acknowledgments - Disclosure Statement & Competing Interests - References - Figure Legends - Tables with legends - Expanded View Figure Legends.
 - Please include a "Disclosure & Competing Interests Statement" in the main text.
 - Molecular Systems Biology supports formal data citations in the Reference list, to cite previously published datasets. In addition to citing the original papers that reported the data, we encourage you to also cite the relevant datasets directly in the Reference list. In the text, references to datasets are included as "Data ref: Smith et al, 2001" or "Data ref: NCBI Sequence Read Archive PRJNA342805, 2017". In the Reference list, data citations are very similar to normal literature references but must be labeled with "[DATASET]" at the end of the reference. For detailed instructions please refer to our Author Guidelines .
 - When you resubmit your manuscript, please download our CHECKLIST (<https://bit.ly/EMBOPressAuthorChecklist>) and include the completed form in your submission.
- *Please note* that the Author Checklist will be published alongside the paper as part of the transparent process (<https://www.embopress.org/page/journal/17444292/authorguide#transparentprocess>).

If you feel you can satisfactorily deal with these points and those listed by the referees, you may wish to submit a revised version of your manuscript. Please attach a covering letter giving details of the way in which you have handled each of the points raised by the referees. A revised manuscript will be once again subject to review and you probably understand that we can give you no guarantee at this stage that the eventual outcome will be favorable.

Kind regards,

Maria

Maria Polychronidou, PhD
Senior Editor
Molecular Systems Biology

We realize that it is difficult to revise to a specific deadline. In the interest of protecting the conceptual advance provided by the work, we recommend a revision within 3 months (20th Jun 2024). Please discuss the revision progress ahead of this time with the editor if you require more time to complete the revisions. Use the link below to submit your revision:

IMPORTANT: When you send your revision, we will require the following items:

1. the manuscript text in LaTeX, RTF or MS Word format
2. a letter with a detailed description of the changes made in response to the referees. Please specify clearly the exact places in the text (pages and paragraphs) where each change has been made in response to each specific comment given
3. three to four 'bullet points' highlighting the main findings of your study
4. a short 'blurb' text summarizing in two sentences the study (max. 250 characters)
5. a 'thumbnail image' (550px width and max 400px height, Illustrator, PowerPoint or jpeg format), which can be used as 'visual title' for the synopsis section of your paper.
6. Please complete the CHECKLIST available at (<https://bit.ly/EMBOPressAuthorChecklist>).

Please note that the Author Checklist will be published alongside the paper as part of the transparent process (<https://www.embopress.org/page/journal/17444292/authorguide#transparentprocess>).

See also figure legend guidelines: <https://www.embopress.org/page/journal/17444292/authorguide#figureformat>

8. Please note that corresponding authors are required to supply an ORCID ID for their name upon submission of a revised manuscript (EMBO Press signed a joint statement to encourage ORCID adoption).

(<https://www.embopress.org/page/journal/17444292/authorguide#editorialprocess>)

Currently, our records indicate that the ORCID for your account is 0000-0001-5203-2205.

Link Not Available

*** PLEASE NOTE *** As part of the EMBO Press transparent editorial process initiative (see our Editorial at <https://dx.doi.org/10.1038/msb.2010.72>), Molecular Systems Biology publishes online a Review Process File with each accepted manuscripts. This file will be published in conjunction with your paper and will include the anonymous referee reports, your point-by-point response and all pertinent correspondence relating to the manuscript. If you do NOT want this File to be published, please inform the editorial office at msb@embo.org within 14 days upon receipt of the present letter.

Reviewer #1:

Jansma et al. previously developed a mathematical definition (MFI) for n-point interactions between binary variables [1,2]. Here, they present the Stator algorithm which builds on the concept of MFIs towards detection of cell states based on interaction between $3 \leq k \leq 7$ genes in binarized single-cell RNA-seq (scRNA-seq). Stator is tested on 3 public datasets: E18 mouse brain, human liver from cirrhosis and control donors, and human hepatocellular carcinoma.

Stator addresses an important open problem in single-cell analysis in a novel way that clearly sets it apart from other methods in the field. However, the manuscript could be significantly strengthened by a more rigorous comparison of Stator to other methods that solve similar problems. In addition, the manuscript mischaracterizes some of the recent research on low-dimensional embeddings of single-cell transcriptome and could better define terms that are currently loosely defined.

Importantly, Stator is limited 40k cells (p. 19) due to its computation costs, which will require users to downsample their input data (100k-1M cell datasets are quite common). The manuscript does not currently offer an analysis of the reproducibility of states after random downsampling of the data. It also doesn't test reproducibility of the recovered states in independent datasets from the same biological conditions.

Major comments

1. Mischaracterization of the scientific discussion on low-dimensional embeddings of scRNA-seq

High-dimensional transcriptomics are often not analyzed in the full dimension but rather projected into a low-dimension space. The projection is commonly done in two steps, and the manuscript does not distinguish between them, resulting in misleading statements.

Transcriptomic data is first projected into a set of K (often $K \sim 50$) dimensions with methods that offer quantitative guarantees for the preservation of Euclidean distances in the low dimension. Two examples are PCA (scanpy and Seurat's default pipelines, both cited in the manuscript) and scvi-tools [3,4]. While the low-dimension embedding does not fully preserve the Euclidean distances of the original space, its distances are still quantitatively meaningful contrary to statements such as "a common approach, whose theory is however problematic (Chary and Pachter, 2023), defines a cell type as a collection of cells that group more closely in gene expression space" or "since embeddings can distort distances, densities cannot be directly interpreted" (Figure 3, in reference to PCA). Subsequently, and for the purpose of visualization, a procedure that does not offer distance guarantees (tSNE or UMAP) is used to project the K-dimension data to 2d. Despite the lack of guarantees, researchers may

read too much into these plots. Even when there is a guarantee then a sufficiently number of dimensions is needed for the preserved distances to be meaningful. These are the points argued by Chary and Pachter in the paper that the authors cite, and see also [5].

In other places, the manuscript relies on visual evaluation of the lack of mixing in 2d PCA plots to argue for discovery of cell states that are not proximal in gene expression space: "some states (e.g., #1 and #2) are localized in a PCA embedding, but most are not...". This is unjustified because of the low level of variance captured by even the first principal components of scRNA-seq (often ~1%), see for example [5].

2. Loosely defined terms (cell types, subtypes, and states)

The distinction between cell types and cell states is crucial to the way the introduction positions the manuscript's contribution and repeated multiple times in the text, yet the manuscript never defines them. Admittedly, these definitions are extremely challenging, yet the authors could aid the reader in appreciating their contribution by engaging with some of the attempts to do so (e.g., transience vs. permanence [6], future phenotype in response to environmental stimulus [7] or perturbation [8]).

Some of the technical terms (d-tuple, minus genes) used in describing the Stator algorithm can be inferred from the context, but the manuscript would be clearer if they were explicitly defined.

3. Lack of rigorous comparison to other methods

It is admittedly difficult to evaluate a method aimed at uncovering novel cell states due to the lack of a ground truth. The authors' validation seems to rely mostly on interpreting the states recovered in the studied dataset and showing that they represent interesting biology, however this method seems anecdotal and prone to confirmation bias. The comparison to CellTypist is moot because the latter solves a different annotation problem and is expected to fail in the lack of relevant marker sets, as is the case of the cell states studied here. The statement "for each dataset, we compared Stator with 4 standard or state-of-the-art methods... [clustering, Milo, NMF, CellTypist]" is incorrect because not all 4 comparisons were made for each of the 3 datasets. Further, lumping together these 4 approaches is problematic because each addresses a different problem in single-cell analysis.

The manuscript can be strengthened by 1) analyzing Stator's performance on simulated data; 2) by a more thorough comparison of Stator-inferred states to states inferred with NMF (see also below) and other methods inferring continuous states [9,10]; 3) showing Stator's state are recapitulated across independent datasets from the same biological system; and 4) testing Stator on transcriptomes taken from multiomic datasets (e.g., CITE-seq, ATAC-seq), where the other data modality can serve as an unbiased control for discovery of states that aren't captured from the RNA alone with clustering or other methods.

Testing Stator's ability to recover the same states across independent datasets is closely related to testing its ability to recover the same states in downsampled data, which is crucial due to Stator's input limits as described above.

Notably, Stator is already tested against NMF on 4 liver tumor samples. These samples are a subset of 19 samples that were previously analyzed together with NMF to recover recurrent cell states across diverse cancer types [11]. Why were only 4 out of the 19 samples analyzed? Could the authors analyze all 19 and compare the states recovered by Stator to the states recovered by the original study with NMF?

Minor comments

1. Figure 1b - red and green edges / circles appear but their meaning is explained only in Fig. S1. Similarly, meaning of red-blue is not explained in Fig1c.
2. Figure 2a and the similar ones later: the font is too small. In my opinion, it is redundant to show PCA plots for all the Stator states because of the limited value of PCA proximity evaluation in that context, as described above (I expect the first PCs to capture ~1% of the variance). The authors should denote the captured variance on PCA axes to allow the reader to better evaluate these plots. Is there a meaning to the order by which state panels appear (currently, 1, 10, 14, 24, ...)?
3. Figure 6 legend typo: "Nevertheless, in the main these cells are only labelled" - in the main what?
4. p. 2: "Automatic cell annotation methods, such as CellTypist (Condo et al. 2022), foundation (Cui et al. 2023), and deep learning models, are not intended..." - the reference to "deep learning" is too general, especially with no studies cited.
5. Methods 4.3: "Dice similarity between X and Y is defined as:" should be a quantity that increases when X,Y are similar but equation 6 gives instead the Dice distance (decreases when X,Y are similar).
6. Methods: no details provided on details of how NMF and Milo were used, parameters provided etc.

References

1. Beentjes SV, Khamseh A. Higher-order interactions in statistical physics and machine learning: A model-independent solution to the inverse problem at equilibrium. *Phys Rev E*. 2020;102: 053314.
2. Jansma A. Higher-order interactions and their duals reveal synergy and logical dependence beyond Shannon-information.

Entropy. 2023;25. doi:10.3390/e25040648

3. Lopez R, Regier J, Cole MB, Jordan MI, Yosef N. Deep generative modeling for single-cell transcriptomics. *Nat Methods*. 2018;15: 1053-1058.
4. Virshup I, Bredikhin D, Heumos L, Palla G, Sturm G, Gayoso A, et al. The scverse project provides a computational ecosystem for single-cell omics data analysis. *Nat Biotechnol*. 2023. doi:10.1038/s41587-023-01733-8
5. Cooley SM, Hamilton T, Deeds EJ, Ray JCJ. A novel metric reveals previously unrecognized distortion in dimensionality reduction of scRNA-Seq data. *bioRxiv*. 2019. p. 689851. doi:10.1101/689851
6. Wagner A, Regev A, Yosef N. Revealing the vectors of cellular identity with single-cell genomics. *Nat Biotechnol*. 2016;34: 1145-1160.
7. Morris SA. The evolving concept of cell identity in the single cell era. *Development*. 2019;146. doi:10.1242/dev.169748
8. Fleck JS, Camp JG, Treutlein B. What is a cell type? *Science*. 2023;381: 733-734.
9. Svensson V, Gayoso A, Yosef N, Pachter L. Interpretable factor models of single-cell RNA-seq via variational autoencoders. *Bioinformatics*. 2020;36: 3418-3421.
10. Choi Y, Li R, Quon G. siVAE: interpretable deep generative models for single-cell transcriptomes. *Genome Biol*. 2023;24: 29.
11. Barkley D, Moncada R, Pour M, Liberman DA, Dryg I, Werba G, et al. Cancer cell states recur across tumor types and form specific interactions with the tumor microenvironment. *Nat Genet*. 2022;54: 1192-1201.

Reviewer #2:

Jansma and Yao propose a new tool of revealing underlying biology of single cell datasets, not at the level of cell types, but instead focusing on higher order gene expression interactions that represent phases/states. This is an interesting complement to the cell type annotation-driven single label approaches currently used, and has the potential to further aid the interpretation of single cell data.

Major comments:

1. The validity of some of the resolved states are not convincing, as a handful or even single DE genes are used to justify them. There is a concern they may be overclustering for noise (such as dropout).
2. A large amount of effort focuses on discussing cell cycle phase, whose inference has been done perhaps less elegantly for years now either through gene set expression scoring or other tools such as Tricycle. The authors have not convincingly demonstrated that Stator reveals similar or cleaner cycle states than previously used approaches.
3. 50+ potential states can be difficult for investigators to navigate and interpret. Can the authors provide guidelines for merging and annotating them, outside of the shiny app interface?

Minor comments:

1. While computation time explains the recommended upper limit of using 1000 highly variable genes, has performance (number and interpretability of the states) been assessed for using less or more genes? Has the author considered subsampling (or Seurat "sketching" type of informed subsampling) of cells to ease computational load?
2. The authors should be applauded for doing the work to package the tool in a Nextflow workflow and docker images, which will facilitate its adoption. Please also add a bit more guidelines on setup and launch of the pipeline on the Github page for users less experienced. I also could not see some of the figures in the accompanying shiny app with HCC demonstration data (GO, rrvgo). Please look into whether this is an issue with code or user error on my part (and perhaps adjust UI/tutorials to help with confusion)
3. For cluster 17 in Supplemental Table 3 and 4, 12 DE genes yielded 98 significant GOBP terms. Should the single hit GO terms even be reported?

Reviewer #3:

In this manuscript, the authors introduce a novel computational method, Stator, designed to identify cell states or types within single-cell RNA-seq datasets. Stator operates under the assumption that cell states can be delineated by the coordinated expression of specific and small sets of genes. Initially, Stator identifies candidate gene combinations marking cell states through a method previously developed by the same authors (MFI), subsequently assigning cells to states based on the presence or absence of these identified gene combinations within each cell.

The authors apply Stator across multiple datasets, claiming, for example, the capability to discern several biologically relevant states (e.g., phases of the cell cycle) within the same cell types.

Characterizing cell types or states is a significant challenge, highlighted by the proliferation of algorithms. Most existing methods assume that cells belonging to the same type/state exhibit greater transcriptomic similarity, and, hence, they rely on cell clustering techniques.

Stator distinguishes itself from these approaches through two conceptual distinctions: firstly, it aims to identify higher-order correlations in gene expression, specifically targeting coordinated expression patterns of up to 7 genes. Secondly, it avoids

traditional cell clustering methodologies, thereby enabling the identification of cell states that may be shared across different cell types, not necessarily proximal within the transcriptomic space.

While the concepts introduced by Stator are interesting, particularly in the context of analyzing scRNA-seq datasets marked by sparsity and large cell populations, this manuscript exhibits important limitations. In particular, I am concerned about the validation of the results and the apparent underestimation of the critical aspect of cell state annotation, which might limit Stator's applicability.

Major points

- **Validation.** A fairly large number of states is identified in all the datasets, but there's little validation shown beyond GO terms and manually selected marker genes. I believe more independent validation is needed to convince the reader that a large number of states (many having the same annotation) are not deriving from overfitting but rather represent "real" cell states. For instance, Stator could be run on datasets where cells have been previously labelled, with, e.g., FACS sorting, etc.
- **Annotation.** In many cases, how the authors established the annotation of cell states is unclear. For example, claims like the following: "The 7 states' s2o-DEGs were predominantly cell cycle markers, confirming them as cell cycle states." (page 7) should be more quantitative (what's the fraction of cell cycle markers? is there a statistically significant enrichment?). Or, when they write "Demanding that at least 3 s2o-DEGs are known markers of an annotation" (page 7): does it ever happen that one state gets more than 3 markers for different annotations? how do the authors resolve the ambiguity in these cases? While cell type/state annotation from scRNA-seq is still an open problem and might not be the focus of this paper, I believe some more quantitative and objective criteria when performing annotation are needed to validate Stator's results.
- **Comparison with other methods.** As the authors also notice, the comparison of Stator with CellTypist doesn't seem appropriate, given that CellTypist needs previously annotated cell states/types. Hence, I suggest either removing or drastically reducing the comparisons with CellTypist and replacing them with the use of other unsupervised methods (for instance, <https://www.nature.com/articles/s41467-023-43406-9> , or methods based on topic modelling, like hSBM <https://www.mdpi.com/2072-6694/14/5/1150>).
- **Robustness with respect to number of starting genes.** How robust are the states (in terms of identity and number) if a different number or subsets of HVGs is chosen? This robustness analysis could also reduce the number of states identified, and could help focus on a smaller (but more robust) number of states
- **Blood contamination.** The authors identify a "blood contamination" issue with some of the datasets they analyze. I guess this could be due to ambient mRNA contamination, so I suggest trying to remove this before running their analysis.
- **Cell cycle.** Regarding the cell cycle, it might be interesting to compare their results with an ad-hoc cell cycle predictor algorithm (like the one found in the `seurat` or `scrna` R libraries).
- **Cirrhotic or healthy liver samples analysis.** In the analysis of cells from cirrhotic or healthy liver samples (Figure 5), it seems like Hepatocytes, Mesothelia and pDCs don't correspond to any state that is exclusively enriched for that cell type (Figure 5B). For example, pDCs are enriched in the states 11, 15 and 45; but these states are also enriched with other cell types. Does this mean that Hepatocytes, Mesothelia and pDCs are not correctly identified by Stator?

Response Summary:

We thank the reviewers for their thorough and constructive comments. Please find below the summary of additional analyses and changes to the manuscript, followed by a point-by-point response to the reviewers' comments. (Green text below refers to the Editor's summary).

- "A more thorough comparison to alternative approaches is required."

1. Comparison of Stator with NMF on all 3 datasets: (a) mouse brain (RP), (b) human liver cirrhosis and (c) human liver cancer.
2. Comparison of Stator with LDVAE (deep learning-based) and LDA (topic modelling) on the mouse brain (RP) dataset.
3. Comparison of Stator cell cycle states with Tricycle on the mouse brain (RP) data.

- "Further analyses are required to better support that the derived cell types/states are "true and reproducible"."

4. Testing Stator on transcriptomes taken from multiomic dataset (scRNA-seq + scATACseq) from Embryonic E18 Mouse Brain (5k cells); additionally, this provides a second and disjoint dataset for the same biological condition.
5. Comparisons with already validated results from cell staining, flow cytometry and immunofluorescence for the liver disease data.
6. Projection of Stator cancer cell states from one cancer type (HCC) to another cancer type (cholangiocarcinoma).
7. Clarification on testing the reproducibility of Stator states on disjoint cells (in the same biological condition), which was previously presented in a supplementary table but not clearly referenced in the main text (Supplementary Figs S7 and S13, now S8 and S15 panels C and D).
8. Additional sensitivity analyses by rerunning Stator on RP data where (i) cells in a given state (blood contamination) are artificially removed, and (ii) artificial cells with a new d-tuple are numerically induced, to test that these states are indeed correctly (i) not detected, and (ii) detected by a new Stator run, respectively.
9. Further text explaining references to authors' previous publications (including various simulations with known ground truth of the core Stator MFI estimator) is now added to the Methods section.

- "The reviewers raise several technical concerns related to the data analysis and the method description."

10. Clarifications on statements regarding dimensionality reduction to avoid mis-characterisation.
11. Changes to Figures 2A and 4A (removal of focus on PC visualisation).

12. Removal of CellTypist comparison, and clarification on comparison with Milo.
13. Addition of a step-by-step recipe for labelling Stator states.
14. Additional online documentation for Stator's nextflow pipeline (GitHub page) and Stator's shiny R app.

Reviewer comments (in black) and response (in blue):

Reviewer #1:

Jansma et al. previously developed a mathematical definition (MFI) for n-point interactions between binary variables [1,2]. Here, they present the Stator algorithm which builds on the concept of MFIs towards detection of cell states based on interaction between $3 \leq k \leq 7$ genes in binarized single-cell RNA-seq (scRNA-seq). Stator is tested on 3 public datasets: E18 mousebrain, human liver from cirrhosis and control donors, and human hepatocellular carcinoma. Stator addresses an important open problem in single-cell analysis in a novel way that clearly sets it apart from other methods in the field. However, the manuscript could be significantly strengthened by a more rigorous comparison of Stator to other methods that solve similar problems. In addition, the manuscript mischaracterizes some of the recent research on low-dimensional embeddings of single-cell transcriptome and could better define terms that are currently loosely defined.

Importantly, Stator is limited 40k cells (p. 19) due to its computation costs, which will require users to downsample their input data (100k-1M cell datasets are quite common). The manuscript does not currently offer an analysis of the reproducibility of states after random downsampling of the data. It also doesn't test reproducibility of the recovered states in independent datasets from the same biological conditions.

Number of cells: Downsampling from 100k-1M to <40k cells is only problematic if this worsens the resolution by which cell (sub)types and states are identified, especially those that are proportionally rare. As we showed in the original submission, even from a diverse set of 20,000 liver cells "Stator has labelled [seven] cell subtypes from among a previously homogeneous set of ECs that were scarce in this dataset (<2.5%)". Three Stator "Cilia" states each label <0.21% of 20,000 cells, and a fourth 3.4% (Figure 2), whereas NMF combined these populations into a single 1.1% Cilia state (Response Fig. 3A, below). When Stator provides fine resolution at fewer cells, it does so at lower cost, an important criterion for most research labs. To further increase state resolution, Stator can also be applied to a cell subpopulation with broadly defined cell type. Stator states can further be projected into data sets of any size, incurring little computational cost. These considerations have now been added to a subsection in Methods "Stator state projection to disjoint data".

State Projection also demonstrated Stator's reproducibility after random downsampling into two disjoint subsets. This was done twice: once for RPs (Fig. S8 panels C and D) and once for neurons (Fig. S15 panels C and D). Unfortunately, this was not explained in the main text and the figures were not properly referenced, for which we apologise. This is now explicitly stated in the main text, with these supplementary figures referenced.

Major comments

1. Mischaracterization of the scientific discussion on low-dimensional embeddings of scRNA-seq

We agree with the reviewer. We have made changes to the manuscript on each of the points related to dimensionality reduction (below).

High-dimensional transcriptomics are often not analyzed in the full dimension but rather projected into a low-dimension space. The projection is commonly done in two steps, and the manuscript does not distinguish between them, resulting in misleading statements.

Transcriptomic data is first projected into a set of K (often $K \sim 50$) dimensions with methods that offer quantitative guarantees for the preservation of Euclidean distances in the low dimension. Two examples are PCA (scanpy and Seurat's default pipelines, both cited in the manuscript) and scvi-tools [3,4]. While the low-dimension embedding does not fully preserve the Euclidean distances of the original space, its distances are still quantitatively meaningful contrary to statements such as "a common approach, whose theory is however problematic (Chary and Pachter, 2023), defines a cell type as a collection of cells that group more closely in gene expression space" or "since embeddings can distort distances, densities cannot be directly interpreted" (Figure 3, in reference to PCA). Subsequently, and for the purpose of visualization, a procedure that does not offer distance guarantees (tSNE or UMAP) is used to project the K -dimension data to 2d. Despite the lack of guarantees, researchers may read too much into these plots. Even when there is a guarantee then a sufficiently number of dimensions is needed for the preserved distances to be meaningful. These are the points argued by Chary and Pachter in the paper that the authors cite, and see also [5].

In agreement with the reviewer, the following sentence has now been removed from the manuscript: "A common approach, whose theory is however problematic (Chary and Pachter, 2023), defines a cell type as a collection of cells that group more closely in gene expression space than other cells."

Instead, we now state: "Two stages of dimensionality reduction is commonly used in scRNA-seq

analysis pipelines. The first is projection of cells into $K \leq 50$ dimensions using methods that, whilst not fully preserving Euclidean distances in lower dimensions, produce an embedding whose distances are still quantitatively meaningful. This includes PCA, which features in Scanpy (Wolf et al., 2018) and Seurat's (Stuart et al., 2019) default pipelines. This dimensionality reduction is done to reduce noise and to avoid the curse of dimensionality in downstream analyses that rely on quantification of Euclidean distances, such as clustering, which would otherwise break down in high dimensions (Aggarwal et al., 2001).”, and “The second stage of dimensionality reduction further reduces the K -dimensional space to 2 or 3 dimensions, often using tSNE and UMAP, for qualitative and exploratory analysis through visual inspection. Due to the lack of guarantees on distance preservation (for tSNE and UMAP), such extreme dimensionality reduction (even for PCA) inevitably results in significant distortions (Cooley et al., 2022; Chary and Pachter, 2023).”

Regarding “since embeddings can distort distances, densities cannot be directly interpreted” in Fig 3, this is in reference to a 2-dimensional visualisation. We have modified this sentence to “since a 2-dimensional PCA embedding can distort distances, densities cannot be directly interpreted.”

In other places, the manuscript relies on visual evaluation of the lack of mixing in 2d PCA plots to argue for discovery of cell states that are not proximal in gene expression space: "some states (e.g., #1 and #2) are localized in a PCA embedding, but most are not...". This is unjustified because of the low level of variance captured by even the first principal components of scRNA-seq (often $\sim 1\%$), see for example [5].

We agree and now have replotted Figs 2A and 4A, replacing PC visualisation for all states with a bar plot. These demonstrate: that clustering (in 20 PCA dimensions) results in few statistically significant clusters; that these clusters correspond in various proportions to multiple Stator states; and thus, that these clusters are unable to separate different biological types and states. In the updated versions of these figures, we have chosen to retain three representative PC panels. This is so that readers may see that cells in different states are differently colocalised and spread. Variance explained is higher than the reviewer anticipated: for RPs, PC1 and PC2 explain 6.2% and 4.9% of the variation respectively, and for neurons, PC1 and PC2 explain 5.6% and 4.9% of the variation, respectively.

Response Fig1: Fig 2A (updated in the manuscript): “(A) Barplot colours indicate the proportion of cells captured by each cluster following hierarchical clustering of cells (see FigS7) that resulted in two significantly different clusters only. Right-hand side: three exemplar Stator states (#2, #8, #11) are highlighted in a PCA embedding of the unbinarised expression data, and annotated with the number of cells they label, the d-tuples from which they are defined, and their five most common d-tuple genes.”

Our emphasis here is that (text now added to Section 2.2 of the manuscript “Cell cycle states in embryonic neurons and RPs”):

“Stator does not rely on Euclidean distances, and thus does not require the first-stage dimensionality reduction to ≤ 50 dimensions to avoid the curse of dimensionality, nor does it require the 2D visualisation in the second step. Moreover, because Stator does not rely on proximity of cells in expression space, it permits different sub-populations to co-exist in the same biological state.

For example, cells of different type, here e.g., RPs and early neurons, can exist in the same biological state, e.g., G2/M phases of the cell cycle. If proximity of cells in expression space is influenced most by cell type, then states occupied by multiple cell types will often be missed. For example, Fig. 3A presents a heterogeneous dataset containing a combination of neurons (predominantly left) and RPs (right), for which PC1 explains $\sim 10\%$ of the total variation (with remaining PCs explaining $<2\%$ each). Yet, each of these cell types clearly includes some cells occupying the same cell cycle state, e.g., G2/M. Stator readily detects such states from homogeneous cells or the combination of two cell types.”

Response Fig2: Joint neuron and RP data, with cell cycle Stator states indicated on PC plots, spanning the two different cell types. PC1 explains ~10% of the variation.

2. Loosely defined terms (cell types, subtypes, and states)

The distinction between cell types and cell states is crucial to the way the introduction positions the manuscript's contribution and repeated multiple times in the text, yet the manuscript never defines them. Admittedly, these definitions are extremely challenging, yet the authors could aid the reader in appreciating their contribution by engaging with some of the attempts to do so (e.g., transience vs. permanence [6], future phenotype in response to environmental stimulus[7] or perturbation [8]).

In the revised manuscript we now cite these publications at the beginning of the Introduction: “We follow others (Wagner et al. (2016); Morris et al. (2019); Fleck et al. (2023)) by distinguishing cell types - with their more permanent phenotypic features - from cell states, whose features are transient and can be elicited by stimulus; cell sub-types are sub-populations of the same cell type that share distinctive features.”

Some of the technical terms (d-tuple, minus genes) used in describing the Stator algorithm can be inferred from the context, but the manuscript would be clearer if they were explicitly defined.

Thank you for pointing out these omissions. The Main Text now defines d-tuples as: “gene tuples that significantly drive interactions (Methods). This step is achieved by comparing the expression of each tuple of genes in the MFI estimator to their expression under the null distribution of independence (see Methods for full details).” Deviating gene tuples (d-tuples) have been defined mathematically in section 4.2 of Methods, and in the legend to Fig 1C. We also have now added “Absence of expression for a gene is denoted by a *minus*.”

3. Lack of rigorous comparison to other methods

It is admittedly difficult to evaluate a method aimed at uncovering novel cell states due to the

lack of a ground truth. The authors' validation seems to rely mostly on interpreting the states recovered in the studied dataset and showing that they represent interesting biology, however this method seems anecdotal and prone to confirmation bias. The comparison to CellTypist is moot because the latter solves a different annotation problem and is expected to fail in the lack of relevant marker sets, as is the case of the cell states studied here. The statement "for each dataset, we compared Stator with 4 standard or state-of-the-art methods... [clustering, Milo, NMF, CellTypist]" is incorrect because not all 4 comparisons were made for each of the 3 datasets. Further, lumping together these 4 approaches is problematic because each addresses a different problem in single-cell analysis.

We agree with the reviewer that comparison with CellTypist is not relevant here. Our comparison with CellTypist was made in response to a request by the editor based on advice from the external advisory board. CellTypist comparisons with Stator have now been removed from the manuscript. Regarding Stator comparison with other methods, we agree that our previous text was inaccurate, and it has now been amended. A new subsection and figure are added to Methods "Comparison with other methods" for comparison of Stator with NMF, LDVAE and LDA on embryonic mouse brain (RP) data. The comparison with NMF liver disease data is added to its corresponding section in Results. Comparison with NMF on liver cancer data remains unchanged.

With regards to Milo, we are aware and remark in the manuscript that Milo is a differential abundance testing method. We recommend using Milo for addressing questions with respect to abundance, e.g., across different biological conditions. Nevertheless, when differential abundance exists, the expectation is that there are enrichments of different states across the conditions which should also be detectable by Stator. This is why we compare Stator with Milo for the healthy vs cirrhotic liver data. We have now clarified this in the manuscript.

The following has been added to introduction of Results:

"For each dataset, we compare Stator with NMF (Barkley et al., 2022; Gaujoux and Seoighe, 2010). We further compare Stator's output on the RPs' dataset with (i) clustering with pairwise significance quantification (Stuart et al., 2019; Gao et al., 2022) (Fig. 2A, S7), (ii) LDVAE (deep learning-based) (Svensson et al., 2020; Gayoso et al., 2022), (iii) LDA (topic modelling) (Blei et al., 2003; Srivastava and Sutton, 2017; Gayoso et al., 2022), and (iv) its cell cycle states with Tricycle (Zheng et al., 2022b). Analyses (ii)-(iv) are presented in Materials & Methods "Comparison with other methods"

The manuscript can be strengthened by 1) analyzing Stator's performance on simulated data; 2) by a more thorough comparison of Stator-inferred states to states inferred with NMF (see also below) and other methods inferring continuous states [9,10]; 3) showing Stator's state are recapitulated across independent datasets from the same biological system; and 4) testing Stator on transcriptomes taken from multiomic datasets (e.g., CITE-seq, ATAC-seq), where the other data modality can serve as an unbiased control for discovery of states that aren't captured from the RNA alone with clustering or other methods.

These are great suggestions. We have now performed the 4 analyses, as detailed below.

In previous work, the core MFI estimator of Stator, for quantification higher-order dependence amongst variables, was demonstrated to successfully recover ground truth results in extensive sets of simulation studies (Beentjes and Khamseh, 2020).

- 1) We performed 2 further simulations using the RP dataset: (a) One in which a fictitious d-tuple of genes was induced and then detected by Stator and, (b) another in which a cell state (blood contamination) was removed from the Stator run and, indeed, was then not detected. Specifically, since we are interested in biological reproducibility, rather than technical reproducibility, we do not demand that a state is exactly reproduced in terms of its defining d-tuples, but rather that the list of genes that defines a state and their expression are sufficiently similar. For this, we refer to a gene together with its binary expression as a 'gene state'. Given a Stator state S from the original run, and a state T arising from a new data set, we say that S and T are similar if they share at least two gene states among the 30 most commonly occurring gene states in each state. If, however, the Stator state is made up of only three gene states, then sharing a single gene state is sufficient to be considered similar. For example, if S and T are both composed of two d-tuples: $S = ([A+, B+, C-], [A+, C-, D+])$ and $T = ([C-, E+, F-], [A+, G-, H+])$, then together they are composed of the eight gene states $(A+, B+, C-, D+, E+, F-, G-, H+)$, of which $A+$ and $C-$ are shared. Since S and T share at least two gene states, we conclude that they are similar. If, instead, $T = ([C-, E+, F-], [G-, H+, I+])$, then they are not similar. Finally, if $T = ([C-, E+, F-])$, then T is composed of just three gene states, so the single shared gene state $C-$ suffices to make S and T similar. Having done this, we verified both (a) and (b).
- 2) (i) Comparisons of mouse embryonic RP Stator states with 3 other methods: NMF, LDVAE and LDA.

“Methodologically, we note that these 3 methods involve subjective choices of (hyper)-parameters and lack uncertainty quantification; also, they do not perform hypothesis testing against an appropriate null distribution and thus, they do not support valid multiple hypothesis correction on the predicted gene modules. Accurate interpretation of these gene modules becomes subjective when there are spurious gene modules that

are due to “noise”, which would have been “statistically zero” had appropriate uncertainty quantification been applied. For NMFs, for example, these effects could have led to several gene modules for which we could not assign a biological label based on the previously used criteria (marker genes reference set, described in Fig 7A caption). Many states are indeed replicated by these methods. However, in general, there is greater specificity for Stator over NMF, LDVAE, LDA, i.e., there is higher relative expression of gene markers that were previously-defined for a cell state as defined by Stator relative to an equivalently-labelled state defined by NMF, LDVAE, or LDA (Table 1).”

Response Fig3. Fig 7: (A) Heatmap presenting the $-\log_{10}FDR$ from the hypergeometric test comparing the co-labelling of cells by Stator states and NMF states. To biologically annotate NMF states, marker genes were taken from 4 sources of literature: Supplementary Table 2; IEGs ((Wu et al. (2017) as previously); cell cycle ((Tirosh et al. (2016), as previously); RGC-like marker genes at E17.5 ((Yuzwa et al. 2017) as previously). The biological labelling of NMF states has been performed exactly as before for Stator, although now more generously using 2 (rather than 3) or more marker genes for non-Stator methods. NMF recapitulates many of the Stator states. (B) For ciliated cells, Stator identified 4 ciliated cell subtypes, among which one (#18) shows

higher expression of *Foxj1*, an ependymal cell marker gene. These 4 subtypes were not distinguished by NMF. (C-E) Scatter plot of genes' differential expression for NMF, LDVAE and LDA vs Stator states. Y-axis: genes' log₂ fold expression change between cells in a Stator state and all other cells; X-axis: genes' log₂ fold expression change between cells in a state or type identified by NMF/LDVAE/LDA and all other cells. Gene symbol colour reflects cell state or type, as annotated using literature marker genes (as above). Genes' differential expression is highly significantly correlated between Stator and (C) NMF, (D) LDVAE or (E) LDA. However, Stator can detect cell states or types with mostly higher differential expression of known marker genes. In general, there is higher specificity for Stator in most categories over NMF, LDVAE and LDA (Table 1).

	Stator vs NMF	Stator vs LDVAE	Stator vs LDA
Activated	27% (18/67)	66% (44/67)	48% (32/67)
Contamination	100% (4/4)	100% (4/4)	100% (4/4)
Cilia	67% (2/3)	100% (state not detected by LDVAE)	100% (3/3)
G1/S	86% (30/35)	97% (34/35)	97% (34/35)
G2/M	57% (21/37)	78% (29/37)	92% (34/37)
Neuron	75% (9/12)	92% (11/12)	100% (state not detected by LDA)
RGC-like	31% (31/101)	100% (state not detected by LDVAE)	62% (63/101)
APC2	0	50% (2/4)	100% (state not detected by LDA)

Table 1: The percentage of genes for each cell state label that show greater expression fold-change in Stator over NMF, LDVAE and LDA, respectively.

(ii) NMF comparison has now been applied to all 3 datasets: liver cancer (as before), mouse RPs (new), and liver disease (new). On the first dataset, Stator's higher resolution for revealing expression programmes that predict patient survival had been shown in the original submission. The text below has been added to the manuscript describing the cirrhotic vs healthy liver dataset NMF comparison:

"NMF analysis was performed on the liver cirrhosis dataset from (Ramachandran et al. (2019)) using the NMF procedure in (Barkley et al. 2022), described in the Methods section "NMF procedure and gene modules". Of the 25 NMF modules identified, 5 were significantly enriched in endothelial cells. Of these 5 modules, only one could be annotated based on ≥ 2 marker genes used in the original submission (Przysinda et al., 2020; Trimm and Red-Horse, 2023): module m_{FCN3} genes included *CLEC4G* and *STAB2*, both markers for liver sinusoidal endothelial cells. The other 4 NMF modules could only be labelled as endothelial cells of unknown type. By comparison, using the same data, 7

Stator states could be labelled with ≥ 3 marker genes from (Przysinda et al., 2020; Trimm and Red-Horse, 2023) (Figure 5E).”

- 3) The manuscript offers an analysis of the reproducibility after random down sampling, where subsets of downsampled cells are disjoint. This was done twice: once for RPs (old Fig. S7C,D, updated figure number S8C,D) and once for neurons (old Fig. S13C,D, updated figure number S15C,D). Reference to these panels was inadvertently missing from the body of the text, for which we apologise. The main body of the text now refers to these panels and Methods “Stator state projection to disjoint data” contains further information on both the projection of states between datasets, and the heatmap visualisation. Additionally, this text explains the benefits of state projection in various applications (e.g., disease progression and cross-species analyses).
- 4) (Reviewer’s items 3 & 4) We projected Stator RNA states from the E18 merged RP+neuron dataset, analysed previously, into an independent dataset of 5,000 cells (3,343 cells after quality control) acquired in the same biological condition that has an additional modality, namely scATAC-seq:
<https://www.10xgenomics.com/datasets/fresh-embryonic-e-18-mouse-brain-5-k-1-standard-2-0-0>.

We have added a new subsection to Supplementary Material “Comparison with multimodal data”:

“We projected Stator states from the original merged RP and neuron dataset into a second dataset’s 5,000 cells (3,343 cells after quality control) using only their scRNA-seq data (10XGenomics, 2021). This yielded states that were predicted by projection rather than being predicted by re-running Stator. Using single cell chromatin accessibility data from these same 3,343 cells we then showed that differential mRNA expression and differential chromatin accessibility between two states can occur for the same genes (Fig S12). This showed: (1) that the transcriptomic heterogeneity identified in the first dataset (RPs and neurons) is evident also in the second dataset, and (2) that this heterogeneity is evident concordantly in both altered gene expression and open chromatin.”

Response Fig4. FigS12: (A) Differential mRNA expression and differential chromatin accessibility between two states tend to occur for the same genes in the same single cells. Stator states #93 (labelled as neurons) and #44 (RPs), from the merged RP and neuron dataset, were projected into a multi-omic (scRNA-Seq and scATAC-Seq) dataset from the same condition, E18 mouse brain. X-axis: Up- or down-regulated s2s-DEGs between cells labelled by projection as state #93 over state #44 (as an illustrative example) or non-expressed genes. Y-axis: Genes' ATAC-seq peaks that are higher or lower between these two states' cells, or else unchanged. The heatmap indicates the enrichment of genes that are both differentially expressed between these states' cells and contain differentially open chromatin between these same cells. Differential peaks were extracted via the Seurat function "FindMarkers", then the closest genes to the genomic regions of these differential peaks were extracted using the "ClosestFeature" function.. For FDR control, the BH procedure was applied (Benjamini and Hochberg, 1995). Values in parantheses represent numbers and percentages of genes (X-axis) and peaks (Y-axis), respectively. (B) In green: Pairs of projected states that show significant enrichment of genes that are both up-regulated in expression and have increased chromatin accessibility in the same cell. Rather than a single pair of states (as in (A)), the enrichment of up-regulated expression and increased chromatin accessibility was tested for all pairs of states in cells labelled by projection from the merged RP and neuron dataset into the multi-omic dataset. Concomitant gene up-regulation and increased chromatin expression occurs more frequently between pairs of interneuron projected states, than between embryonic RP states or between medium spiny neuron (MSN) states, indicating that there is greater molecular diversity among these interneuron states. The original merged (neuron and RP) dataset and the multiomic data set may not capture molecular heterogeneity equally. Similarly, the multiomic dataset may not capture expression and chromatin heterogeneity equally.

The following figure shows transcriptional heterogeneity in the joint neuron and RP dataset, through s2s-DEGs, as been added to Fig. S11 (as panels B and C) in the manuscript. Embryonic RPs have evidence of considerable transcriptional heterogeneity.

Response Fig5. Fig.S11: (A) Heatmap showing numbers of s2s-DEGs for the merged RP and neuron dataset. Colour indicates the number of DEGs. Stator state pairs with different labels have more s2s-DEGs than those with the same label, as expected. (B) As in (A), but for embryonic RP states only. This shows considerable transcriptional heterogeneity among these Stator RP states.

Testing Stator's ability to recover the same states across independent datasets is closely related to testing its ability to recover the same states in downsampled data, which is crucial due to Stator's input limits as described above.

This issue is addressed above: items 3 and 4.

Notably, Stator is already tested against NMF on 4 liver tumor samples. These samples are a subset of 19 samples that were previously analyzed together with NMF to recover recurrent cell states across diverse cancer types [11]. Why were only 4 out of the 19 samples analyzed? Could the authors analyze all 19 and compare the states recovered by Stator to the states recovered by the original study with NMF?

Due to our research interests, we initially chose to focus on the HCC samples from this dataset. Extending this to all 19 samples across multiple cancer types would be an extensive project on its own, and so lies beyond the scope of this work. Nevertheless, we have also now projected Stator states from HCC (30 states) into cholangiocarcinoma cells. This did not yield strong concordance between equivalent states as labelled in the original publication. For example, the HCC Stator states enriched in the original 'Cycle' label when projected into cholangiocarcinoma cells found 4 Stator states that were also enriched in 'Cycle' labelled cells. Whether this lack of

strong concordance can be explained by how the original NMF method and/or Stator was applied, or by low power to assign Stator states by projection, remains unknown. Due to these substantial uncertainties, and to keep to our timeline agreed with the Editor for returning our revised manuscript, we were unable to continue this line of investigation.

Minor comments

1. Figure 1b - red and green edges / circles appear but their meaning is explained only in Fig. S1. Similarly, meaning of red-blue is not explained in Fig1c.

Well spotted. We have now added the following to the Fig. 1 caption: "Green edges denote positive values, red edges denote negative values, the larger green triangle represents a positive 3-point dependence" and "Red and blue represent even and odd numbers of expressed genes (equal to 1) in the MFI."

2. Figure 2a and the similar ones later: the font is too small. In my opinion, it is redundant to show PCA plots for all the Stator states because of the limited value of PCA proximity evaluation in that context, as described above (I expect the first PCs to capture ~1% of the variance). The authors should denote the captured variance on PCA axes to allow the reader to better evaluate these plots. Is there a meaning to the order by which state panels appear (currently, 1, 10, 14, 24, ...)?

Following the reviewer's suggestion, we have now removed the many PC plots previously shown in Figures 2a and 4a, replacing them with a barplot showing clustering distribution across the states. Only 3 PC panels are kept to allow visualisation that cells in different Stator states are indeed distinctive. PC variation is now added to the Figure (~6.2% for RPs PC1, and 5.6% for neurons). This variation increases as the cell population becomes more heterogeneous as expected: merged RP and neurons show a PC1 variation of 10%, as above.

3. Figure 6 legend typo: "Nevertheless, in the main these cells are only labelled" - in the main what?

Agreed. Now amended to "Nevertheless, most cells are labelled only as single Stator states."

4. p. 2: "Automatic cell annotation methods, such as CellTypist (Condo et al. 2022), foundation (Cui et al. 2023), and deep learning models, are not intended..." - the reference to "deep learning" is too general, especially with no studies cited.

The reviewer is completely right. Reference to deep learning has now been removed in this sentence.

5. Methods 4.3: "Dice similarity between X and Y is defined as:" should be a quantity that increases when X,Y are similar but equation 6 gives instead the Dice distance (decreases when X,Y are similar).

Now changed to "Dice dissimilarity" throughout.

6. Methods: no details provided on details of how NMF and Milo were used, parameters provided etc.

We have now added the following subsection to Methods: "NMF procedure and gene modules":

Following methods used in (Barkley et al. 2022) we applied NMF to the mouse embryonic RP dataset (data as in Fig. 2) and the liver disease dataset (data as in Fig. 5). For the liver cancer dataset, we re-use the cell annotation provided, following application of NMF by (Barkley et al. 2022). The input to NMF is the normalized centered expression data of the 2000 HVG, with all negative values set to zero. Specifically, we applied 'nsNMF' within a reasonable initial range (10 to 30) for the number of components to be identified, using the R package NMF (Gaujoux and Seoighe, 2010). The output of NMF is a (gene-by-component) weight matrix, whose entries represent the contribution of a gene to that component, and a (component-by-cell) coefficient matrix, whose entries represent cell usage, defined as how much each set of gene modules is 'used' by each cell in the dataset (Kotliar et al., 2019). To construct non-overlapping gene modules, we ranked genes using the algorithm described in (Barkley et al. 2022) via two lists: list 1 ranks the genes' contribution to each component, and list 2 ranks the components to which each gene contributes. For each component, genes were added in the order of their rank (list 1), until a gene was reached that contributed more to a second component (list 2). Components with fewer than 5 genes were removed, and the procedure repeated. We obtained the gene modules for each number of initial components (10-30). The number of gene modules thus never exceeds the number of components. The largest initial number of components was selected for downstream analysis, for which the number of gene modules equals the number of components. For mouse RPs we obtained 27 gene modules; for the liver disease dataset, we obtained 25 gene modules.

Once gene modules were predicted, we then scored each cell based on the expression of these gene modules' genes as before (Barkley et al. 2022) for each module, we generated 1000 random gene lists of the same number whose genes have similar expression levels (defined by the `MakeRand` function in `seurat_functions_public.R` of (Barkley et al. 2022)). Then, for

each cell, the average centered expression of these random gene lists and the NMF gene module were calculated. We computed a p-value as the proportion of random gene lists that have a higher value of this expression than the corresponding value for the given gene module. The score was then calculated as $-\log_{10}(\text{p-value})$ and rescaled linearly to [0,1]. We only considered a gene module to be expressed in a cell if the corresponding score exceeded 0.5. Finally, a cell was assigned to the highest scoring module. We then performed a hypergeometric test, controlling for FDR, for the overlap between cell types or states inferred by Stator and NMF.”

Regarding Milo hyperparameters, we did not apply this method. Instead, we compared with labels generated by authors of Milo on the same dataset (Dann et al., 2022).

Reviewer #2:

Jansma and Yao propose a new tool of revealing underlying biology of single cell datasets, not at the level of cell types, but instead focusing on higher order gene expression interactions that represent phases/states. This is an interesting complement to the cell type annotation-driven single label approaches currently used, and has the potential to further aid the interpretation of single cell data.

Major comments:

1. The validity of some of the resolved states are not convincing, as a handful or even single DE genes are used to justify them. There is a concern they may be overclustering for noise (such as dropout).

Stator predicts cell (sub)types or states using $m \geq 3$ previously-discovered marker genes. These marker genes are biologically coherent: they indicate the same (sub)type/state, rather than inconsistent ones.

As Reviewer 1 notes “It is admittedly difficult to evaluate a method aimed at uncovering novel cell states due to the lack of a ground truth.” Nevertheless, **7 separate lines of evidence** support Stator’s state predictions being biological and not, for example, due to an unexpected source of noise:

(1) Results from 2 simulations are not consistent with technical artifacts (Response to Reviewer 1, issue 3, above).

(2) Stator state predictions were replicated using other methods, most often with NMF (see Response Fig3. (Fig7 in manuscript) above).

(3) Transcriptional or technical noise is not expected to yield large numbers of differentially expressed genes between states. Nevertheless, this is what Stator yields, for example >10 between pairs of identically-labelled states (Response Fig5. FigS11, above).

(4) Genes that are significantly differentially expressed (after FDR control) between differently labelled cells are commonly well-known marker genes for *the same* cell (sub)type or state. Noise and imperfect control of false discoveries would result in known marker genes being randomly assigned to states. To assign cell state labels, we required differential expression of $m \geq 3$ well-known marker genes (not $m=1$ as stated). We now provide a step-by-step description of label assignment, Methods “Assigning labels to Stator States”. Note from Supplementary Tables 1 and 2 that m exceeds 3 for many Stator states.

(5) Projection also supports biological state replication. Projection is when cells from a second disjoint dataset are tagged as having ≥ 1 d-tuples that define a state in a first disjoint dataset (referred to as “original” in the manuscript, see new Methods section “Stator state projection to disjoint data”). Projected and original states typically yield differentially expressed marker genes (beyond d-tuple genes) that indicate the same state label. Replication was undertaken twice: once for RPs (Fig. S8C,D) and once for neurons (Fig. S15C,D) – links to these Figures were absent in the original submission, for which we apologise.

(6) Projecting Stator states to a second, unseen multi-omics (scRNA-seq+scATAC-seq) dataset, verified the predicted heterogeneity through orthogonal scATAC-seq peaks (please see response to Reviewer 1’s items 3 & 4, and Response Fig4. FigS12, above).

(7) For the liver disease dataset, 3 of 7 Stator endothelial cell (EC) Stator states have already been validated by cell staining, flow cytometry and immunofluorescence (see response to Reviewer 3 “Validation”, below).

2. A large amount of effort focuses on discussing cell cycle phase, whose inference has been done perhaps less elegantly for years now either through gene set expression scoring or other tools such as Tricycle. The authors have not convincingly demonstrated that Stator reveals similar or cleaner cycle states than previously used approaches.

For the same embryonic RP dataset, Stator recapitulates Tricycle’s cell cycle phase predictions (Response Fig6.), below:

Response Fig6. Fig8: (A) Cell cycle phases predicted by Tricycle (left) and Seurat (right) for the embryonic RP dataset. Seurat requires all cells to belong to one of the 3 phases shown and appears to have lower resolution than Tricycle. (B) Stator's 7 states recapitulate Tricycle's 5 cell cycle phase predictions. Heatmap showing the $-\log_{10}FDR$ value of the hypergeometric test comparing the overlap of cells from cell cycle relevant Stator states with cells from cell cycle phase identified by Tricycle.

We have now added this figure alongside the following text to Methods "Comparison with other methods":

"Tricycle makes use of pre-defined gene markers for G1/S, S, and G2/M phases from Ref [15] of (Schwabe et al., 2020). By contrast, state definition by Stator is fully data driven and so it need not classify by these phases, or use these (or any) pre-specified gene markers. As a result, it maps cells to a continuum of cell cycle sub-phases, defined and ordered by the expression (Figure 2B) and non-expression (Figure 3B) of ab initio-discovered genes that are known markers for cell cycle phases."

3. 50+ potential states can be difficult for investigators to navigate and interpret. Can the authors provide guidelines for merging and annotating them, outside of the shiny app interface?

Thank you for this suggestion. We have now added a new subsection to Methods "Assigning labels to Stator states". This provides a 3-step guide for how to biologically annotate Stator states. This new section is now referenced in Fig 1G caption.

Minor comments:

1. While computation time explains the recommended upper limit of using 1000 highly variable genes, has performance (number and interpretability of the states) been assessed for using less or more genes? Has the author considered subsampling (or Seurat "sketching" type of informed subsampling) of cells to easy computational load?

The method's robustness to an increasing number of genes for identifying significant MFIs is investigated more fully elsewhere (Abel Jansma, PhD thesis (Jansma, 2023b) Chapter 4, "Robustness to gene selection"), which we now cite. For completeness we also now include in the resubmitted manuscript a new supplementary figure (*Response Fig7*, Fig S6, below). This shows that the number of genes in the Markov Blanket (MB) stabilises at ~15 for increasing numbers of HVG; the MB is the smallest set of genes \underline{G} conditional on which Gene i and Gene j are independent of all other genes. This means that the number of MFI is also stable at higher HVG number because stochasticity only arises due to variation in MB genes. To explain these points the revised manuscript now includes: "as demonstrated in (Jansma, 2023b), when increasing the number of HVGs from 300 to 700, all significant 1-point interactions, >98% of 2-point interactions, and approximately 96% of 3-point interactions kept the same sign and 95% confidence interval."

In response to Reviewer 1, we performed additional robustness analyses (see above). These included (i) adding a fictitious d-tuple/state, re-running Stator and observing that it was successfully detected; and, (ii) removing cells detected by Stator as blood contamination, re-running Stator on all remaining cells, and observing that the state, as expected, disappears, while leaving all other states detected.

Response Fig7. FigS6. The mean size of the Markov blanket stabilises to within the standard error on the mean when calculations include at least 200 most highly variable genes, as does the 95th percentile around the median. This plot here was generated using neurons from the mouse brain dataset.

With regards to cell sampling strategies (restricting to a given broadly defined cell type, or randomly downsampling, or Sketching), these can indeed be undertaken prior to using the Stator pipeline, depending on the researcher's specific question of interest.

2. The authors should be applauded for doing the work to package the tool in a Nextflow workflow and docker images, which will facilitate its adoption. Please also add a bit more guidelines on setup and launch of the pipeline on the Github page for users less experienced. I

also could not see some of the figures in the accompanying shiny app with HCC demonstration data (GO, rrvgo). Please look into whether this is an issue with code or user error on my part (and perhaps adjust UI/tutorials to help with confusion)

We have now made a more extensive vignette available on GitHub for the nextflow pipeline. We have also further improved the tutorial for the R shiny app, which can be found here. We have checked the app on our servers, and it seems to be currently functional. Other collaborators have managed to perform analyses using the server. Please note that the “Submit” button needs to be clicked when switching to a new tab; a note has been added to the GitHub version of the app stating this. A second possible explanation could be online dependencies on Ensembl which may at times not be responsive. This is outside our control, but can be resolved by revisiting the server.

3. For cluster 17 in Supplemental Table 3 and 4, 12 DE genes yielded 98 significant GOBP terms. Should the single hit GO terms even be reported? (keep more than 3 genes)

Thank you. We agree, and have now removed single GO terms from the updated version of the Table.

Reviewer #3:

In this manuscript, the authors introduce a novel computational method, Stator, designed to identify cell states or types within single-cell RNA-seq datasets. Stator operates under the assumption that cell states can be delineated by the coordinated expression of specific and small sets of genes. Initially, Stator identifies candidate gene combinations marking cell states through a method previously developed by the same authors (MFI), subsequently assigning cells to states based on the presence or absence of these identified gene combinations within each cell.

The authors apply Stator across multiple datasets, claiming, for example, the capability to discern several biologically relevant states (e.g., phases of the cell cycle) within the same cell types.

Characterizing cell types or states is a significant challenge, highlighted by the proliferation of algorithms. Most existing methods assume that cells belonging to the same type/state exhibit greater transcriptomic similarity, and, hence, they rely on cell clustering techniques.

Stator distinguishes itself from these approaches through two conceptual distinctions: firstly, it aims to identify higher-order correlations in gene expression, specifically targeting coordinated expression patterns of up to 7 genes. Secondly, it avoids traditional cell clustering methodologies, thereby enabling the identification of cell states that may be shared across different cell types, not necessarily proximal within the transcriptomic space.

While the concepts introduced by Stator are interesting, particularly in the context of analyzing scRNA-seq datasets marked by sparsity and large cell populations, this manuscript exhibits important limitations. In particular, I am concerned about the validation of the results and the

apparent underestimation of the critical aspect of cell state annotation, which might limit Stator's applicability.

Major points

- ****Validation.**** A fairly large number of states is identified in all the datasets, but there's little validation shown beyond GO terms and manually selected marker genes. More independent validation is needed to convince the reader that many states (many having the same annotation) are not deriving from overfitting but rather represent "real" cell states. For instance, Stator could be run on datasets where cells have been previously labelled, with, e.g., FACS sorting, etc.

In our response to Reviewer 2 (above) we collected **7 separate lines of evidence** that Stator states are biologically valid.

Here, we provide further details of the last of these. Specifically, 3 of the 7 Stator endothelial cell (EC) Stator states had previously been validated by cell staining, flow cytometry and immunofluorescence. Equivalents of these states were previously identified by Ramachandran et al. 2019 and then validated using the following methods:

(1) Lymphatic EC *PDPN+*, *FOXC2+* Stator state #5 (Figure 5E) was identified in Ramachandran et al. 2019 by immunofluorescence as *PDPN+*, *FOXC2+* lymphatic cells (i.e. Endo(2)); their Extended Data Fig. 8g,j).

(2) Cirrhosis-enriched Stator state #33 (*ACKR1+*; our Figure 5C,E) was identified in Ramachandran et al. 2019 by cell staining, flow cytometry and immunofluorescence as a *ACKR1+* cirrhosis-restricted EC type (i.e. Endo(7)); their Figure 4E; Extended Data Fig. 8d,j).

(3) Stator state #34 (*CLEC4G+*, *FCGR2B+*, *MRC1+*, *STAB1+*, *STAB2+*; our Figure 5E) was identified in Ramachandran et al. 2019 by immunofluorescence (their Figure 4D) as [*CLEC4G+*, *FCGR2B+*, *MRC1+*, *STAB1+*, *STAB2+*] liver sinusoidal endothelial cells (LSEC), restricted to perisinusoidal cells within the liver parenchyma (i.e. Endo(1)); their Supplementary Note 3).

To reflect these validations, the following sentence has now been added to the Main text: "Equivalents to Stator states #5, #33 and #34 were found by Ramachandran et al. (2019) (i.e., Endo(2), Endo(7) and Endo(1)) and then validated by cell staining, flow cytometry and/or immunofluorescence."

- ****Annotation.**** In many cases, how the authors established the annotation of cell states is unclear. For example, claims like the following: "The 7 states' s2o-DEGs were predominantly cell cycle markers, confirming them as cell cycle states." (page 7) should be more quantitative (what's the fraction of cell cycle markers? is there a statistically significant enrichment?).

Thank you, the following text has now been added to the manuscript:

"Specifically, of 45 genes that were among the 10 most significantly differentially expressed in 1

or more of the set of 7 s2o-DEGs, 40 (89%) were G1/S or G2/M stage marker genes according to Tirosh et al. (2016) (Table S5) and Fischer et al. (2016) (Table S10)."

Or, when they write "Demanding that at least 3 s2o-DEGs are known markers of an annotation" (page 7): does it ever happen that one state gets more than 3 markers for different annotations? how do the authors resolve the ambiguity in these cases? While cell type/state annotation from scRNA-seq is still an open problem and might not be the focus of this paper, I believe some more quantitative and objective criteria when performing annotation are needed to validate Stator's results.

To provide more detailed criteria, we now include a new subsection to the Methods "Assigning labels to Stator states". This provides a 3-step guide for how to biologically annotate Stator states (see also response to Reviewer 2, above). In answer to Reviewer 3's question, this guide states: "that Stator's data-driven approach can result in state i being labelled by both cell type *and* cell state, for example a radial glia-like cell in G2/M cell cycle phase" (Step 2). Rather than introducing ambiguity, this example shows how the same single cell can be annotated with 2 (or more) labels. When we applied Stator to the 3 datasets, no predicted state was characterised by multiple instances of ≥ 3 markers that would have labelled a mature cell type ambiguously (e.g., both *neuron* and *astrocyte*).

- **Comparison with other methods** . As the authors also notice, the comparison of Stator with CellTypist doesn't seem appropriate, given that CellTypist needs previously annotated cell states/types. Hence, I suggest either removing or drastically reducing the comparisons with CellTypist and replacing them with the use of other unsupervised methods (for instance, <https://www.nature.com/articles/s41467-023-43406-9> , or methods based on topic modelling, like hSBM <https://www.mdpi.com/2072-6694/14/5/1150>).

We agree: comparison with CellTypist is not relevant here (this point was also raised by Reviewer 1, above). Comparison with CellTypist was a request by the editor based on advice from the external advisory board. CellTypist comparisons with Stator have now been removed from the manuscript.

Following all reviewers' suggestions of comparisons with other methods, we have compared Stator with:

(i) For the embryonic RP dataset clustering, NMF, LDVAE, and topic modelling (NB: the referenced method has been used only for bulk RNA-seq, so instead we used another topic modelling method, LDA from scVI). Results are presented in Figs 7 and 8, and Table 1, in Methods "Comparison with other methods" in the manuscript and shown above in response to Reviewer 1.

(ii) NMF on the brain, liver disease, and liver cancer datasets. Again, we present their results above, in response to Reviewer 1. In summary:

- (a) these methods consistently replicate multiple Stator states;
- (b) there is greater expression specificity for Stator states over NMF, LDVAE or LDA states/modules, i.e., there is higher relative expression of known gene markers for the cell state as defined by Stator over the equivalent cell state defined by NMF, LDVAE or LDA;
- c) these other methods lack uncertainty quantification for the reported gene modules, which can result in reported modules not being biologically identifiable, (e.g., NMF results on embryonic RPs, see above). Had these methods benefited from uncertainty quantification and FDR control, similar to Stator, then some reported modules may then be “statistically zero” which would avoid false positives and over-interpretation of results.

These considerations are discussed in the Methods section “Comparison with other methods”. The new NMF results for liver disease data have been added to its corresponding section in the Main text.

- **Robustness with respect to number of starting genes.** How robust are the states (in terms of identity and number) if a different number or subsets of HVGs is chosen? This robustness analysis could also reduce the number of states identified, and could help focus on a smaller (but more robust) number of states

Please see our response to the same question from Reviewer 2 (above) and *Response Fig7*.

- **Blood contamination.** The authors identify a "blood contamination" issue with some of the datasets they analyze. I guess this could be due to ambient mRNA contamination, so I suggest trying to remove this before running their analysis.

We agree that ambient mRNAs likely contaminate an estimated ~1% of cells in 2 disjoint eRP cell sets (Figure S8). Rather than being a disadvantage in the data, detection of this contamination is an advantage, because (a) it allows cells with this contamination (and potentially others) to be identified in a data driven and reproducible manner (Figure S8), and (b) such contaminant states are independent from, and thus do not obfuscate, other states in the same cells.

- **Cell cycle.** Regarding the cell cycle, it might be interesting to compare their results with an ad-hoc cell cycle predictor algorithm (like the one found in the *seurat* or *scan* R libraries).

This comparison has been made for both *Seurat* and *Tricycle* (please see response to Reviewer 2, above, for details), and Fig. 8 in Methods. We note that *Seurat* is not a cell cycle state predictor, although it does output a "phase" in the following way: it first uses the `AddModuleScore()` to calculate module scores for S and G2/M phases for each cell using a list of

S and G2/M phase marker genes, before then assigning a cell cycle phase to each cell by (1) if S score > G2/M score, then the cell is assigned to S phase, (2) if S score < G2/M score, then it is assigned to G2/M phase, and then (3) if both scores < 0, then it is assigned to G1 phase. This means that all cells are required to be assigned a phase, leading to potential false-positives, for e.g., G1 phase cells.

Tricycle is a more specialised tool for predicting cell cycle phase. Stator recapitulates the cell phases detected by the Tricycle method (see Response Fig6 above). However, unlike Tricycle, Stator is data-driven: it does not rely on pre-defined marker genes to choose the 5 stages of the cell cycle defined by (Schwabe et al., 2020). Rather, Stator identifies a continuum of cell cycle (sub)phases through higher-order dependence of gene expression.

- **Cirrhotic or healthy liver samples analysis.** In the analysis of cells from cirrhotic or healthy liver samples (Figure 5), it seems like Hepatocytes, Mesothelia and pDCs don't correspond to any state that is exclusively enriched for that cell type (Figure 5B). For example, pDCs are enriched in the states 11, 15 and 45; but these states are also enriched with other cell types. Does this mean that Hepatocytes, Mesothelia and pDCs are not correctly identified by Stator?

The paragraph below has now been added to legend to Fig 5 (B):

“This panel implies that Hepatocytes, Mesothelia and pDCs do not correspond to any Stator state that is exclusively enriched for these cell types. Nevertheless, this is due to the conservative thresholds applied here. Expected correspondences emerge when thresholds are further relaxed (Dice dissimilarity > 0.5, log2FC > 1) where there are 6, 2 and 2 Stator states that are exclusively enriched for Hepatocyte, Mesothelia and pDC annotations, respectively.”

27th Sep 2024

Manuscript Number: MSB-2024-12214RRR

Title: High order expression dependencies finely resolve cryptic states and subtypes in single cell data

Dear Dr Khamseh,

Thank you for the submission of your revised manuscript to Molecular Systems Biology. I am pleased to inform you that we will be able to accept your manuscript pending the following final amendments and appropriate response to reviewers:

- 1) Please provide a .docx (or LaTeX) version of the manuscript text (including legends for main figures).
- 2) In the main manuscript file, please move the keywords to just below the Abstract (instead of as a footnote).
- 3) Please include in the Data availability section how Stator is available to users (link to code or how it can be found). This section should be formatted according to the example below:
"The computer code produced in this study are available in the following databases:
- Modeling computer scripts: GitHub (<https://github.com/SysBioChalmers/GECKO/releases/tag/v1.0>)
- [data type]: [full name of the resource] [accession number/identifier] ([doi or URL or identifiers.org/DATABASE:ACCESSION])"
- 4) Data not shown: We do not allow statements/conclusions with "data not shown". As per our guidelines, on "Unpublished Data" the journal does not permit citation of "Data not shown". All data referred to in the paper should be displayed in the main or Expanded View figures. Please remove from pages 7 and 43.
- 5) All Materials and Methods need to be described in the main text using our 'Structured Methods' format. According to this format, the Methods section includes a Reagents and Tools Table (listing key reagents, experimental models, software and relevant equipment and including their sources and relevant identifiers) followed by a Methods and Protocols section describing the methods, ideally using a step-by-step protocol format. The aim is to facilitate adoption of the methodologies across labs. Please download and fill our Reagents and Tools Table template (.docx), which you can find in our author guidelines: <https://www.embopress.org/page/journal/14693178/authorguide#structuredmethods>.
When submitting your revised manuscript, please do not include the Reagents and Tools Table in the Methods section of the manuscript but upload it as a separate file choosing the file type "Reagent Table".

An example of a Method paper with Structured Methods can be found here:
<https://www.embopress.org/doi/10.15252/msb.20178071>. "

- 6) Please place individual sections of the manuscript in the following order: Title page - Abstract & Keywords - Introduction - Results - Discussion - Methods - Data Availability - Acknowledgements - Disclosure and Competing Interests Statement - References - Figure Legends - Expanded View Figure Legends.
- Please ensure that the supplementary material is removed the manuscript file and that the figure legends are listed below the references, and keywords are placed below the abstract.
- 7) For the figures and figure legends, please take care of the following:
- Please remove all figures from main manuscript file and leave only main figure legends placed after the references.
- We have replaced Supplementary Information by the Expanded View (EV format). Supplementary Tables 1-27 should be provided as EV Tables (for less complex tables, not longer than one page) or EV Datasets (for more complex tables, longer than one page). Please provide one file per EV Table/Dataset. In each file, a description of the table/dataset should be provided in a separate tab. These EV Tables/Datasets should be uploaded individually with correct nomenclature and callouts, not only in the "Simple submission of all files as a ZIP - Zip file containing all files"
- All additional supplementary figures should be provided in a PDF called Appendix. Please include a Table of Contents (with page numbers) in the beginning of the Appendix. Appendix figures should be labeled and called out as: "Appendix Figure S1, Appendix Figure S2..." etc. Each legend should be below the corresponding Figure in the Appendix. For detailed instructions regarding expanded view please refer to our Author Guidelines: .
- Main figures and EV figures should be uploaded as individual, high-resolution files. Please check "Author Guidelines" for more information: <https://www.embopress.org/page/journal/17574684/authorguide#figureformat>
- Please note that information related to n is missing in the legend of Figure 7b.
- Please note that we require exact p-values to be reported either in the figure or figure legend. Currently exact p-values are not provided for Figure 7b.
- Please note that boxplot elements should be defined in the legend (mean/median, quartiles, etc). These are missing for Figure 7b.
- 8) Synopsis:
- Synopsis image and text: Please provide the synopsis image and text as 2 separate files. The Synopsis image should be uploaded as an individual file (550px width and 250-400px height, jpeg format). The Synopsis text should be uploaded as a separate .docx or LaTeX file.
- Please check your synopsis text and image before submission with your revised manuscript. Please be aware that in the proof stage minor corrections only are allowed (e.g., typos).

9) As part of the EMBO Publications transparent editorial process initiative (see our policy here: https://www.embopress.org/transparent-process#Review_Process), Molecular Systems Biology will publish online a Peer Review File (PRF) to accompany accepted manuscripts. This file will be published in conjunction with your paper and will include the anonymous referee reports, your point-by-point response and all pertinent correspondence relating to the manuscript. Let us know whether you agree with the publication of the PRF and as here, if you want to remove or not any figures from it prior to publication. Please note that the Authors checklist will be published at the end of the PRF.

10) Please provide a point-by-point letter INCLUDING my comments as well as the reviewer's reports and your detailed responses (as Word file).

I look forward to reading a new revised version of your manuscript as soon as possible.

Yours sincerely,

Poonam Bheda, PhD
Scientific Editor
Molecular Systems Biology

Reviewer #1:

The authors have adequately addressed my comments. At the editor's request, I gave particular attention to assessing the response to reviewer 2's comments, and I believe the authors addressed those as well.

Two issues that the authors may want to address in their final publication:

1. Simulations (supplementary section 8.3): The authors mention two new simulations on top of their previously published work, but as far as I can tell, they aren't referenced in the main text, and their results aren't shown in a supplemental figure.
2. The statements concerning Stator outperforming NMF-derived states should be qualified. NMF, like Stator, allows genes to participate in multiple programs and cells to belong to multiple programs (both with quantitative weights). However, for the purpose of comparing NMF and Stator, the authors forced each gene to belong to one NMF program, and subsequently each cell to a single NMF state (Methods 4.1).

Minor comments:

1. "projection of cells into K {less than or equal to} 50 dimension" (p. 1 and other places): Better qualify, the chosen K is typically around 50, but it's not a strict rule as the current wording implies
2. "Absence of expression for a gene is denoted by a minus" (p. 3): This sentence relates to Fig. 1E but seems out of context. Is it a copy-paste error?
3. "The majority (75.7%; $N = 9,044$) of cells occupy one or more state, and 34.7% (4,151) of cells are unique to a single state" (p. 5): Majority and minority sum to more than 100% and more than the number of cells in the data (11,950 according to the previous paragraph). Is that an error? Possibly the author did not unique the multi-state cell IDs?
4. "The second stage of dimensionality reduction, often used in scRNA-seq analysis pipelines to date, further reduces the K -dimensional space to 2 or 3 dimensions, often using Principal Component Analysis (PCA), t-distributed Stochastic Neighbor Embedding (tSNE) or Uniform Manifold Approximation and Projection (UMAP)" (p. 2): A misleading statement since tSNE and UMAP are computing further dimensionality reduction to 2d/3d whereas in PCA the authors reference merely a selection of 2 or 3 axes that had already been computed for visualization. Incidentally, the reference to PCA in this sentence like a last-minute addition because the rebuttal letter that cites the same paragraph does not mention PCA in the same context.
5. Fig 7a: please mark explicitly that rows and columns are NMF and stator states, respectively.

Reviewer #3:

The authors have addressed all my comments, as well as Reviewer 2's comments, in a satisfactory manner.

My only remaining suggestion is to include, as a supplementary figure, the results of the additional analysis performed on the liver samples using less stringent thresholds (see last question in my previous report). It would be interesting to compare the cell states identified by Stator under less versus more stringent thresholds, for example, using the Clustree package (<https://github.com/lazappi/clustree>).

We thank the editor and reviewers for further comments and suggestions that have improved the manuscript. Please find below a point-by-point response with a summary of changes made.

Manuscript Number: MSB-2024-12214RRR

Title: High order expression dependencies finely resolve cryptic states and subtypes in single cell data

Dear Dr Khamseh,

Thank you for the submission of your revised manuscript to Molecular Systems Biology. I am pleased to inform you that we will be able to accept your manuscript pending the following final amendments and appropriate response to reviewers:

1) Please provide a .docx (or LaTeX) version of the manuscript text (including legends for main figures).

2) In the main manuscript file, please move the keywords to just below the Abstract (instead of as a footnote).

Done.

3) Please include in the Data availability section how Stator is available to users (link to code or how it can be found). This section should be formatted according to the example below:

"The computer code produced in this study are available in the following databases:

- Modeling computer scripts: GitHub

(<https://github.com/SysBioChalmers/GECKO/releases/tag/v1.0>)

- [data type]: [full name of the resource] [accession number/identifier] ([doi or URL or identifiers.org/DATABASE:ACCESSION])"

The following has been added to the "Data Availability" section:

Stator's code and nextflow pipeline, as well as documentation on installation and a vignette, are available on GitHub at https://github.com/AJnsm/NF_TL_pipeline/tree/main.

The code for Stator's R Shiny App is available on GitHub at

<https://github.com/YuelinYao/MFIs>, with a Docker container image available on Dockerhub

<https://hub.docker.com/r/yuelinyao120/stator-app>. Additionally, the Stator App is hosted

at <https://shiny.igc.ed.ac.uk/MFIs/>.

All datasets analysed in this manuscript are publicly available and can be found using the following URLs.

- scRNA-seq, 10X 1.3 million Brain Cells from E18 Mice, accessible via the 10X website <https://www.10xgenomics.com/datasets/1-3-million-brain-cells-from-e-18-mice-2-standard-1-3-0>
- scRNA-seq and scATAC-seq, Fresh Embryonic E18 Mouse Brain (5k): Single Cell Multiome ATAC + Gene Expression Dataset by Cell Ranger ARC 2.0.0, accessible via the 10X website <https://www.10xgenomics.com/datasets/fresh-embryonic-e-18-mouse-brain-5-k-1-standard-2-0-0>.

- scRNA-seq, 10X cirrhotic and healthy human liver, with raw sequencing data available at Gene Expression Omnibus (GEO) under accession GSE136103.
- scRNA-seq, 10X liver cancer (HCC) dataset, available at GEO under accession GSE203612 and <https://github.com/yanailab/PanCancer>.

4) Data not shown: We do not allow statements/conclusions with "data not shown". As per our guidelines, on "Unpublished Data" the journal does not permit citation of "Data not shown". All data referred to in the paper should be displayed in the main or Expanded View figures. Please remove from pages 7 and 43.

For both Fig 2B and Figure EV3 (old pages 7 and 43) DEG between any two states, including the pairs with less than 25 co-labelled cells, had previously been provided in EV Datasets 5 and 12 respectively. We have now clarified this in the figure captions (text below) and provide a new Appendix Figure S8 for the number of co-labelled cells amongst cell-cycle states.

For Page 7: We have changed "State pairs with ≤ 25 cells are not shown" to "For clarity, state pairs with ≥ 25 cells are shown. DEGs between any two states, including state pairs with fewer than 25 co-labelled cells, are provided in EV Dataset 5. Appendix Figure S8 (panel A) additionally provides the number of co-labelled cells between any two states."

For page 43: We have changed "State pairs with ≤ 25 cells are not shown" to "For clarity, state pairs with ≥ 25 cells are shown. DEGs between any two states, including state pairs with fewer than 25 co-labelled cells, are provided in EV Dataset 12. Appendix Figure S8 (panel B) additionally provides the number of co-labelled cells between any two states."

5) All Materials and Methods need to be described in the main text using our 'Structured Methods' format. According to this format, the Methods section includes a Reagents and Tools Table (listing key reagents, experimental models, software and relevant equipment and including their sources and relevant identifiers) followed by a Methods and Protocols section describing the methods, ideally using a step-by-step protocol format. The aim is to facilitate adoption of the methodologies across labs.

Please download and fill our Reagents and Tools Table template (.docx), which you can find in our author guidelines: <https://www.embopress.org/page/journal/14693178/authorguide#structuredmethods>.

An example of a Method paper with Structured Methods can be found here: <https://www.embopress.org/doi/10.15252/msb.20178071>. "

A table detailing all software tools used in the manuscript is now included using the template provided.

6) Please place individual sections of the manuscript in the following order: Title page - Abstract & Keywords - Introduction - Results - Discussion - Methods - Data Availability - Acknowledgements - Disclosure and Competing Interests Statement - References - **Figure Legends** - **Expanded View Figure Legends**.

- Please ensure that the supplementary material is removed from the manuscript file and that the figure legends are listed below the references, and keywords are placed below the abstract.

7) For the figures and figure legends, please take care of the following:

- Please remove all figures from main manuscript file and leave only main figure legends placed after the references.

- We have replaced Supplementary Information by the Expanded View (EV format). Supplementary Tables 1-27 should be provided as EV Tables (for less complex tables, not longer than one page) or EV Datasets (for more complex tables, longer than one page). Please provide one file per EV Table/Dataset. In each file, a description of the table/dataset should be provided in a separate tab. These EV Tables/Datasets should be uploaded individually with correct nomenclature and callouts, not only in the "Simple submission of all files as a ZIP - Zip file containing all files"

- All additional supplementary figures should be provided in a PDF called Appendix. Please include a Table of Contents (with page numbers) in the beginning of the Appendix. Appendix figures should be labeled and called out as: "Appendix Figure S1, Appendix Figure S2..." etc. Each legend should be below the corresponding Figure in the Appendix. For detailed instructions regarding expanded view please refer to our Author

Guidelines: <https://www.embopress.org/page/journal/17444292/authorguide#expandedview>.

- Main figures and EV figures should be uploaded as individual, high-resolution files. Please check "Author Guidelines" for more

information: <https://www.embopress.org/page/journal/17574684/authorguide#figureformat>

The above edits have been made.

- Please note that information related to n is missing in the legend of Figure 7b.

The following has been added to the figure caption:

"18 (n=410), 19 (n=25), 20 (n=16), 23 (n=25)"

- Please note that we require exact p-values to be reported either in the figure or figure legend. Currently exact p-values are not provided for Figure 7b.

The following has now been added to the figure.

“***: $p=0.00065$

** : $p=0.0066$

* : $p=0.029$ ”

- Please note that boxplot elements should be defined in the legend (mean/median, quartiles, etc). These are missing for Figure 7b.

The following has now been added to the figure caption:

“In the boxplot, the median (middle quartile) marks the mid-point of the data and is shown by the line that divides the box into two parts, and the box itself indicates the range in which the middle 50% of all values lie.”

8) Synopsis:

- **Synopsis image and text:** Please provide the synopsis image and text as 2 separate files. The Synopsis image should be uploaded as an individual file (550px width and 250-400px height, jpeg format). The Synopsis text should be uploaded as a separate .docx or LaTeX file.

The Synopsis figure is provided as JPEG, with text in a separate .docx file.

9) As part of the EMBO Publications transparent editorial process initiative (see our policy here: https://www.embopress.org/transparent-process#Review_Process), Molecular Systems Biology will publish online a Peer Review File (PRF) to accompany accepted manuscripts. This file will be published in conjunction with your paper and will include the anonymous referee reports, your point-by-point response and all pertinent correspondence relating to the manuscript. Let us know whether you agree with the publication of the PRF and as here, if you want to remove or not any figures from it prior to publication. Please note that the Authors checklist will be published at the end of the PRF.

We agree with the publication of the PRF.

10) Please provide a point-by-point letter INCLUDING my comments as well as the reviewer's reports and your detailed responses (as Word file).

I look forward to reading a new revised version of your manuscript as soon as possible.

Yours sincerely,

Poonam Bheda, PhD

Scientific Editor
Molecular Systems Biology

Reviewer #1:

The authors have adequately addressed my comments. At the editor's request, I gave particular attention to assessing the response to reviewer 2's comments, and I believe the authors addressed those as well.

Two issues that the authors may want to address in their final publication:

1. Simulations (supplementary section 8.3): The authors mention two new simulations on top of their previously published work, but as far as I can tell, they aren't referenced in the main text, and their results aren't shown in a supplemental figure.

We thank the reviewer for pointing out the missing information. The following text supplementary figures have been added to the manuscript, and referenced at the end of Methods section “Deviating gene tuples (d-tuples)”.

“For simulation (a), a fictitious d-tuple of genes was induced by randomly choosing 90% of the cells where (*Cited2*, *Basp1*, *Fhl1*) = (1, 0, 1) and changing this to (1, 1, 1). Stator was then re-run on the new data which included the original set of genes together the induced tuple. D-tuple (*Cited2*, *Basp1*, *Fhl1*)=(1,1,1) was recovered as part of state 14 in Appendix Figure S7 (panel A) with FDR < 0.05 and > 6-fold enrichment, at the default Dice dissimilarity threshold. For simulation (b), the cells tagged as blood contamination in the original run were removed and Stator was re-run. The results in Appendix Figure S7 (panel B) indicate that the blood contamination state does not map to any states in the run.”

Reproducibility of states: Artificial state

Reproducibility of states: Blood removed

Appendix Figure S7. Simulations indicating an artificially induced d-tuple (panel A, left) and an artificially removed state (blood contamination, panel B, right), are recovered and removed by Stator, respectively. In each panel the left-hand blue boxes represent the original states, and the right-hand blue boxes represent the new Stator run. The lines map the states as described in the text. The lines map to multiple states, as expected, since Stator allows cell to exist in multiple states. (A, left) A fictitious d-tuple of genes is induced by randomly choosing 90% of the cells where $(Cited2, Basp1, Fhl1) = (1, 0, 1)$ and changing this to $(1, 1, 1)$. Stator is rerun, and this duple is recovered as part of state 14, with $FDR < 0.05$ and > 6 -fold enrichment, at the default Dice dissimilarity threshold. (B, right) Cells tagged as blood contamination in the original run were removed and Stator was rerun. The results indicate that the blood contamination state (bottom row) does not map to any state in the new run, as expected.

2. The statements concerning Stator outperforming NMF-derived states should be qualified. NMF, like Stator, allows genes to participate in multiple programs and cells to belong to multiple programs (both with quantitative weights). However, for the purpose of comparing NMF and Stator, the authors forced each gene to belong to one NMF program, and subsequently each cell to a single NMF state (Methods 4.1).

As requested by the reviewer, we have now qualified the NMF approach taken by adding the following text to the end of section “4.10. NMF procedure and gene modules” of the manuscript:

“For the purpose of comparing Stator with NMF, we followed the approach in \citep{CancerCellStateBarkley2022}, in the step-by-step manner described above. In this approach, each gene is restricted to belong to one NMF programme, and subsequently each cell to a single NMF state. Alternative NMF methods exist for selecting gene modules and for assigning cells to different (multiple) modules, although there is no definitive approach for gene selection and cell assignment \citep{wang2022non, kim2003subsystem, gavish2023hallmarks, kinker2020pan, kotliar2019identifying}.”

Minor comments:

1. "projection of cells into K {less than or equal to} 50 dimension" (p. 1 and other places): Better qualify, the chosen K is typically around 50, but it's not a strict rule as the current wording implies

This sentence was changed to "projection of cells into e.g., $K \leq 50$ dimensions (or e.g., K determined via a scree plot for Principal Component Analysis (PCA)) ..."

2. "Absence of expression for a gene is denoted by a minus" (p. 3): This sentence relates to Fig. 1E but seems out of context. Is it a copy-paste error?

There are genes with a minus on the branches of the tree in Fig 1E. To make this clearer, we have added the following to the above sentence, "e.g., G_1^- and G_2^- on the rightmost branch in Fig. 1E."

3. "The majority (75.7%; $N = 9,044$) of cells **occupy one or more state**, and 34.7% (4,151) of cells are unique to a single state" (p. 5): Majority and minority sum to more than 100% and more than the number of cells in the data (11,950 according to the previous paragraph). Is that an error? Possibly the author did not unique the multi-state cell IDs?

The majority (75.7%; $N = 9,044$) of cells, which occupy **one or more states**, include the 4,151 cells that are unique to a single state, i.e., the latter is a subset of the former, and thus will not add up to 100%.

4. "The second stage of dimensionality reduction, often used in scRNA-seq analysis pipelines to date, further reduces the K -dimensional space to 2 or 3 dimensions, often using Principal Component Analysis (PCA), t-distributed Stochastic Neighbor Embedding (tSNE) or Uniform Manifold Approximation and Projection (UMAP)" (p. 2): A misleading statement since tSNE and UMAP are computing further dimensionality reduction to 2d/3d whereas in PCA the authors reference merely a selection of 2 or 3 axes that had already been computed for visualization. Incidentally, the reference to PCA in this sentence like a last-minute addition because the rebuttal letter that cites the same paragraph does not mention PCA in the same context.

The reviewer is correct about this. We have removed "Principal Component Analysis (PCA)" from the above sentence.

5. Fig 7a: please mark explicitly that rows and columns are NMF and stator states, respectively.

Great point, this is now added to the figure, X-axis: Stator state, Y-axis: NMF.

Reviewer #3:

The authors have addressed all my comments, as well as Reviewer 2's comments, in a satisfactory manner.

My only remaining suggestion is to include, as a supplementary figure, the results of the additional analysis performed on the liver samples using less stringent thresholds (see last question in my previous report). It would be interesting to compare the cell states identified by Stator under less versus more stringent thresholds, for example, using the Clustree package (<https://github.com/lazappi/clustree>).

A new Appendix Figure S13 has been added at the lower dice distance for this dataset.

31st Oct 2024

Manuscript number: MSB-2024-12214RRRR

Title: High order expression dependencies finely resolve cryptic states and subtypes in single cell data

Dear Dr Khamseh,

Thank you again for sending us your revised manuscript. We are now satisfied with the modifications made and I am pleased to inform you that your paper has been accepted for publication.

Yours sincerely,

Poonam Bheda, PhD
Scientific Editor
Molecular Systems Biology
